

# A panoply of Schwinger-Keldysh transport

**Kristan Jensen[1]⋆, Raja Marjieh[2]†, Natalia Pinzani-Fokeeva[3]‡ and Amos Yarom[2]◦**

**1** Department of Physics and Astronomy,
San Francisco State University, San Francisco, CA 94132, USA
**2** Department of Physics, Technion, Haifa 32000, Israel
**3** Institute for Theoretical Physics, KU Leuven
Celestijnenlaan 200D, Leuven B-3001, Belgium

⋆ kristanj@sfsu.edu † rajamarjieh@gmail.com
‡ natascia.pinzanifokeeva@kuleuven.be ◦ ayarom@physics.technion.ac.il

## Abstract

We classify all possible allowed constitutive relations of relativistic fluids in a statistical mechanical limit using the Schwinger-Keldysh effective action for hydrodynamics. We find that microscopic unitarity enforces genuinely new constraints on the allowed transport coefficients that are invisible in the classical hydrodynamic description; they are not implied by the second law or the Onsager relations. We term these conditions Schwinger-Keldysh positivity and provide explicit examples of the various allowed terms.

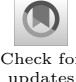

# 1 Introduction

Relativistic hydrodynamics is an effective theory capable of describing diverse phenomena relevant in heavy ion collisions, cosmology and astrophysics, and in condensed matter systems such as graphene. Until recently, the equations of motion of hydrodynamics were constructed so as to be the most general ones possible compatible with the symmetries of the problem, a local version of the second law of thermodynamics, and Onsager relations which encode certain CPT properties of correlation functions.

In the hydrodynamic theory the conserved currents of the underlying microscopic theory may be expressed as local functions of the hydrodynamic variables, provided that their gradients are small. We may take the hydrodynamic variables to be a local temperature $T$, a local velocity $u^\mu$ satisfying $u^2 = -1$, and when the microscopic theory has a $U(1)$ global symmetry, a local chemical potential $\mu$. Current and energy-momentum conservation are then interpreted as the equations of motion for the hydrodynamic variables. The expressions for the conserved currents in terms of the hydrodynamic variables are referred to as constitutive relations. When working in a gradient expansion, Lorentz invariance strongly constrains the tensor structure of the constitutive relations such that the only undetermined degrees of freedom are scalar functions of $T$ and $\mu$. These scalar functions are usually referred to as transport coefficients.

The transport coefficients of the theory are not only constrained by Lorentz invariance, but also by a local version of the second law of thermodynamics. This second law posits the existence of an entropy current $S^\mu$ which, for an ideal fluid, reduces to the entropy flux current $su^\mu$ (with $s$ the entropy density) and which satisfies $\nabla_\mu S^\mu \geq 0$ under the equations of motion [1]. This local second law is known to force some of the transport coefficients to vanish and constrain others to be non-negative [1–3]. The constitutive relations are also constrained by the Onsager reciprocity relations [4,5]. These relations originate from the invariance of the microscopic theory under CPT and further constrain the transport

coefficients of the fluid.

While the equations of motion so obtained seem to be correct and have successfully described a variety of phenomena, it is somewhat disturbing that in a textbook treatment they are not derived by an action principle which would incorporate all the aforementioned constraints in one sweep. Indeed, given the phenomenological nature of the hydrodynamic equations, this raises the possibility that some constraints have been overlooked and that the theory is incomplete. There has been significant progress this decade in putting the local second law on a more solid footing, using a combination of results from Euclidean thermal field theory [6–8] and unitarity constraints on spectral functions (see e.g. [9]), albeit without an action principle. Even more recent developments allow one to construct effective actions for hydrodynamics in the Schwinger-Keldysh formalism [10–22], at least in certain limiting regimes. The actions so obtained are more intricate than those in ordinary effective field theory, but they have the virtue that various microscopic considerations, such as unitarity and CPT, can be made manifest.

The main goal of this work is to study the effect of these microscopic considerations on the Schwinger-Keldysh effective actions for hydrodynamics, and in turn the constraints on the hydrodynamic equations of motion that follow. Our findings are surprising. We show that the restrictions imposed on the equations of motion from the Onsager relations and positivity of the divergence of the entropy current are necessary but not sufficient to account for all the constraints on the transport coefficients of the fluid. In addition to the Onsager relations and entropy production one must impose an additional constraint which we refer to as the "Schwinger-Keldysh positivity constraint" which is a byproduct of unitarity of the underlying microscopic theory.

Throughout we work in a "statistical mechanical limit" (see [14]) in which one systematically accounts for thermal fluctuations, but neglects quantum fluctuations. In Section 2 we will review the definition and construction of the Schwinger-Keldysh effective action and discuss the statistical mechanical limit in some detail. The formalism discussed in this Section is slightly different from that in [19] but, as we show in Appendix C, the actions so constructed are identical. A summary describing the essential features of the Schwinger-Keldysh effective action in the statistical mechanical limit can be found in Section 3.

Having gained familiarity with the Schwinger-Keldysh effective actions for fluids, we show, in Section 4, how they can be used to construct the constitutive relations of an ideal fluid. This analysis has already been carried out in [14, 15, 18] but we have included it to familiarize the reader with the notation and formalism of the current work.

After this simple example we turn our attention to the local second law. In a companion paper [23] we showed how (in a probe limit) the entropy current can be coupled to an external source and that its divergence is non-negative owing to microscopic unitarity and the KMS condition (see also [24]). We adapt that construction to the statistical mechanical limit in Section 5. Our analysis complements that of [25] in that it couples the entropy current to an external source. This simplifies the computation of the entropy current, its correlation functions, and the entropy production.

Finally, in Section 6 we discuss the constitutive relations of the hydrodynamic theory which follow from our formalism. In Section 6.1 we carry out a detailed analysis of the behavior of the transport coefficients of the theory under CPT. The resulting analysis also allows us to study the emergence of the Onsager reciprocity relations. We then proceed in 6.2 to study the explicit form of the constitutive relations of the underlying theory and match them to the existing literature [26, 27]. Barring 't Hooft anomalies, the allowed terms in the classification of [26, 27] seem to be related to the ones we find. A preliminary analysis of anomalies in the context of the Schwinger-Keldysh effective action

has been carried out in [28]. In the Appendix F we present the effective action for any 't Hooft anomaly described by an anomaly polynomial. We end Section 6 by identifying those constraints coming from the Schwinger-Keldysh positivity condition which are not captured by the entropy current analysis or the Onsager relations.

In Section 7 we carry out explicit computations of the constitutive relations of various types of fluids from an action. We compute the constitutive relations of parity violating fluids in $2 + 1$ dimensions to first order in derivatives, and the same for parity-preserving uncharged fluids in $d + 1$ dimensions. Our results nicely match [9] and [29]. We urge the reader who is unfamiliar with the recent formulations of the Schwinger-Keldysh effective theory and who is interested in a hands-on computation to go through this Section in detail.

For the reader interested in a summary of our main results without delving in the details of our analysis we recommend skipping to Section 8 where we present our classification scheme, especially Table 2. There we compare our findings with the literature [26, 27] and provide a few simple examples. We end this Section with a discussion.

*Note:* While this manuscript was nearing completion two related works [30, 31] were posted to the arXiv.

## 2 The Schwinger-Keldysh effective action

The Schwinger-Keldysh partition function $Z[A_1, A_2]$ associated with an initial state density matrix $\rho_{-\infty}$ is given by

$$Z[A_1, A_2] = \text{Tr}\left(U[A_1]\rho_{-\infty}U^\dagger[A_2]\right), \tag{1}$$

where $A_1$ and $A_2$ collectively denote doubled sources, and $U[A]$ is the time evolution operator from the infinite past to the infinite future in the presence of the sources $A$. Define the generating functional of connected correlation functions, $W = -i \ln Z$. Varying $W$ with respect to the doubled sources gives correlation functions of the conjugate operators in the state $\rho_{-\infty}$ with various time orderings. Letting $O$ denote the operator conjugate to $A$, we have

$$\frac{\delta^{n+m}}{\delta A_1(t_1)\dots\delta A_1(t_n)\delta A_2(\tau_1)\dots\delta A_2(\tau_m)}W\bigg|_{A=0}$$
$$= \text{Tr}\left(\mathcal{T}(O(t_1)\dots O(t_n))\overline{\mathcal{T}}(O(\tau_1)\dots O(\tau_m))\right), \tag{2}$$

where $\mathcal{T}$ is the time-ordering operator, $\overline{\mathcal{T}}$ is the anti-time-ordering operator, and we have specified only the time dependence of the fields. Often, it is convenient to use linear combinations of $A_1$ and $A_2$ to obtain physical observables. For instance, the one point function of $O$ is given by

$$\text{Tr}(\rho_{-\infty}O(t)) = \frac{1}{2}\frac{\delta W}{\delta A_1(t)} - \frac{1}{2}\frac{\delta W}{\delta A_2(t)}\bigg|_{A_1=A_2=0}. \tag{3}$$

We refer the reader to, e.g., [14] for a modern summary and discussion.

In this work we will be interested in the low-energy Schwinger-Keldysh effective action of many systems in a thermal initial state. More formally, we would like to find an effective action $S_{eff}(\xi; A_1, A_2)$ such that at low energies the Schwinger-Keldysh partition function is given by

$$Z[A_1, A_2] = \int D\xi\, e^{\frac{i}{\hbar}S_{eff}}, \tag{4}$$

for low-energy degrees of freedom $\xi$. The "slow modes" of most systems at finite temperature are the conserved currents, and with this in mind we write actions such that the Euler-Lagrange equations for the $\xi$ are simply current and energy-momentum conservation. These actions will turn out to be effective actions for dissipative hydrodynamics.

In the remainder of this Section we will describe the construction of these effective actions in the statistical mechanical limit. Our discussion closely follows the analysis in [18] (see also [14, 15]). Our end result for the effective action is identical to that in [19] though we group our dynamical fields in a slightly different way. We present an analysis comparing the results of this Section with that of [19] in Appendix C.

To find the low-energy Wilsonian effective action we follow the usual logic of identifying low-energy degrees of freedom and symmetries, and construct the most general action compatible with these symmetries. As discussed in [13–15] (see also [18]), the relevant symmetries are as follows:

1. Doubled diffeomorphism invariance whereby $Z[A_1, A_2]$ is invariant under independent diffeomorphisms that act on the sources. When the microscopic theory has a flavor symmetry $G$, one also demands that $Z$ is invariant under doubled flavor transformations.

2. A topological Schwinger-Keldysh symmetry, which states that when the sources are aligned (that is, equal to one another $A_1 = A_2 = A$) the partition function is trivial,

$$Z[A, A] = 1. \tag{5}$$

3. The generating functional need not be real. It satisfies a reality condition

$$W[A_1, A_2]^* = -W[A_2^*, A_1^*]. \tag{6}$$

4. A KMS symmetry of the partition function, which, following [14], can be written as

$$Z[A_1(t_1), A_2(t_2)] = Z[\eta_A A_1(-t_1), \eta_A A_2(-t_2 - ib)]. \tag{7}$$

Here the initial state is $\rho_{-\infty} \propto e^{-b\mathcal{H}}$ with $\mathcal{H}$ the generator of time translations, and $\eta_A$ is the CPT eigenvalue of the operator conjugate to $A$. The KMS symmetry can also be written covariantly. We will discuss it shortly in some detail.

In addition to these symmetries, unitarity imposes an additional constraint on the imaginary part of the Schwinger-Keldysh partition function [14, 25],

$$|Z| \leq 1, \tag{8a}$$

or, equivalently,

$$\mathrm{Im}(W) \geq 0, \tag{8b}$$

which we reproduce in Appendix A and refer to as the "Schwinger-Keldysh positivity condition". The inequality $|Z| \leq 1$ plays a crucial role in deriving the local version of the second law as we discuss in Section 5 and in providing further constraints on transport coefficients which we discuss in Section 6. The KMS symmetry is one of the ingredients which ensures the Onsager relations which we also discuss in Section 6.

## 2.1 Degrees of freedom and doubled symmetries

We wish to ensure that the equations of motion of our effective theory are the (doubled) conservation equations for the energy-momentum tensor. To do so we take the degrees of freedom to be maps $X_1^\mu(\sigma)$ and $X_2^\mu(\sigma)$ between what we refer to as a worldvolume with coordinates $\sigma$ and two target, or physical, spaces. The sources are defined in these target spaces, and are given by $A_1(x_1)$ and $A_2(x_2)$. When the microscopic theory has a continuous global symmetry $G$, there are additional $G$-valued fields $C_1(\sigma)$ and $C_2(\sigma)$ which ensure current conservation. In what follows, we will take $G = U(1)$ in order to simplify the presentation.

In order for the action to be invariant under the doubled diffeomorphisms and flavor transformations, we demand that the $X$'s and $C$'s always appear in combination with the target space sources via pullbacks:

$$
\begin{aligned}
B_{s\,i}(X_s(\sigma), C_s(\sigma)) &= B_{s\,\mu}(X_s(\sigma))\partial_i X_s^\mu(\sigma) + \partial_i C_s(\sigma)\,, \\
g_{s\,ij}(X_s(\sigma)) &= g_{s\,\mu\nu}(X_s(\sigma))\partial_i X_s^\mu(\sigma)\partial_j X_s^\nu(\sigma)\,,
\end{aligned}
\tag{9}
$$

where $s = 1, 2$ specifies the target spaces. With the $X_s$'s transforming as coordinates under target space diffeomorphisms and the $C_s$'s transforming as phases under $U(1)$ transformations, the $g_{s\,ij}$'s and $B_{s\,i}$'s are invariant under target space diffeomorphisms and $U(1)$ transformations. Note, however, that the $B_{s\,i}$'s and $g_{s\,ij}$'s transform as one-forms and symmetric tensors respectively under worldvolume diffeomorphisms. Likewise the $B_{s\,i}$'s (through their dependence on the $C$'s) transform as $U(1)$ connections under a worldvolume gauge transformation: $B_{s\,i} \to B_{s\,i} + \partial_i \Lambda$.

In addition to the dynamical degrees of freedom, in order to account for the initial thermal state, we will introduce a thermal vector $\beta^i$ and a flavor transformation parameter $\Lambda_\beta$. Together they generate a worldvolume time transformation $\delta_\beta$, which we take to be such that, in the far past, it is the same transformation generated by the grand potential appearing in the initial state $\exp(-b\mathcal{H})$. We will insist that the effective action is invariant under worldvolume diffeomorphism and flavor transformations, under which the thermal data $\beta^i$ and $\Lambda_\beta$ suitably transform.

## 2.2 The statistical mechanical limit

So far we have described the degrees of freedom and how we impose doubled diffeomorphism and (a possible) doubled flavor invariance. As we mentioned in the previous Subsection, we have ensured that our action is double-diffeomorphism invariant by combining the sources $g_{s\,\mu\nu}(x_s)$ together with the $X_s^\mu$'s into pullback fields $g_{s\,ij}(\sigma)$. When the microscopic theory has a $U(1)$ flavor symmetry, we have also grouped the external $U(1)$ fields $B_{s\,\mu}(x_s)$ together with the $C_s$'s into pullback fields $B_{s\,i}(\sigma)$.

It is challenging to implement the remaining topological Schwinger-Keldysh symmetry and the $\mathbb{Z}_2$ KMS symmetry. In [18] three of us have discussed how to implement these symmetries in a probe limit, where charge is transported in a fixed thermal background. The virtue of the probe limit is that it allows one to consider both classical and quantum fluctuations. This stands in contrast to a statistical mechanical limit introduced in [14,19,21], or to the (seemingly equivalent) high temperature limit of [13], where the entire system is dynamical but quantum fluctuations are treated perturbatively. One virtue of the statistical mechanical limit is that the KMS symmetry (7) becomes local. In this Subsection, we will rederive the statistical mechanical limit, working in a formalism closely related to that of [18]. In Sections 2.3 and 2.5 we will see how this will help us implement the Schwinger-Keldysh and KMS symmetries. As mentioned earlier our end result matches that of [14, 19, 21] as we elaborate on in Appendix C. A full implementation of

doubled diffeomorphism invariance, Schwinger-Keldysh symmetry, reality condition, and KMS symmetry at the quantum level is currently unavailable.

Before delving into the statistical mechanical limit, it is helpful to change basis from the 1 and 2 fields and define so-called average ($r$) and difference ($a$) operators and sources, given schematically by

$$O_r(t) = \frac{1}{2}(O_1(t) + O_2(t)), \qquad O_a(t) = O_1(t) - O_2(t). \tag{10}$$

In the $r/a$ basis, the variation of the generating functional is

$$\delta W = \int d^d x \left( O_1 \delta A_1 - O_2 \delta A_2 \right) = \int d^d x \left( O_r \delta A_a + O_a \delta A_r \right), \tag{11}$$

so that $r$-sources are conjugate to $a$-operators and $a$-sources to $r$ operators.[1] In terms of these, the Schwinger-Keldysh symmetry (5) is the statement that $Z = 1$ when the $a$-sources vanish. Equivalently, it is the statement that correlation functions of the $a$-type operators identically vanish among themselves,

$$\langle O_a(t_1) \dots O_a(t_n) \rangle = 0. \tag{12}$$

In the statistical mechanical limit we restore $\hbar$ as a formal expansion parameter and take a suitable $\hbar \to 0$ limit. In taking this limit there are two observations to keep in mind which will guide the analysis to follow. The first is that, after restoring $\hbar$, the thermal density matrix $e^{-bH}$ is an evolution operator by an imaginary time $-\hbar b$. Correspondingly, the KMS symmetry (7) is non-local, relating the partition function with source $A_2(t)$ to one with source $A_2(-t - i\hbar b)$. As we will see shortly, once we take $\hbar$ to be small, the KMS symmetry will become local. The second, more relevant for us here, is that we restrict our attention to configurations where the $a$-type fields, external and quantum, are $O(\hbar)$. This is reminiscent of the non-relativistic limit of certain relativistic field theories, whereby one restores $c$ and takes a suitable $c \to \infty$ limit (see e.g. [32, 33]).

At the level of the effective action, taking the $\hbar \to 0$ limit amounts to the following. Starting with an action $S_{eff}$ of the $r$- and $a$-fields, we rescale the $a$-fields by a power of $\hbar$ so that the $r$- and $a$-fields are both $O(\hbar^0)$ as $\hbar \to 0$. We then expand the effective action in powers of the $a$-field, which we schematically represent as

$$\frac{1}{\hbar} S_{eff}[\phi_r, \phi_a; \hbar] \to \frac{1}{\hbar} S_{eff}[\phi_r, \hbar \phi_a; \hbar] = \sum_{n=1} \hbar^{n-1} S_n[\phi_r; \hbar] \phi_a^n, \tag{13}$$

where the sum on the far right starts at $n = 1$ due to the Schwinger-Keldysh symmetry. We posit that that the $\hbar \to 0$ limit is regular. That is, we assume that $\hbar^{n-1} S_n[\phi_r; \hbar]$ has a finite $\hbar \to 0$ limit,

$$\lim_{\hbar \to 0} \hbar^{n-1} S_n[\phi_r; \hbar] = \mathcal{S}_n[\phi_r]. \tag{14}$$

The statistical mechanical limit of the effective action is then

$$S_{SM}[\phi_r, \phi_a] = \sum_{n=1} \mathcal{S}_n[\phi_r] \phi_a^n. \tag{15}$$

Let us now carefully implement the statistical mechanical limit in the effective theory for fluids. We restrict our attention to sources which, in some choice of target space coordinates and $U(1)$ gauges, are nearly aligned, i.e.

$$g_{1\,\mu\nu}(x) = g_{2\,\mu\nu}(x) + O(\hbar), \qquad B_{1\,\mu}(x) = B_{2\,\mu}(x) + O(\hbar). \tag{16}$$

---

[1] We have intentionally omitted the measure from the schematic expression in (11). We will deal with it in detail later in this Section.

Further, we only consider nearly-aligned configurations of the dynamical fields,

$$X_1^\mu(\sigma) = X_r^\mu(\sigma) + \frac{\hbar}{2} X_a^\mu(\sigma) + O(\hbar^2), \qquad X_2^\mu(\sigma) = X_r^\mu(\sigma) - \frac{\hbar}{2} X_a^\mu(\sigma) + O(\hbar^2),$$
$$C_1(\sigma) = C_r(\sigma) + \frac{\hbar}{2} C_a(\sigma) + O(\hbar^2), \qquad C_2(\sigma) = C_r(\sigma) - \frac{\hbar}{2} C_a(\sigma) + O(\hbar^2). \tag{17}$$

These equations effectively define $r$- and $a$-type combinations of the dynamical fields. Note that we have rescaled the $a$-type combinations so that they are finite in the $\hbar \to 0$ limit. With this choice the pullback fields are nearly aligned as well,

$$g_{1\,ij}(\sigma) = g_{2\,ij}(\sigma) + O(\hbar), \qquad B_{1\,i}(\sigma) = B_{2\,i}(\sigma) + O(\hbar). \tag{18}$$

The full doubled diffeomorphism and flavor invariance is not manifest in the statistical mechanical limit. A general diffeomorphism and flavor transformation will lead to metrics and flavor fields which are no longer aligned to $O(\hbar)$. For this reason we only demand invariance under diffeomorphisms and flavor transformations which maintain the near-alignment $X_1^\mu = X_2^\mu + O(\hbar)$ and $C_1 = C_2 + O(\hbar)$. More precisely, we allow infinitesimal diffeomorphisms $\xi_1^\mu(x_1)$ and $\xi_2^\nu(x_2)$ and $U(1)$ transformations $\Lambda_1(x_1)$ and $\Lambda_2(x_2)$ which are nearly aligned, satisfying

$$\xi_1^\mu(x) = \xi_2^\mu(x) + O(\hbar), \qquad \Lambda_1(x) = \Lambda_2(x) + O(\hbar). \tag{19}$$

Under a general diffeomorphism or flavor transformation, the dynamical fields shift as

$$\delta_\chi X_1^\mu(\sigma) = -\xi_1^\mu(X_1(\sigma)), \qquad \delta_\chi C_1(\sigma) = -\Lambda_1(X_1(\sigma)), \tag{20}$$

while the sources vary as

$$\delta_\chi g_{1\,\mu\nu} = \mathcal{L}_{\xi_1} g_{1\,\mu\nu}, \qquad \delta_\chi B_{1\,\mu} = \mathcal{L}_{\xi_1} B_{1\,\mu} + \partial_\mu \Lambda_1, \tag{21}$$

and similarly for the 2 fields. In the $\hbar \to 0$ limit we define $r$- and $a$-type combinations of the $\xi^\mu$ and $\Lambda$ to be their $O(\hbar^0)$ and $O(\hbar)$ terms

$$\xi_1^\mu(x) = \xi_r^\mu(x) + \frac{\hbar}{2} \xi_a^\mu(x) + O(\hbar^2), \qquad \xi_2^\mu(x) = \xi_r^\mu(x) - \frac{\hbar}{2} \xi_a^\mu(x) + O(\hbar^2),$$
$$\Lambda_1(x) = \Lambda_r(x) + \frac{\hbar}{2} \Lambda_a(x) + O(\hbar^2), \qquad \Lambda_2(x) = \Lambda_r(x) - \frac{\hbar}{2} \Lambda_a(x) + O(\hbar^2). \tag{22}$$

Written this way, it is clear that these transformations are the combination of a "diagonal" transformation $(\xi_r^\mu, \Lambda_r)$ as well as a linearized "axial" transformation $(\xi_a^\mu, \Lambda_a)$. According to (17) the $r$-type combinations of the dynamical fields then vary as

$$\delta_\chi X_r^\mu(\sigma) = -\xi_r^\mu(X_r(\sigma)),$$
$$\delta_\chi C_r(\sigma) = -\Lambda_r(X_r(\sigma)), \tag{23}$$

where we remind the reader that we are in the $\hbar \to 0$ limit.

One may also be tempted to deduce that $\delta_\chi X_a^\mu(\sigma) = -X_a^\nu(\sigma) \partial_\nu \xi_r^\mu(X_r(\sigma)) - \xi_a^\mu(X_r(\sigma))$ or $\delta_\chi C_a(\sigma) = -X_a^\mu(\sigma) \partial_\mu \Lambda_r(X_r(\sigma)) - \Lambda_a(X_r(\sigma))$. However, as we will see in the next Subsection (in Equation (26)), transformations of the $a$-type fields must be modified by ghost terms so as to be consistent with the Schwinger-Keldysh symmetry.

### 2.3 Schwinger-Keldysh symmetry and superspace

Recall that the Schwinger-Keldysh symmetry (5) is the statement that $Z = 1$ when the $a$-sources vanish or, equivalently, that

$$\langle O_a(t_1) \dots O_a(t_n) \rangle = 0 \tag{24}$$

in the absence of sources. That is, the correlation functions of the $a$-operators are topological, in that they do not depend on the locations at which the $O_a$ are inserted. This feature is reminiscent of Witten-type topological field theories in which the correlation functions of the stress tensor are topological. Adapting the cohomological construction of Witten-type theories [34, 35], the Schwinger-Keldysh symmetry can be implemented in the effective theory as follows. We posit the existence of a scalar Grassmann-odd operator $Q$ with $Q^2 = 0$, ensure that the action is $Q$-closed when the $a$-type sources vanish, and require the $a$-type operators to be $Q$-exact.

For each bosonic field in the theory we introduce a Grassman-odd ghost partner with suitable transformation laws under $Q$ so that $Q$ is a symmetry when the sources are aligned. We include ghost partners $X_g^\mu$ and $X_{\bar{g}}^\mu$ to $X_r^\mu$ and $X_a^\mu$, as well as partners $C_g$ and $C_{\bar{g}}$ to $C_r$ and $C_a$. We then define a cohomological supercharge $Q$ to enforce the Schwinger-Keldysh symmetry. It acts on the dynamical fields as

$$\begin{aligned}
[Q, X_r^\mu] &= X_{\bar{g}}^\mu, & \{Q, X_{\bar{g}}^\mu\} &= [Q, X_a^\mu] = 0, & \{Q, X_g^\mu\} &= X_a^\mu, \\
[Q, C_r] &= C_{\bar{g}}, & \{Q, C_{\bar{g}}\} &= [Q, C_a] = 0, & \{Q, C_g\} &= C_a,
\end{aligned} \tag{25}$$

and therefore obeys $Q^2 = 0$. We assume that the thermal data $\beta^i$ and $\Lambda_\beta$ are inert under $Q$. In what follows, we refer to the transformation generated by $Q$ as $\delta_Q$, so that, e.g., $\delta_Q X_r^\mu = X_{\bar{g}}^\mu$.

Having introduced ghosts and a supercharge $Q$, we will impose an additive ghost number symmetry on our effective action. We assign $(Q, X_{\bar{g}}^\mu, C_{\bar{g}})$ ghost number $+1$ and $(X_g^\mu, C_g)$ ghost number $-1$. We will demand that our effective action has ghost number 0.

Let us denote transformations which involve a diffeomorphism associated with $\xi_a$ and a gauge transformation associated with $\Lambda_a$ by $\delta_a$ and transformations associated the $r$-type fields by $\delta_r$ so that $\delta_\chi = \delta_r + \delta_a$. Requiring (23), $[\delta_Q, \delta_r] = 0$ and that, in the absence of ghosts, $X_a^\mu$ transforms as a vector under $\delta_r$ strongly constrains the transformation laws of the ghosts and the $a$ fields in the presence of ghosts under $\delta_r$. We find

$$\begin{aligned}
\delta_r X_{\bar{g}}^\mu &= -X_{\bar{g}}^\nu \partial_\nu \xi_r^\mu(X_r(\sigma)), & \delta_r X_g^\mu &= -X_g^\nu \partial_\nu \xi_r^\mu(X_r(\sigma)), \\
\delta_r C_{\bar{g}} &= -X_{\bar{g}}^\mu \partial_\mu \Lambda_r(X_r(\sigma)), & \delta_r C_g &= -X_g^\mu \partial_\mu \Lambda_r(X_r(\sigma)),
\end{aligned} \tag{26a}$$

and that the transformations of the bosonic fields in the presence of ghosts are

$$\begin{aligned}
\delta_r X_r^\mu &= -\xi_r^\mu(X_r(\sigma)), & \delta_r X_a^\mu &= -X_a^\nu \partial_\nu \xi_r^\mu(X_r(\sigma)) - X_{\bar{g}}^\nu X_g^\rho \partial_\nu \partial_\rho \xi_r^\mu(X_r(\sigma)), \\
\delta_r C_r &= -\Lambda_r(X_r(\sigma)), & \delta_r C_a &= -X_a^\mu \partial_\mu \Lambda_r(X_r(\sigma)) - X_{\bar{g}}^\mu X_g^\nu \partial_\mu \partial_\nu \Lambda_r(X_r(\sigma)).
\end{aligned} \tag{26b}$$

We may consistently choose for all but the $a$-fields to be inert under $a$-transformations, and that the variation of the $a$-fields is given by

$$\delta_a X_a^\mu = -\xi_a^\mu(X_r(\sigma)), \qquad \delta_a C_a = -\Lambda_a(X_r(\sigma)). \tag{27}$$

We refer the reader to Appendix B for details. Observe that if we repackage the $X$-ghosts as worldvolume vectors,

$$\bar{\psi}^i = X_{\bar{g}}^\mu (\partial_i X_r^\mu)^{-1}, \qquad \psi^i = X_g^\mu (\partial_i X_r^\mu)^{-1}, \tag{28}$$

then $\bar{\psi}^i$ and $\psi^j$ are invariant under target space diffeomorphisms. Later we will also find it useful to introduce a worldvolume companion for $X_a^\mu$,

$$\rho_a^i = (\partial_i X_r^\mu)^{-1} X_a^\mu\,. \tag{29}$$

The action of $Q$ (25) and the $r/a$-transformations (26), (27) on the dynamical fields can be efficiently represented using superspace. We introduce two Grassmann-odd coordinates $\theta$ and $\bar{\theta}$, of ghost number $-1$ and $+1$ respectively, and group the supermultiplets $(X_r^\mu, X_{\bar{g}}^\mu, X_g^\mu, X_a^\mu)$ and $(C_r, C_{\bar{g}}, C_g, C_a)$ into superfields

$$\begin{aligned}
\mathbb{X}^\mu &= X_r^\mu + \theta X_{\bar{g}}^\mu + \bar{\theta} X_g^\mu + \bar{\theta}\theta X_a^\mu\,, \\
\mathbb{C} &= C_r + \theta C_{\bar{g}} + \bar{\theta} C_g + \bar{\theta}\theta C_a\,,
\end{aligned} \tag{30}$$

on which $Q$ can be shown to act via the superdifferential operator $\frac{\partial}{\partial\theta}$, i.e.

$$[Q, \mathbb{X}^\mu] = \frac{\partial \mathbb{X}^\mu}{\partial\theta}\,, \qquad [Q, \mathbb{C}] = \frac{\partial \mathbb{C}}{\partial\theta}\,. \tag{31}$$

Note that $\mathbb{X}^\mu$ and $\mathbb{C}$ have ghost number $0$.[2] In terms of superfields, the action of the $r/a$-transformations (26) and (27) can be written as

$$\begin{aligned}
\delta_\chi \mathbb{X}^\mu &= -\mathbb{\xi}^\mu = -\left(\xi_r^\mu(\mathbb{X}) + \bar{\theta}\theta \xi_a^\mu(\mathbb{X})\right), \\
\delta_\chi \mathbb{C} &= -\mathbb{\Lambda} = -\left(\Lambda_r(\mathbb{X}) + \bar{\theta}\theta \Lambda_a(\mathbb{X})\right).
\end{aligned} \tag{32}$$

Recall that we obtained the $r$-transformation laws of the ghosts by demanding that $[Q, \delta_r] = 0$. The vanishing of this commutator is manifest here: when the $a$-transformations vanish, the variations of $\mathbb{X}^\mu$ and $\mathbb{C}$ are functions of superfields, and so $Q$ acts on the superfields $\mathbb{X}^\mu$ and $\mathbb{C}$ in the same way as on their $r$-variations.

In (17) we defined $r$ and $a$-type combinations of the dynamical fields. Following standard methods for symmetry breaking in quantum field theory, we would like to construct $r$- and $a$-type combinations of the pulled back sources so that the $a$-type pulled back sources vanish when the sources are aligned. A naive choice would be $\frac{1}{2}(g_{1\,ij}(\sigma) + g_{2\,ij}(\sigma))$ for the $r$-type combination and $g_{1\,ij}(\sigma) - g_{2\,ij}(\sigma)$ for the $a$-type pullback. The virtue of this choice is that both the $r$- and $a$-type fields would then be invariant under independent target space diffeomorphisms. However, with this definition it is challenging to enforce the Schwinger-Keldysh symmetry using cohomological techniques. The obstruction is as follows: microscopically, the statement that the sources are aligned is that there exists some choice of target space coordinates such that $g_{1\,\mu\nu}(x) - g_{2\,\mu\nu}(x) = 0$ everywhere (and a similar equation for the other sources). This microscopic statement is not equivalent to saying that the naive $a$-type pullback $g_{1\,ij}(\sigma) - g_{2\,ij}(\sigma)$ vanishes. It is instead equivalent to saying that there is a particular field configuration $X_1^\mu = \overline{X}_1^\mu(\sigma)$ and $X_2^\mu = \overline{X}_2^\mu(\sigma)$ for which this naive $a$-pullback vanishes, $\bar{g}_{1\,ij}(\sigma) - \bar{g}_{2\,ij}(\sigma) = 0$. But, for a different field configuration, e.g. $X_1^\mu = \overline{X}_1^\mu$ and $X_2^\mu = \overline{X}_2^\mu + \delta\overline{X}_2^\mu$, the pullback metrics will generally differ and the candidate $a$-metric is nonzero. So there seems to be a conflict between the

---

[2]Note that, in principle, we could have implemented the Schwinger-Keldysh symmetry by a single superspace coordinate and two superfields, say, $\mathbb{X}_r^\mu$ and $\mathbb{X}_a^\mu$, the first with vanishing ghost number and the second with a non-vanishing one. Instead, we have used two superspace coordinates, $\theta$ and $\bar{\theta}$ to group these into a single superfield with the understanding that the Lagrangian may depend explicitly on $\bar{\theta}$. We will see later that under the KMS symmetry we will be forced to remove any explicit $\bar{\theta}$ dependence from the Lagrangian.

doubled diffeomorphism invariance, having $X^\mu$'s as the low-energy degrees of freedom, and using cohomology to enforce the Schwinger-Keldysh symmetry.[3]

In [18] this conflict was evaded by appealing to a probe limit where the $X^\mu$'s are, for all intents and purposes, inert. In the statistical mechanical limit this conflict is evaded since doubled diffeomorphism invariance is effectively broken down to the diagonal subgroup that acts simultaneously on the 1 and 2 fields, while the "axial" subgroup survives as a linearized invariance.[4]

In equations, we define $r$- and $a$-metrics in the $\hbar \to 0$ limit via

$$
\begin{aligned}
g_{r\,ij}(X_r(\sigma)) &= \lim_{\hbar \to 0} \frac{1}{2}\Big(g_{1\,\mu\nu}(X_r(\sigma)) + g_{2\,\mu\nu}(X_r(\sigma))\Big)\partial_i X_r^\mu \partial_j X_r^\nu \\
&= \lim_{\hbar \to 0} \frac{1}{2}\left(g_{1\,ij}(\sigma) + g_{2\,ij}(\sigma)\right), \\
g_{a\,ij}(X_r(\sigma)) &= \lim_{\hbar \to 0} \frac{\Big(g_{1\,\mu\nu}(X_r(\sigma)) - g_{2\,\mu\nu}(X_r(\sigma))\Big)\partial_i X_r^\mu \partial_j X_r^\nu}{\hbar},
\end{aligned}
\tag{33}
$$

and we remind the reader of the expansion (16), (17) and (18). Observe that, if the metrics are aligned, $g_{1\,\mu\nu}(x) = g_{2\,\mu\nu}(x)$, then this $a$-combination vanishes for all field configurations. So we can consistently demand that our effective action is $Q$-closed when the $a$-combinations vanish, and therefore use cohomology to enforce the Schwinger-Keldysh symmetry. Both $g_{r\,ij}$ and $g_{a\,ij}$ are tensors under worldvolume diffeomorphisms. We similarly define the $r$- and $a$-flavor fields to be

$$
\begin{aligned}
B_{r\,i}(X_r(\sigma), C_r(\sigma)) &= \lim_{\hbar \to 0}\left[\frac{1}{2}\Big(B_{1\,\mu}(X_r(\sigma)) + B_{2\,\mu}(X_r(\sigma))\Big)\partial_i X_r^\mu + \partial_i C_r\right] \\
&= \lim_{\hbar \to 0} \frac{1}{2}\left(B_{1\,i}(\sigma) + B_{2\,i}(\sigma)\right), \\
B_{a\,i}(X_r(\sigma)) &= \lim_{\hbar \to 0} \frac{\Big(B_{1\,\mu}(X_r(\sigma)) - B_{2\,\mu}(X_r(\sigma))\Big)\partial_i X_r^\mu}{\hbar}.
\end{aligned}
\tag{34}
$$

They are one-forms under worldvolume diffeomorphisms while under worldvolume $U(1)$ transformations $B_{r\,i}$ transforms as a connection and $B_{a\,i}$ is invariant.

The various fields in (33) and (34) are obviously not tensors under general target space diffeomorphisms and $U(1)$ transformations. However, we do not consider general transformations in the statistical mechanical limit, but only nearly-aligned transformations (22). Under them, the $r$-pullbacks are invariant, which follows from the fact that they are the $\hbar \to 0$ limit of invariant pullbacks.

In contrast to the $r$-pullbacks, the $a$-pullbacks are not invariant under target diffeomorphisms. They transform as

$$
\begin{aligned}
\delta_\chi g_{a\,ij}(\sigma) &= \mathcal{L}_{\xi_a} g_{r\,ij}(\sigma), \\
\delta_\chi B_{a\,i}(\sigma) &= \mathcal{L}_{\xi_a}(B_{r\,i}(\sigma) - \partial_i C_r) + \partial_i \Lambda_a(X_r(\sigma)),
\end{aligned}
\tag{35}
$$

---

[3] The authors of [14] have proposed a method for defining a cohomological supercharge $Q$ which becomes a symmetry whenever the sources are aligned regardless of the configuration of the $X$'s. In the current language it involves adding to the difference fields $g_{1\,ij}(\sigma) - g_{2\,ij}(\sigma)$ a $Q$-exact term which compensates for the mismatch associated with different field configurations. At this point, it is unclear if that proposal is capable of satisfying the doubled diffeomorphism invariance. Regardless, the authors of [14] eventually resorted to the statistical mechanical approximation described below in order to resolve yet another problem once the KMS symmetry was to be implemented.

[4] It may be helpful to think about this in analogy with the non-relativistic limit of relativistic field theories. In that limit, one typically takes a Lorentz-invariant massive field theory with a $U(1)$ global symmetry, tunes the chemical potential to threshold, $\mu = mc^2$, and then sends $c \to \infty$ while zooming in on field configurations with finite energies and momenta [32,33]. After taking that limit the full Poincaré symmetry is no longer manifest, and it is effectively contracted to Galilean symmetry.

where the Lie derivatives are taken along the worldvolume vector

$$\xi_a^i(\sigma) = \xi_a^\mu(X_r(\sigma))(\partial_i X_r^\mu)^{-1}\,. \tag{36}$$

We would like to find diffeomorphism and flavor-invariant completions of $g_{a\,ij}$ and $B_{a\,i}$. Given the transformation laws of the $X$-supermultiplet and $C$-supermultiplet (26) and (27), we find the following combinations are invariant under target space transformations:

$$\delta_\chi \left( g_{a\,ij} + \mathcal{L}_{\rho_a} g_{r\,ij} + \frac{1}{2}\left([\mathcal{L}_{\bar\psi}, \mathcal{L}_\psi] - \mathcal{L}_{[\bar\psi,\psi]}\right) g_{r\,ij}\right) = 0\,,$$

$$\delta_\chi \left( B_{a\,i} + \mathcal{L}_{\rho_a}(B_{r\,i} - \partial_i C_r) + \partial_i C_a + \frac{1}{2}\left([\mathcal{L}_{\bar\psi}, \mathcal{L}_\psi] - \mathcal{L}_{[\bar\psi,\psi]}\right)(B_{r\,i} - \partial_i C_r)\right) = 0\,, \tag{37}$$

where the Lie derivatives are taken along $\bar\psi^i$, $\psi^j$, and $\rho_a^k$ defined in (28) and (29), and

$$[\bar\psi, \psi]^i = \bar\psi^j\partial_j\psi^i - \psi^j\partial_j\bar\psi^i\,. \tag{38}$$

Next we would like to package the $r$- and $a$-metric into a superfield on which $Q$ acts simply, i.e. a super-pullback metric. We define

$$\mathfrak{g}_{ij} = g_{r\,ij}(\mathbb{X}) + \bar\theta\theta\, g_{a\,ij}(\mathbb{X}) \tag{39}$$

$$= g_{r\,ij} + \theta\mathcal{L}_{\bar\psi}g_{r\,ij} + \bar\theta\mathcal{L}_\psi g_{r\,ij} + \bar\theta\theta\left(g_{a\,ij} + \mathcal{L}_{\rho_a}g_{r\,ij} + \frac{1}{2}\left([\mathcal{L}_{\bar\psi}, \mathcal{L}_\psi] - \mathcal{L}_{[\bar\psi,\psi]}\right)g_{r\,ij}\right)\,,$$

where

$$g_{r\,ij}(\mathbb{X}) = \lim_{\hbar\to 0}\frac{1}{2}(g_{1\,\mu\nu}(\mathbb{X}) + g_{2\,\mu\nu}(\mathbb{X}))\partial_i\mathbb{X}^\mu\partial_j\mathbb{X}^\nu\,,$$

$$g_{a\,ij}(\mathbb{X}) = \lim_{\hbar\to 0}\frac{(g_{1\,\mu\nu}(\mathbb{X}) - g_{2\,\mu\nu}(\mathbb{X}))\partial_i\mathbb{X}^\mu\partial_j\mathbb{X}^\nu}{\hbar}\,. \tag{40}$$

The super-pullback $\mathfrak{g}_{ij}$ is invariant under $r$- and $a$-type diffeomorphisms: its bottom and middle components are manifestly invariant, and the top component is the diffeomorphism-invariant completion of $g_{a\,ij}$ given in (37). The invariance is also visible in superspace. The $r/a$-transformations act on the external metrics as

$$\delta_\chi g_{1\,\mu\nu}(\mathbb{X}) = \mathcal{L}_{\xi_r(\mathbb{X}) + \frac{\hbar}{2}\xi_a(\mathbb{X}) + O(\hbar^2)}g_{1\,\mu\nu}(\mathbb{X})\,, \quad \delta_\chi g_{2\,\mu\nu}(\mathbb{X}) = \mathcal{L}_{\xi_r(\mathbb{X}) - \frac{\hbar}{2}\xi_a(\mathbb{X}) + O(\hbar^2)}g_{2\,\mu\nu}(\mathbb{X})\,, \tag{41}$$

and on the dynamical fields as $\delta_\chi\mathbb{X}^\mu = -\xi^\mu$. It follows that

$$\delta_\chi g_{r\,ij}(\mathbb{X}) = \lim_{\hbar\to 0}\Big[\frac{1}{2}\mathcal{L}_{\xi_r(\mathbb{X})}(g_{1\,\mu\nu}(\mathbb{X}) + g_{2\,\mu\nu}(\mathbb{X}))\partial_i\mathbb{X}^\mu\partial_j\mathbb{X}^\nu$$

$$+ \frac{1}{2}\mathcal{L}_{-\xi}(g_{1\,\mu\nu}(\mathbb{X}) + g_{2\,\mu\nu}(\mathbb{X}))\partial_i\mathbb{X}^\mu\partial_j\mathbb{X}^\nu\Big]$$

$$= \lim_{\hbar\to 0}\frac{1}{2}\left[\mathcal{L}_{\xi_r(\mathbb{X})-\xi}(g_{1\,\mu\nu}(\mathbb{X}) + g_{2\,\mu\nu}(\mathbb{X}))\right]\partial_i\mathbb{X}^\mu\partial_j\mathbb{X}^\nu$$

$$= -\bar\theta\theta\mathcal{L}_{\xi_a}g_{r\,ij}(\sigma)\,, \tag{42}$$

where in the last line the Lie derivative is along $\xi_a^i$ as defined in (36). The first line of (42) is the variation of the metrics, and the second comes from $\delta_\chi\mathbb{X}^\mu = -\xi^\mu$. Combined with the non-invariance of $g_{a\,ij}$ (35),

$$\delta_\chi g_{a\,ij}(\sigma) = \mathcal{L}_{\xi_a}g_{r\,ij}(\sigma)\,, \tag{43}$$

it follows that $\mathfrak{g}_{ij} = g_{r\,ij}(\mathbb{X}) + \bar{\theta}\theta g_{a\,ij}(\mathbb{X})$ is invariant under $\delta_\chi$. Furthermore, observe that when the $a$-metric vanishes, $\mathfrak{g}_{ij}$ is a function of the superfield $\mathbb{X}^\mu$, in which case $Q$ acts on $\mathfrak{g}_{ij}$ in the same way as on $\mathbb{X}^\mu$ itself, that is,

$$\left.[Q, \mathfrak{g}_{ij}]\right|_{g_{a\,ij}=0} = \left.\frac{\partial \mathfrak{g}_{ij}}{\partial \theta}\right|_{g_{a\,ij}=0} . \tag{44}$$

By the same sort of logic we write the super-flavor field

$$\begin{aligned}
\mathbb{B}_i &= B_{r\,i}(\mathbb{X}, \mathbb{C}) + \bar{\theta}\theta B_{a\,i}(\mathbb{X}) \\
&= B_{r\,i} + \theta\left(\mathcal{L}_{\bar{\psi}}(B_{r\,i} - \partial_i C_r) + \partial_i C_{\bar{g}}\right) + \bar{\theta}\left(\mathcal{L}_\psi(B_{r\,i} - \partial_i C_r) + \partial_i C_g\right) \\
&\quad + \bar{\theta}\theta\left(B_{a\,i} + \mathcal{L}_{\rho_a}(B_{r\,i} - \partial_i C_r) + \partial_i C_a + \frac{1}{2}\left([\mathcal{L}_{\bar{\psi}}, \mathcal{L}_\psi] - \mathcal{L}_{[\bar{\psi}, \psi]}\right)(B_{r\,i} - \partial_i C_r)\right),
\end{aligned} \tag{45}$$

where

$$\begin{aligned}
B_{r\,i}(\mathbb{X}, \mathbb{C}) &= \lim_{\hbar \to 0}\left[\frac{1}{2}\left(B_{1\,\mu}(\mathbb{X}) + B_{2\,\mu}(\mathbb{X})\right)\partial_i \mathbb{X}^\mu + \partial_i \mathbb{C}\right], \\
B_{a\,i}(\mathbb{X}) &= \lim_{\hbar \to 0}\frac{(B_{1\,\mu}(\mathbb{X}) - B_{2\,\mu}(\mathbb{X}))\partial_i \mathbb{X}^\mu}{\hbar} .
\end{aligned} \tag{46}$$

The super-flavor field is also invariant under the $r$- and $a$-transformations, and varies as a connection under worldvolume $U(1)$ transformations. As before, when the $a$-source vanishes, $Q$ acts on $\mathbb{B}_i$ as $\frac{\partial}{\partial \theta}$,

$$\left.[Q, \mathbb{B}_i]\right|_{B_{a\,i}=0} = \left.\frac{\partial \mathbb{B}_i}{\partial \theta}\right|_{B_{a\,i}=0} . \tag{47}$$

Recall that, to account for the initial thermal state, we introduced the bosonic fields $\beta^i$ and $\Lambda_\beta$. We may regard $\beta^i$, $\Lambda_\beta$, and the transformation they generate, $\delta_\beta$, as superfields with no middle or top component, e.g.,

$$\beta^i = \beta^i . \tag{48}$$

By assumption, $\beta^i$ and $\Lambda_\beta$ are inert under $Q$, and so we may consistently write the (vanishing) action of $Q$ on $\beta^i$ and $\Lambda_\beta$ as $[Q, \beta^i] = \frac{\partial \beta^i}{\partial \theta} = 0$ and $[Q, \Lambda_\beta] = \frac{\partial \Lambda_\beta}{\partial \theta} = 0$, that is, the same action as on $\mathbb{X}^\mu$, $\mathbb{C}$, and on the super-pullbacks (when the $a$-sources vanish).

We can now use the super-pullbacks $\mathfrak{g}_{ij}$ and $\mathbb{B}_i$ together with the thermal data $\beta^i$ and $\Lambda_\beta$ to construct an effective action. In order for the effective action to be invariant under worldvolume diffeomorphisms and flavor transformations, we must construct invariant combinations of the superpullbacks and thermal data. Toward this end, let us collect a number of objects that can be constructed from $\mathfrak{g}_{ij}$ and $\mathbb{B}_k$ which can appear in the action. From the super-metric $\mathfrak{g}_{ij}$ we construct an inverse super-metric $\mathfrak{g}^{ij}$, which satisfies $\mathfrak{g}^{ik}\mathfrak{g}_{jk} = \delta^i{}_j$. Neglecting the ghosts for simplicity, this inverse super-metric is given by

$$\mathfrak{g}^{ij} = g_r^{ij} - \bar{\theta}\theta\, g_r^{ik} g_r^{jl}\left(g_{a\,kl} + \mathcal{L}_{\rho_a} g_{r\,kl}\right), \tag{49}$$

where $g_r^{ij}$ is the inverse of $g_{r\,ij}$. With the super-metric $\mathfrak{g}_{ij}$ and its inverse, we construct a super-Christoffel connection and Riemann curvature, by the usual formulae,

$$\begin{aligned}
\Gamma^i{}_{jk} &= \frac{1}{2}\mathfrak{g}^{il}\left(\partial_j \mathfrak{g}_{kl} + \partial_k \mathfrak{g}_{jl} - \partial_l \mathfrak{g}_{jk}\right), \\
\mathbb{R}^i{}_{jkl} &= \partial_k \Gamma^i{}_{jl} - \partial_l \Gamma^i{}_{jk} + \Gamma^i{}_{mk}\Gamma^m{}_{jl} - \Gamma^i{}_{ml}\Gamma^m{}_{jk} .
\end{aligned} \tag{50}$$

Similarly, from $\mathbb{B}_i$ we construct a super-field strength,

$$\mathbb{G}_{ij} = \partial_i \mathbb{B}_j - \partial_j \mathbb{B}_i \,. \tag{51}$$

The super-connection $\Gamma^i{}_{jk}$ is invariant under target space diffeomorphisms and varies as a connection under worldvolume diffeomorphisms. So, we use $\Gamma^i{}_{jk}$ to build a worldvolume covariant derivative which we notate as $\mathbb{W}_i$. It acts on worldvolume tensors in the usual way, e.g.

$$\mathbb{W}_i \beta^j = \partial_i \beta^j + \Gamma^j{}_{ki} \beta^k \,, \tag{52}$$

and, under it, the super-metric is covariantly constant,

$$\mathbb{W}_i \mathfrak{g}_{jk} = \partial_i \mathfrak{g}_{jk} - \Gamma^l{}_{ji} \mathfrak{g}_{lk} - \Gamma^l{}_{ki} \mathfrak{g}_{jl} = 0 \,. \tag{53}$$

Apart from the field strengths and covariant derivatives, there are two important objects that we may construct out of the superpullbacks and the initial data,

$$\mathbb{T} = \frac{1}{\sqrt{-\beta^i \beta^j \mathfrak{g}_{ij}}} \,, \quad \text{and} \quad \nu = \beta^i \mathbb{B}_i + \Lambda_\beta \,, \tag{54}$$

which are scalars under worldvolume diffeomorphisms and $U(1)$ transformations (using that $\Lambda_\beta$ varies under $U(1)$ transformations as $\delta_\Lambda \Lambda_\beta = -\beta^i \partial_i \Lambda$). We will see later that the bottom components of these superfields are the local temperature and the reduced chemical potential of the fluid.

Crucially, when the $a$-fields vanish, $Q$ acts as $\frac{\partial}{\partial \theta}$ on the basic superfields $\mathfrak{g}_{ij}$ and $\mathbb{B}_i$, as well as on the other objects constructed from them, including $\Gamma^i{}_{jk}$, $\mathbb{R}^i{}_{jkl}$, $\mathbb{G}_{ij}$, and $\mathbb{W}_i$. To ensure that our action is invariant under $Q$ when the $a$-fields vanish, we demand invariance under $\frac{\partial}{\partial \theta}$, even when the $a$-sources are nonzero, and do so from here on. That is, we impose invariance under a spurionic symmetry, which we denote as $\delta_Q$, which acts as $\frac{\partial}{\partial \theta}$ on $\mathbb{X}^\mu$, $\mathfrak{g}_{ij}$, etc. By construction, the spurionic symmetry $\delta_Q$ becomes a genuine symmetry when the $a$-fields vanish. In the remainder of this Section we will parameterize the most general such action.

There are four basic Grassmann-odd objects $\{\frac{\partial}{\partial \theta}, \frac{\partial}{\partial \bar\theta}, \theta, \bar\theta\}$ at hand. All but $\theta$ anti-commute with $\frac{\partial}{\partial \theta}$ and so may appear in our action. With some foresight, we package them into the three quantities

$$D_\theta \equiv \frac{\partial}{\partial \theta} - i\bar\theta \delta_\beta \,, \qquad D_{\bar\theta} \equiv \frac{\partial}{\partial \bar\theta} \,, \tag{55}$$

and $\bar\theta$. Here $D_\theta$ and $\bar\theta$ have ghost number $-1$, and $D_{\bar\theta}$ ghost number $+1$. As a result, the most general action invariant under the transformation $\frac{\partial}{\partial \theta}$ is of the form

$$S = \int d^d \sigma \, d\theta \, d\bar\theta \sqrt{-\mathfrak{g}} \, L(\mathfrak{g}_{ij}, \mathbb{B}_k, \mathbb{W}_l; D_\theta, D_{\bar\theta}, \bar\theta; \beta^i, \Lambda_\beta) \,. \tag{56}$$

## 2.4 The reality condition

Having accounted for target and worldvolume diffeomorphism and flavor invariance and the Schwinger-Keldysh symmetry, it remains to impose the reality condition (6) and the KMS symmetry (7). With a Lagrangian at hand, it is straightforward to impose the reality condition, which is equivalent to

$$W[A_1, A_2] = -W[A_2^*, A_1^*]^* \,. \tag{57}$$

Following our previous work [18], we impose this condition on our effective action by defining a transformation $R$ which includes complex conjugation and whose action on the sources is given by $A_1 \to A_2^*$ and $A_2 \to A_1^*$. We then demand that $S_{eff}$ is odd under $R$. For our theory of fluids, the dynamical fields transform under $R$ as

$$
\begin{aligned}
R(X_1^\mu) &= X_2^\mu\,, & R(X_2^\mu) &= X_1^\mu\,, & R(X_{\bar{g}}^\mu) &= -X_{\bar{g}}^\mu\,, & R(X_g^\mu) &= X_g^\mu\,, \\
R(C_1) &= C_2\,, & R(C_2) &= C_1\,, & R(C_{\bar{g}}) &= -C_{\bar{g}}\,, & R(C_g) &= C_g\,,
\end{aligned}
\tag{58a}
$$

the external fields as

$$
R(g_{1\,\mu\nu}(x)) = g_{2\,\mu\nu}(x)\,, \qquad R(g_{2\,\mu\nu}(x)) = g_{1\,\mu\nu}(x)\,,
\tag{58b}
$$

and the Grassmannian coordinates as

$$
R(\theta) = -\theta\,, \qquad R(\bar{\theta}) = \bar{\theta}\,.
\tag{58c}
$$

So defined, the dynamical superfields and super-pullbacks are invariant under $R$

$$
R(\mathbb{X}^\mu) = \mathbb{X}^\mu\,, \qquad R(\mathbb{C}) = \mathbb{C}\,, \qquad R(\mathfrak{g}_{ij}) = \mathfrak{g}_{ij}\,, \qquad R(\mathbb{B}_i) = \mathbb{B}_i\,,
\tag{58d}
$$

as are the Grassmann-odd objects

$$
R(iD_\theta) = iD_\theta\,, \qquad R(D_{\bar{\theta}}) = D_{\bar{\theta}}\,, \qquad R(\bar{\theta}) = \bar{\theta}\,.
\tag{58e}
$$

Demanding the effective action to be odd under $R$ and writing the action as a superspace integral,

$$
S_{eff} = \int d^d\sigma d\theta d\bar{\theta}\sqrt{-\mathfrak{g}}\,L\,,
\tag{58f}
$$

we see that the reality condition implies that $L$ is invariant under $R$.

Putting the pieces together, we find that the most general action which respects target and worldvolume diffeomorphism/flavor invariance, the Schwinger-Keldysh symmetry, and the reality condition, (i.e., all the symmetries of the problem except for KMS), takes the form

$$
S_{eff} = \int d^d\sigma d\theta d\bar{\theta}\sqrt{-\mathfrak{g}}\,L\big(\mathfrak{g}_{ij}, \mathbb{B}_k, \mathbb{W}_l; iD_\theta, D_{\bar{\theta}}, \bar{\theta}; \beta^i, \Lambda_\beta\big)\,,
\tag{59}
$$

where now $L$ is a real function of its arguments, is invariant under worldvolume diffeomorphisms and flavor transformations, and has ghost number 0. It remains to impose the KMS symmetry. This is the subject of the next Subsection.

## 2.5   The KMS symmetry

The KMS symmetry (7) is a $\mathbb{Z}_2$ symmetry. A natural way to impose a $\mathbb{Z}_2$ symmetry is to construct a Lagrangian $L$ which satisfies all other symmetries of the problem and add to it its $\mathbb{Z}_2$ image which we denote by $\widetilde{L}$. This way, the action $\int L + \widetilde{L}$ will be $\mathbb{Z}_2$-invariant and satisfy all other symmetries of the problem as long as $\widetilde{L}$ does. As it turns out, the KMS $\mathbb{Z}_2$ symmetry does not commute with the Schwinger-Keldysh symmetry associated with $\delta_Q$. Demanding that the group axioms are satisfied, we infer the existence of a second, emergent Grassmann-odd symmetry $\delta_{\overline{Q}}$, which is exchanged with $\delta_Q$ under KMS. Towards the end of this Section we will see that the appearance of this new symmetry implies that the Lagrangian $L$ defined (59) should be further modified so that it does not depend explicitly on $\bar{\theta}$. Once we do so, actions of the form $\int L + \widetilde{L}$ will be invariant under all symmetries of the problem.

This Section is structured as follows. In 2.5.1 we derive the KMS symmetry (7) for Lagrangian theories, and further show that symmetry is best thought of as a family of $\mathbb{Z}_2$

symmetries. We then implement the KMS symmetry by imposing a single $\mathbb{Z}_2$ symmetry on the worldvolume. (The authors of [14,21] used a similar mechanism for ensuring KMS symmetry, which they termed a dynamical KMS symmetry.) We work out the action of this worldvolume KMS symmetry on bosonic and ghost fields in 2.5.2. In 2.5.3, we proceed to demonstrate the existence of an emergent Grassmann-odd symmetry $\delta_{\overline{Q}}$. Finally in 2.5.4 we put all the pieces together and write effective actions invariant under all symmetries.

### 2.5.1 A family of $\mathbb{Z}_2$ symmetries

We begin with the derivation of the KMS symmetry (7) for Lagrangian theories in Minkowski space. Given an initial state density matrix $\rho_{-\infty} \propto e^{-bH}$ with $H$ the Hamiltonian, the Schwinger-Keldysh partition function $Z = \mathrm{tr}\left(U_1 \rho_{-\infty} U_2^\dagger\right)$ may be written as a functional integral

$$Z[A_1, A_2] = \int [d\phi_1][d\phi_2] \exp\left(\frac{i}{\hbar}(S[\phi_1; A_1] - S[\phi_2; A_2])\right), \tag{60}$$

where $\phi$ collectively represents the quantum fields, $A$ the external fields, and $S[\phi; A]$ is the action. We assume that this action is real, diffeomorphism- and flavor-invariant, and CPT-invariant. All fields, quantum and external, obey boundary conditions at future and past infinity,

$$\lim_{t \to \infty} (\phi_1(t, \vec{x}) - \phi_2(t, \vec{x})) = 0,$$
$$\lim_{t \to -\infty} (\phi_1(t, \vec{x}) - \phi_2(t - i\hbar b, \vec{x})) = 0. \tag{61}$$

We now define KMS-conjugated fields as

$$\phi_1^K(t, \vec{x}) = \eta_\phi \phi_1(-t, -x^1, \vec{x}_\perp), \qquad \phi_2^K(t, \vec{x}) = \eta_\phi \phi_2(-t - i\hbar b, -x^1, \vec{x}_\perp), \tag{62}$$

where $\eta_\phi$ is the CPT-eigenvalue of $\phi$. These tilde'd fields are obtained after the combination of CPT [5], complex conjugation, and, for $\phi_2^K$, a translation in imaginary time. The fact that the microscopic action $S$ is real, diffeomorphism-invariant, and CPT-invariant implies that

$$S[\phi_1; A_1] = S[\phi_1^K; A_1^K], \qquad S[\phi_2; A_2] = S[\phi_2^K; A_2^K], \tag{63}$$

where

$$A_1^K(t, \vec{x}) = \eta_\phi A_1(-t, -x^1, \vec{x}_\perp), \qquad A_2^K(t, \vec{x}) = \eta_\phi A_2(-t - i\hbar b, -x^1, \vec{x}_\perp), \tag{64}$$

so that

$$Z[A_1, A_2] = \int [d\phi_1^K][d\phi_2^K] \exp\left(\frac{i}{\hbar}(S[\phi_1^K; A_1^K] - S[\phi_2^K; A_2^K])\right). \tag{65}$$

To obtain (7) it remains to deduce the boundary conditions on the KMS-conjugated fields that follow from those of the ordinary fields, c.f, (61). We find

$$\lim_{t \to \infty} (\phi_1^K(t, \vec{x}) - \phi_2^K(t, \vec{x})) = \eta_\phi \lim_{t \to \infty} (\phi_1(-t, -x^1, \vec{x}_\perp) - \phi_2(-t - i\hbar b, -x^1, \vec{x}_\perp))$$
$$= \eta_\phi \lim_{t' \to -\infty} (\phi_1(t', \vec{x}') - \phi_2(t' - i\hbar b, \vec{x}')) \tag{66a}$$
$$= 0,$$

---

[5] We can take the action of CPT on Minkowski spacetime in any dimension to be the combination of $t \to -t$ and $x^1 \to -x^1$ while leaving the other coordinates invariant.

where we have defined $t' = -t$ and $\vec{x}' = (-x^1, \vec{x}_\perp)$. Similarly,

$$
\begin{aligned}
\lim_{t \to -\infty} \left( \phi_1^K(t, \vec{x}) - \phi_2^K(t - i\hbar b, \vec{x}') \right) = \eta_\phi \lim_{t \to -\infty} \big( \phi_1(-t, -x^1, \vec{x}_\perp) \\
- \phi_2(-(t - i\hbar b) - i\hbar b, -x^1, \vec{x}_\perp) \big) \\
= \eta_\phi \lim_{t' \to \infty} \left( \phi_1(t', \vec{x}') - \phi_2(t', \vec{x}') \right) \\
= 0 \, .
\end{aligned}
\tag{66b}
$$

These boundary conditions are precisely those appropriate for a Schwinger-Keldysh partition function with initial state $e^{-bH}$. Combined with (65), we find the KMS symmetry

$$
\begin{aligned}
Z[A_1(t, \vec{x}), A_2(t, \vec{x})] &= Z[A_1^K(t, \vec{x}), A_2^K(t, \vec{x})] \\
&= Z[\eta_A A_1(-t, -x^1, \vec{x}_\perp), \eta_A A_2(-t - i\hbar b, -x^1, \vec{x}_\perp)] \, .
\end{aligned}
\tag{67}
$$

Acting with this series of manipulations twice, we end up back where we started. The KMS symmetry is $\mathbb{Z}_2$. Further, we note that because the initial state $\exp(-bH)$ is CPT-invariant, the KMS symmetry relates $Z$ to a partition function with KMS-conjugated sources in the same state.

It is straightforward to write this result covariantly in a more general spacetime. The most general thermal initial state $\rho_{-\infty} \propto \exp(-b\mathcal{H})$ is characterized by a grand potential $b\mathcal{H}$ which acts on fields via a combination of a Lie derivative along a timelike vector $b^\mu$ and a flavor gauge transformation $\Lambda_b$. We denote this combined transformation by $\delta_b$. See e.g. [18, 36] for details. In this language, the thermal translation $t \to t - i\hbar b$ is a translation along the integral curves of $b^\mu$ by an affine parameter $-i\hbar$, enacted by the differential operator $e^{-i\hbar \delta_b}$.

The KMS transformation includes a CPT-flip. A general initial state is not CPT-invariant. For example, a chemical potential flips sign under CPT. We refer to the CPT-flipped grand potential as $b\mathcal{H}^{\mathrm{CPT}}$, and the corresponding generator as $\delta_b^{\mathrm{CPT}}$. The covariant KMS symmetry relates the partition function in the initial state $e^{-b\mathcal{H}}$ to one in the initial state $e^{-b\mathcal{H}^{\mathrm{CPT}}}$. Additionally, on a more general spacetime, CPT does not necessarily act as $(t, \vec{x}) \to (-t, -x^1, \vec{x}_\perp)$. In what follows we denote the action of a CPT transformation on spacetime as $\Theta$.

As before, for a theory with a functional integral description we have

$$
Z[A_1, A_2] = \int [d\phi_1][d\phi_2] \exp\left( \frac{i}{\hbar} (S[\phi_1; A_1] - S[\phi_2; A_2]) \right),
\tag{68}
$$

with the boundary conditions

$$
\begin{aligned}
\lim_{t \to \infty} (\phi_1 - \phi_2) &= 0 \, , \\
\lim_{t \to -\infty} \left( \phi_1 - e^{-i\hbar \delta_b} \phi_2 \right) &= 0 \, .
\end{aligned}
\tag{69}
$$

The only place $\delta_b$ appears is in the infinite past, and so, in fact, we can take the past boundary condition to be

$$
\lim_{t \to -\infty} \left( \phi_1 - e^{-i\hbar \delta_{b'}} \phi_2 \right) = 0 \, ,
\tag{70}
$$

where $\delta_{b'}$ is any transformation which smoothly asymptotes to $\delta_b$ in the far past. A covariant expression for the KMS-conjugated fields (62) is

$$
\phi_1^K = \eta_\phi \Theta^* \phi_1 \, , \qquad \phi_2^K = \eta_\phi \Theta^* (e^{-i\hbar \delta_{b'}} \phi_2) \, ,
\tag{71}
$$

where $\Theta^*$ specifies a CPT transformation followed by complex conjugation of its argument. With some prescience, we find it useful to define a linear operation $K$ such that

$$
\phi_1^K = \Theta^* K(\phi_1) \, , \qquad \phi_2^K = \Theta^* K(\phi_2) \, ,
\tag{72}
$$

i.e.,

$$K(\phi_1) = \eta_\phi \phi_1\,, \qquad K(\phi_2) = \eta_\phi e^{-i\hbar\delta_{b'}}\phi_2\,, \tag{73}$$

so that KMS conjugation is given by the action of $K$ followed by the linear operation $\Theta^*$. We define $K$ so that it acts on $(b'^\mu, \Lambda_{b'})$ and derivatives as

$$K(b'^\mu) = -\eta_\mu b'^\mu\,, \qquad K(\Lambda_{b'}) = -\Lambda_{b'}\,, \qquad K\frac{\partial}{\partial x^\mu} = \eta_\mu \frac{\partial}{\partial x^\mu}\,. \tag{74}$$

In Minkowski spacetime, where CPT acts by flipping $x^0$ and $x^1$, we have

$$\eta_\mu = \begin{cases} -1 & \mu = 0, 1\,, \\ 1 & \text{otherwise} \end{cases}. \tag{75}$$

More generally, they are such that

$$K(\delta_{b'}) = -\delta_{b'}\,. \tag{76}$$

So defined, $K$ squares to the identity,

$$K^2(\phi_1) = K(\eta_\phi \phi_1) = \phi_1\,, \qquad K^2(\phi_2) = K(\eta_\phi e^{-i\hbar\delta_{b'}}\phi_2) = K(e^{-i\hbar\delta_{b'}})K(\eta_\phi\phi_2) = \phi_2\,, \tag{77}$$

as it ought: the KMS transformation is the combination of $K$ and $\Theta^*$, and since the KMS transformation and $\Theta^*$ each square to the identity, so must $K$.

As before the underlying diffeomorphism, flavor, and CPT invariance of the action imply that

$$S[\phi_1; A_1] = S[\phi_1^K; A_1^K]\,, \qquad S[\phi_2; A_2] = S[\phi_2^K; A_2^K]\,. \tag{78}$$

Furthermore, the boundary conditions in the far past and future (69) imply that the KMS-conjugated fields obey the boundary conditions appropriate for a thermal partition function in an initial thermal state $\exp(-b\mathcal{H}^{\text{CPT}})$,

$$\lim_{t\to\infty}(\phi_1^K - \phi_2^K) = \eta_\phi\Theta^* \lim_{t\to-\infty}(\phi_1(x) - e^{-i\hbar\delta_{b'}}\phi_2(x)) = 0\,,$$
$$\lim_{t\to-\infty}(\phi_1^K - e^{i\hbar\delta_{b'}^{\text{CPT}}}\phi_2^K) = \eta_\phi\Theta^* \lim_{t\to\infty}(\phi_1(x) - \phi_2(x)) = 0 \tag{79}$$

(In the second line it should be noted that, with our conventions, $\exp(i\hbar\delta_{b'}^{\text{CPT}})$ acts on the reversed time as $t \to t - i\hbar b'$, and so this is the appropriate past boundary condition corresponding to the initial state $\rho_{-\infty} \propto \exp(-b\mathcal{H}^{\text{CPT}})$.) This implies

$$Z[A_1, A_2; \delta_{b'}] = Z[A_1^K, A_2^K; \delta_{b'}^{\text{CPT}}]\,, \tag{80}$$

for any $\delta_{b'}$ which asymptotes to $\delta_b$. Acting with the KMS transformation twice brings us back to the original partition function, and so each of these symmetries is $\mathbb{Z}_2$.

Ultimately, the existence of this infinite family of $\mathbb{Z}_2$ symmetries is due to diffeomorphism and flavor-invariance. For two different transformations $\delta_{b_1}^{\text{CPT}}$ and $\delta_{b_2}^{\text{CPT}}$ which both asymptote to $\delta_b^{\text{CPT}}$ in the far past, there is a diffeomorphism and flavor transformation which vanishes at infinity and which sends $\delta_{b_1}^{\text{CPT}} \to \delta_{b_2}^{\text{CPT}}$, giving

$$Z[A_1^K, A_2^K; \delta_{b_1}^{\text{CPT}}] = Z[A_1^K, A_2^K; \delta_{b_2}^{\text{CPT}}]\,, \tag{81}$$

where the conjugated field $A_2^K$ on the left hand side is obtained from the ordinary one using $\delta_{b_1}$, $A_2^K = \eta_A\Theta^*(e^{-i\hbar\delta_{b_1}}A_2)$, and the one on the right hand side using $\delta_{b_2}$. Thus, it is possible to implement the KMS symmetry in the effective action by imposing (80) for a particular $b'$ together with target-space diffeomorphism/flavor-invariance.

### 2.5.2 Worldvolume KMS symmetry

In this work we implement the KMS symmetry (80) by imposing a $\mathbb{Z}_2$ KMS symmetry on the worldvolume. A priori, it is not clear that a worldvolume KMS symmetry will impose the proper KMS symmetry (80), which is stated in the physical space. Towards the end of this Section, we will provide a perturbative proof that indeed our worldvolume KMS symmetry imposes the KMS symmetry for a particular $\delta_{b'}$ (80).

Let us start by introducing a vector field $\beta^i$ and flavor gauge transformation $\Lambda_\beta$, which together generate a worldvolume transformation $\delta_\beta$. We impose boundary conditions on the $X^\mu$'s and $C$'s so that they are trivial in the far past,

$$\lim_{\sigma^0 \to -\infty} X_s^\mu = \delta_i^\mu \sigma^i, \qquad \lim_{\sigma^0 \to -\infty} C_s = 0. \tag{82}$$

We choose the worldvolume $\delta_\beta$ to be such that, in the far past, it coincides with $\delta_b$ when it is pushed forward to the physical space. Next, we will use the worldvolume $\delta_\beta$ to define KMS-conjugated versions of our dynamical fields and pullbacks. As in (73), we find it convenient to split the action of KMS conjugation into two: we denote the worldvolume CPT transformation as $\vartheta$, and it acts on the worldvolume coordinates as

$$\sigma^i \to (\vartheta \sigma)^i, \tag{83}$$

and define KMS conjugation as $\vartheta^* K$.

Note that an action which is invariant under $K$ will also be invariant under a worldvolume KMS transformation. To see this consider

$$S = \int d^d\sigma \sqrt{-g_r}\, L(\phi, A), \tag{84}$$

with real $L$. We find

$$
\begin{aligned}
K\left(\int d^d\sigma \sqrt{-g_r}\, L(\phi; A)\right) &= \int d^d\sigma \sqrt{-K(g_r)}\, L(K(\phi); K(A)) \\
&= \int d^d\sigma \vartheta^*\left(\sqrt{-K(g_r)}\, L(K(\phi); K(A))\right) \\
&= \int d^d\sigma \sqrt{-\vartheta^* K(g_r)}\, L(\vartheta^* K(\phi); \vartheta^* K(A)) \\
&= \int d^d\sigma \sqrt{-g_r^K}\, L(\phi^K; A^K).
\end{aligned}
\tag{85}
$$

Thus,

$$\int d^d\sigma \sqrt{-g_r}\, L(\phi, A) = \int d^d\sigma \sqrt{-g_r^K}\, L(\phi^K; A^K) \tag{86a}$$

if and only if

$$S = K(S). \tag{86b}$$

Let us now state more precisely our strategy for constructing a KMS invariant action, outlined at the beginning of this Section. Given a Lagrangian $L$ we construct an action $S = \int d^d\sigma \left(\sqrt{-g}L + K(\sqrt{-g}L)\right)$. Such an action will clearly be KMS invariant due to (86) and will have the same symmetries as $\int d^d\sigma \sqrt{-g}L$ as long as $\int d^d\sigma K(\sqrt{-g}L)$ retains those symmetries. The action (59) satisfies all the symmetries of the problem but for the KMS symmetry. To proceed we wish to construct an appropriate $K$, identify its action on the other symmetries of $\int d^d\sigma \sqrt{-g}L$ and then tune the action in (59) so that worldvolume and target-space diffeomorphism/flavor invariance, the Schwinger-Keldysh symmetry, and the reality condition are retained after acting on it with $K$.

Let us start by defining the action of $K$ on the dynamical bosonic fields following (73). Throughout we restrict our attention to spacetimes that are asymptotically flat, so that we can write CPT transformations explicitly. However, our final effective action may be written on more general spacetimes (e.g., a cylinder, $\mathbb{R} \times \mathbb{S}^{d-1}$). Our strategy is to define $K$ such that $\vartheta^* K(A(x)) = A^K(x)$ when acting on target space sources, with $A^K$ given by (64). We further define the action of $K$ on the external data $\beta^i$ and $\Lambda_\beta$ and on the dynamical fields $X^\mu$ and $C$ in a way which is commensurate with its action on the sources. Let us denote

$$K(\phi) = \eta_\phi \widetilde{\phi} \,, \tag{87}$$

where $\phi$ is a source, thermal parameter or dynamical field. Given (64) we define the action of $K$ on sources as

$$
\begin{aligned}
K(g_{1\,\mu\nu}(x)) &= \eta_\mu \eta_\nu g_{1\,\mu\nu}(\eta x)\,, & K(g_{2\,\mu\nu}(x)) &= \eta_\mu \eta_\nu g_{2\,\mu\nu}(\eta x)\,, \\
K(B_{1\,\mu}(x)) &= \eta_\mu B_{1\,\mu}(\eta x)\,, & K(B_{2\,\mu}(x)) &= \eta_\mu B_{2\,\mu}(\eta x)\,,
\end{aligned}
\tag{88}
$$

where

$$(\eta x)^\mu = \eta_\mu x^\mu \,, \tag{89}$$

and

$$
\begin{aligned}
K(X_1^\mu) &= \eta_\mu X_1^\mu\,, & K(X_2^\mu) &= \eta_\mu e^{-i\hbar\delta_\beta} X_2^\mu \,. \\
K(C_1) &= C_1\,, & K(C_2) &= e^{-i\hbar\delta_\beta} C_2 \,,
\end{aligned}
\tag{90}
$$

and define the action of $K$ on the thermal data and worldvolume derivatives to be

$$K(\beta^i) = -\eta_i \beta^i\,, \qquad K(\Lambda_\beta) = -\Lambda_\beta\,, \qquad K\left(\frac{\partial}{\partial\sigma^i}\right) = \eta_i \frac{\partial}{\partial\sigma^i}\,, \tag{91}$$

so that $K(\delta_\beta) = -\delta_\beta$. So defined $K$ squares to the identity $K^2 = 1$.

Recall that in order to implement the Schwinger-Keldysh symmetry we have switched from the $1/2$ basis to the $r/a$ basis. In this basis we find that

$$K(X_r^\mu) = \eta_\mu \widetilde{X}_r^\mu\,, \qquad K(X_a^\mu) = \eta_\mu \widetilde{X}_a\,, \qquad K(C_r) = \widetilde{C}_r\,, \qquad K(C_a) = \widetilde{C}_a\,, \tag{92}$$

where

$$
\begin{aligned}
\widetilde{X}_r^\mu(\sigma) &= \lim_{\hbar\to 0} \frac{\widetilde{X}_1^\mu + \widetilde{X}_2^\mu}{2} = \lim_{\hbar\to 0} \frac{X_1^\mu(\sigma) + e^{-i\hbar\delta_\beta} X_2^\mu(\sigma)}{2} \\
&= X_r^\mu(\sigma)\,, \\
\widetilde{X}_a^\mu(\sigma) &= \lim_{\hbar\to 0} \frac{\widetilde{X}_1^\mu - \widetilde{X}_2^\mu}{\hbar} = \lim_{\hbar\to 0} \frac{X_1^\mu(\sigma) - e^{-i\hbar\delta_\beta} X_2^\mu(\sigma)}{\hbar} \\
&= X_a^\mu(\sigma) + i\delta_\beta X_r^\mu(\sigma) = X_a^\mu(\sigma) + i\beta^i \partial_i X_r^\mu(\sigma)\,,
\end{aligned}
\tag{93}
$$

and

$$
\begin{aligned}
\widetilde{C}_r(\sigma) &= \lim_{\hbar\to 0} \frac{\widetilde{C}_1 + \widetilde{C}_2}{2} = \lim_{\hbar\to 0} \frac{C_1(\sigma) + e^{-i\hbar\delta_\beta} C_2(\sigma)}{2} \\
&= C_r(\sigma)\,, \\
\widetilde{C}_a(\sigma) &= \lim_{\hbar\to 0} \frac{\widetilde{C}_1 - \widetilde{C}_2}{\hbar} = \lim_{\hbar\to 0} \frac{C_1(\sigma) - e^{-i\hbar\delta_\beta} C_2(\sigma)}{\hbar} \\
&= C_a(\sigma) + i\delta_\beta C_r(\sigma) = C_a(\sigma) + i\left(\beta^i \partial_i C_r(\sigma) + \Lambda_\beta(\sigma)\right)\,.
\end{aligned}
\tag{94}
$$

Note that $\widetilde{X}_r^\mu = X_r^\mu$ and $\widetilde{C}_r = C_r$, and so the $r$- and $\tilde{r}$-combinations are equal in the $\hbar \to 0$ limit. Using the CPT-eigenvalues of the $X^\mu$ and $C$, we find that,

$$K(X_r^\mu) = \eta_\mu X_r^\mu, \qquad K(C_r) = C_r, \tag{95}$$

and that $K$ exchanges the $a$-combination with the $\tilde{a}$-combination,

$$K(X_a^\mu) = \eta_\mu \widetilde{X}_a^\mu, \qquad K(\widetilde{X}_a^\mu) = \eta_\mu X_a^\mu, \qquad K(C_a) = \widetilde{C}_a, \qquad K(\widetilde{C}_a) = C_a. \tag{96}$$

Since the action (59) depends on the dynamical fields only through the pullbacks of the sources, our goal is to study the action of $K$ on such pullbacks. We find that

$$K(g_{r\,ij}) = \eta_i \eta_j \widetilde{g}_{r\,ij}, \qquad K(g_{a\,ij}) = \eta_i \eta_j \widetilde{g}_{a\,ij}, \tag{97}$$

where

$$
\begin{aligned}
\widetilde{g}_{r\,ij}(\sigma) &= \lim_{\hbar \to 0} \frac{1}{2}\Big(g_{1\,\mu\nu}(X_r(\sigma)) + e^{-i\hbar\delta_\beta} g_{2\,\mu\nu}(X_r(\sigma))\Big)\partial_i X_r^\mu \partial_j X_r^\nu \\
&= g_{r\,ij}(\sigma), \\
\widetilde{g}_{a\,ij}(\sigma) &= \lim_{\hbar \to 0} \frac{\Big(g_{1\,\mu\nu}(X_r(\sigma)) - e^{-i\hbar\delta_\beta} g_{2\,\mu\nu}(X_r(\sigma))\Big)\partial_i X_r^\mu \partial_j X_r^\nu}{\hbar} \\
&= \big(g_{a\,ij}(\sigma) + i\delta_\beta g_{r\,ij}(\sigma)\big).
\end{aligned}
\tag{98}
$$

Here, when $\delta_\beta$ acts on $g_{2\,\mu\nu}$ we are using $(X_r, \Lambda_r)$ to map the worldvolume transformation $\delta_\beta$ to one in the target space. Note that while $K$ maps the dynamical fields $X_r^\mu$ and $X_a^\mu$ to their tilde'd versions, $\widetilde{X}_r^\mu$ and $\widetilde{X}_a^\mu$, it maps the $r$- and $a$-metrics $g_{r\,ij}$ and $g_{a\,ij}$ to their tilde'd versions up to an overall sign. Similarly we have

$$K(B_{r\,i}) = \eta_i \widetilde{B}_{r\,i}, \qquad K(B_{a\,i}) = \eta_i \widetilde{B}_{a\,i}, \tag{99}$$

where

$$
\begin{aligned}
\widetilde{B}_{r\,i}(\sigma) &= \lim_{\hbar \to 0} \left[\frac{1}{2}\Big(B_{1\,\mu}(X_r(\sigma)) + e^{-i\hbar\delta_\beta} B_{2\,\mu}(X_r(\sigma))\Big)\partial_i X_r^\mu + \partial_i \widetilde{C}_r(\sigma)\right] \\
&= B_{r\,i}(\sigma), \\
\widetilde{B}_{a\,i}(\sigma) &= \lim_{\hbar \to 0} \frac{\Big(B_{1\,\mu}(X_r(\sigma)) - e^{-i\hbar\delta_\beta} B_{2\,\mu}(X_r(\sigma))\Big)\partial_i X_r^\mu}{\hbar} \\
&= \big(B_{a\,i}(\sigma) + i\delta_\beta B_{r\,i}(\sigma)\big) = \eta_{B_i}\Big(B_{a\,i}(\sigma) + i\big(\mathcal{L}_\beta B_{r\,i}(\sigma) + \partial_i \Lambda_\beta\big)\Big).
\end{aligned}
\tag{100}
$$

Let us turn our attention to the ghost fields. A priori, there seems to be much freedom in the possible action of $K$ on ghosts. However, we may constrain $K$ by demanding that it be commensurate with the ghost number symmetry. That is, we require that $K$ either preserves or flips the ghost number. On bosonic fields we have $K^2 = 1$, but on ghosts we allow for the possibility that it squares to either $+1$ or $-1$, so that $K^2 = 1$ when acting on the effective action. In the first case, we have that $K^2 = 1$, and in the second that $K^2 = (-1)^g$ where $g$ is ghost number. We will see shortly that the former is more restrictive than the latter.

The possible actions of $K$ on $\mathbb{X}$ and $\mathbb{C}$ which preserve ghost number are of the form, $K(C_{\bar{g}}) = \pm C_{\bar{g}}$ and $K(C_g) = C_g$. The possible actions of $K$ on the dynamical fields which

flip ghost number are $K(C_{\bar{g}}) = \pm\lambda C_g$ and $K(C_g) = \lambda^{-1}C_{\bar{g}}$. While we could carry out a full analysis of all these possibilities, we focus here on two,

$$K(X_g^\mu) = \begin{cases} X_g^\mu & K^2 = 1 \\ X_{\bar{g}}^\mu & K^2 = (-1)^g \end{cases}, \qquad K(X_{\bar{g}}^\mu) = \begin{cases} X_{\bar{g}}^\mu & K^2 = 1 \\ -X_g^\mu & K^2 = (-1)^g \end{cases}$$

$$K(C_g) = \begin{cases} C_g & K^2 = 1 \\ C_{\bar{g}} & K^2 = (-1)^g \end{cases}, \qquad K(C_{\bar{g}}) = \begin{cases} C_{\bar{g}} & K^2 = 1 \\ -C_g & K^2 = (-1)^g \end{cases}, \tag{101a}$$

which are compatible with

$$K(\theta) = \begin{cases} \theta & K^2 = 1 \\ -\bar{\theta} & K^2 = (-1)^g \end{cases}, \qquad K(\bar{\theta}) = \begin{cases} \bar{\theta} & K^2 = 1 \\ \theta & K^2 = (-1)^g \end{cases}. \tag{101b}$$

Thus,

$$K(\mathbb{X}^\mu) = \widetilde{\mathbb{X}}^\mu, \qquad K(\mathbb{C}) = \widetilde{\mathbb{C}}, \tag{102}$$

where we have defined

$$\widetilde{\mathbb{X}}^\mu = X_r^\mu + \theta X_{\bar{g}}^\mu + \bar{\theta}X_g^\mu + \bar{\theta}\theta\widetilde{X}_a^\mu, \\ \widetilde{\mathbb{C}} = C_r + \theta C_{\bar{g}} + \bar{\theta}C_g + \bar{\theta}\theta\widetilde{C}_a. \tag{103}$$

Acting with $K$ on the super-pullbacks $\mathfrak{g}_{ij}$ and $\mathbb{B}_k$, we find

$$K(\mathfrak{g}_{ij}) = \eta_i\eta_j\widetilde{\mathfrak{g}}_{ij}, \qquad K(\mathbb{B}_i) = \eta_i\widetilde{\mathbb{B}}_i, \tag{104}$$

where

$$\widetilde{\mathfrak{g}}_{ij} = g_{r\,ij}(\widetilde{\mathbb{X}}) + \bar{\theta}\theta\,g_{a\,ij}(\widetilde{\mathbb{X}}) = \widetilde{g}_{r\,ij}(\mathbb{X}) + \bar{\theta}\theta\,\widetilde{g}_{a\,ij}(\mathbb{X}), \\ \widetilde{\mathbb{B}}_i = B_{r\,i}(\widetilde{\mathbb{X}}, \widetilde{\mathbb{C}}) + \bar{\theta}\theta\,B_{a\,i}(\widetilde{\mathbb{X}}) = \widetilde{B}_{r\,i}(\mathbb{X}, \mathbb{C}) + \bar{\theta}\theta\widetilde{B}_{a\,i}(\mathbb{X}). \tag{105}$$

The other possibilities for actions of $K$ on the dynamical fields will be ruled out later on account of the group structure associated with the KMS symmetry and the Schwinger-Keldysh symmetry.

With the action of $K$ on the dynamical fields and sources at hand, our next task is to study its compatibility with the other symmetries we have discussed, namely doubled diffeomorphism/flavor invariance, the reality condition and the Schwinger-Keldysh symmetry. In the remainder of this Section we will show that $K$ is commensurate with the former two but incompatible with the Schwinger-Keldysh symmetry. We will resolve this mismatch in Section 2.5.3.

Given (58), it is straightforward to check that the reality condition commutes with $K$, ensuring that the $K$ transformation of (59) still satisfies the condition (6). The tilde'd super-pullbacks are invariant under target space transformations. To see this we require the $r$-transformations (26) and $a$-transformations (27) of $\mathbb{X}^\mu$ and $\mathbb{C}$ from which

$$\delta_\chi\widetilde{\mathbb{X}}^\mu = -\widetilde{\mathfrak{F}} = -\left(\xi_r^\mu(\widetilde{\mathbb{X}}) + \bar{\theta}\theta\xi_a^\mu(\widetilde{\mathbb{X}})\right), \\ \delta_\chi\widetilde{\mathbb{C}} = -\widetilde{\mathbb{A}} = -\left(\Lambda_r(\widetilde{\mathbb{X}}) + \bar{\theta}\theta\Lambda_a(\widetilde{\mathbb{X}})\right) \tag{106}$$

follows. Using the same sort of superspace argument in (42) that we used to show that $\mathfrak{g}_{ij}$ is an invariant pullback, it follows that $\widetilde{\mathfrak{g}}_{ij}$ and $\widetilde{\mathbb{B}}_i$ are invariant under $r/a$-transformations.

We can also check this invariance by expanding in components. We find

$$
\begin{aligned}
\widetilde{\mathfrak{g}}_{ij} = {} & g_{r\,ij} + \theta \mathcal{L}_{\bar{\psi}} g_{r\,ij} + \bar{\theta} \mathcal{L}_{\psi} g_{r\,ij} \\
& + \bar{\theta}\theta \left( g_{a\,ij} + \mathcal{L}_{\widetilde{\rho}_a} g_{r\,ij} + \frac{1}{2} \left( [\mathcal{L}_{\bar{\psi}}, \mathcal{L}_{\psi}] - \mathcal{L}_{[\bar{\psi},\psi]} \right) g_{r\,ij} \right) , \\
= {} & g_{r\,ij} + \theta \mathcal{L}_{\bar{\psi}} g_{r\,ij} + \bar{\theta} \mathcal{L}_{\psi} g_{r\,ij} \\
& + \bar{\theta}\theta \left( \widetilde{g}_{a\,ij} + \mathcal{L}_{\rho_a} g_{r\,ij} + \frac{1}{2} \left( [\mathcal{L}_{\bar{\psi}}, \mathcal{L}_{\psi}] - \mathcal{L}_{[\bar{\psi},\psi]} \right) g_{r\,ij} \right) ,
\end{aligned}
\tag{107a}
$$

and similarly,

$$
\begin{aligned}
\widetilde{\mathbb{B}}_i = {} & B_{r\,i} + \theta \mathcal{L}_{\bar{\psi}} B_{r\,i} + \bar{\theta} \mathcal{L}_{\psi} B_{r\,i} + \bar{\theta}\theta \big( \widetilde{B}_{a\,i}(\sigma) + \mathcal{L}_{\rho_a}(B_{r\,i}(\sigma) - \partial_i C_r(\sigma)) + \partial_i C_a(\sigma) \\
& + \frac{1}{2} \left( [\mathcal{L}_{\bar{\psi}}, \mathcal{L}_{\psi}] - \mathcal{L}_{[\bar{\psi},\psi]} \right) (B_{r\,i}(\sigma) - \partial_i C_r(\sigma)) \big) ,
\end{aligned}
\tag{107b}
$$

where $\widetilde{\rho}_a^i = \widetilde{X}_a^\mu (\partial_i X_r^\mu)^{-1}$. In Subsections 2.2 and 2.3 we showed how $r$- and $a$- transformations act on the various fields. Because the $\tilde{r}$-combinations equal the $r$-combinations as $\hbar \to 0$, they transform in the same way as before. Thus, all but the top components of $\widetilde{\mathfrak{g}}_{ij}$ and $\widetilde{\mathbb{B}}_i$ are manifestly invariant. The variations of the $\tilde{a}$-combinations of the dynamical fields follow from (26) and (27) and are given by

$$
\begin{aligned}
\delta_\chi \widetilde{X}_a^\mu(\sigma) &= -\widetilde{X}_a^\nu(\sigma) \partial_\nu \xi_r^\mu(X_r(\sigma)) - \xi_a^\mu(X_r(\sigma)) - X_{\bar{g}}^\nu X_g^\rho \partial_\nu \partial_\rho \xi_r^\mu(X_r(\sigma)) , \\
\delta_\chi \widetilde{C}_a(\sigma) &= -\widetilde{X}_a^\mu(\sigma) \partial_\mu \Lambda_r(X_r(\sigma)) - \Lambda_a(X_r(\sigma)) - X_{\bar{g}}^\mu X_g^\nu \partial_\mu \partial_\nu \Lambda_r(X_r(\sigma)) .
\end{aligned}
\tag{108}
$$

The $\tilde{a}$-pullbacks vary in the same way as the $a$-combinations, (35),

$$
\begin{aligned}
\delta_\chi \widetilde{g}_{a\,ij}(\sigma) &= \mathcal{L}_{\xi_a} g_{r\,ij}(\sigma) , \\
\delta_\chi \widetilde{B}_{a\,i}(\sigma) &= \mathcal{L}_{\xi_a}(B_{r\,i}(\sigma) - \partial_i C_r(\sigma)) + \partial_i \Lambda_a(X_r(\sigma)) ,
\end{aligned}
\tag{109}
$$

where the Lie derivatives are taken along $\xi_a^i(\sigma)$. As in (37), the $\tilde{a}$-pullbacks may be combined with $r$-fields into invariant pullbacks. We find that these pullbacks may be written in two equivalent ways. The invariant metric is

$$
\begin{aligned}
\widetilde{g}_{a\,ij}(\sigma) & + \mathcal{L}_{\rho_a} g_{r\,ij}(\sigma) + \frac{1}{2} \left( [\mathcal{L}_{\bar{\psi}}, \mathcal{L}_{\psi}] - \mathcal{L}_{[\bar{\psi},\psi]} \right) g_{r\,ij}(\sigma) \\
& = g_{a\,ij}(\sigma) + \mathcal{L}_{\widetilde{\rho}_a} g_{r\,ij}(\sigma) + \frac{1}{2} \left( [\mathcal{L}_{\bar{\psi}}, \mathcal{L}_{\psi}] - \mathcal{L}_{[\bar{\psi},\psi]} \right) g_{r\,ij}(\sigma) ,
\end{aligned}
\tag{110}
$$

where we have defined $\widetilde{\rho}_a^i = \widetilde{X}_a^\mu (\partial_i X_r^\mu)^{-1}$, and the invariant flavor field is

$$
\begin{aligned}
\widetilde{B}_{a\,i}(\sigma) & + \mathcal{L}_{\rho_a}(B_{r\,i}(\sigma) - \partial_i C_r(\sigma)) + \partial_i C_a(\sigma) + \frac{1}{2} \left( [\mathcal{L}_{\bar{\psi}}, \mathcal{L}_{\psi}] - \mathcal{L}_{[\bar{\psi},\psi]} \right) (B_{r\,i}(\sigma) - \partial_i C_r(\sigma)) \\
& = B_{a\,i}(\sigma) + \mathcal{L}_{\widetilde{\rho}_a}(B_{r\,i}(\sigma) - \partial_i C_r(\sigma)) + \partial_i \widetilde{C}_a(\sigma) + \frac{1}{2} \left( [\mathcal{L}_{\bar{\psi}}, \mathcal{L}_{\psi}] - \mathcal{L}_{[\bar{\psi},\psi]} \right) (B_{r\,i}(\sigma) - \partial_i C_r(\sigma)) .
\end{aligned}
\tag{111}
$$

Thus, the top components of $\widetilde{\mathfrak{g}}_{ij}$ and $\widetilde{\mathbb{B}}_i$ are the invariant completion of $\widetilde{g}_{a\,ij}$.

Our final task is to check for the compatibility of $K$ with $\delta_Q$. Recall that we enforce the Schwinger-Keldysh symmetry by demanding that our action is invariant under a Grassmann-odd transformation $\delta_Q$ (which acts on superfields as $\frac{\partial}{\partial \theta}$). Consider

$$
K(\delta_Q X_g^\mu) = K(X_a^\mu) = \widetilde{X}_a^\mu = X_a^\mu + i \delta_\beta X_r^\mu .
\tag{112}
$$

It is straightforward to check that the right-hand side of (112) is not $\delta_Q$ closed, let alone $\delta_Q$ exact. Thus,

$$
[\delta_Q, K] X_g^\mu \neq 0 .
\tag{113}
$$

Since $K$ and $\delta_Q$ do not commute then in order for them to form a group, there must exist an additional generator. This is the topic of the next Subsection.

### 2.5.3 Reconciling the KMS and Schwinger-Keldysh symmetries

Since $K\delta_Q \neq \delta_Q K$, the group axioms imply the existence of an additional, emergent Grassmann-odd symmetry $\delta_{Q'}$ obeying

$$K\delta_Q = -\delta_{Q'}K \, . \tag{114a}$$

Put differently, since $K$ and $\delta_Q$ do not commute when we add to (59) its image under $K$, that image will not be $\delta_Q$ invariant unless we ensure that (59) is invariant under both $\delta_{Q'}$ and $\delta_Q$. To understand the action of $\delta_{Q'}$ on the superfields, we note that (114a) leads to

$$K\delta_{Q'} = -\begin{cases} \delta_Q K \, , & K^2 = 1 \, , \\ (-1)^g \delta_Q K \, , & K^2 = (-1)^g \, . \end{cases} \tag{114b}$$

Using the transformation laws of the $X$- and $C$-supermultiplet under $\delta_Q$, the ghost number symmetry, and assuming that $\delta_{Q'}$ acts linearly on those supermultiplets, we are able to solve the intertwining conditions (114) for $\delta_{Q'}$ and $K$. We find that there are exactly two solutions, depending on whether $\delta_{Q'}$ has ghost number $+1$ or $-1$ given by (101). The other possible actions of $K$ on the fields, specified in the discussion prior to (101), are not allowed.

In the first solution where $K^2 = 1$, $\delta_{Q'}$ acts on $\mathbb{X}^\mu$ and $\mathbb{C}$ as

$$\delta_{Q'} \to \frac{\partial}{\partial\theta} - i\bar{\theta}\delta_\beta = D_\theta \, . \tag{115}$$

The Grassmann-odd objects which anticommute with $\delta_Q$ are $D_\theta$, $D_{\bar{\theta}}$, and $\bar{\theta}$, and so the most general effective action invariant under $\delta_Q$ had a super-Lagrangian (59) which could depend upon these three objects. Imposing $\delta_{Q'}$ as a spurionic symmetry, the super-Lagrangian may now only depend on $D_\theta$ and $\bar{\theta}$, but not $D_{\bar{\theta}}$. However, both $D_\theta$ and $\bar{\theta}$ have ghost number $+1$, and, because all other available superfields have ghost number-0, a ghost number-0 super-Lagrangian cannot depend on either. We conclude that the most general effective action invariant under double diffeomorphisms, the Schwinger-Keldysh symmetry, the reality condition and $\delta_{Q'}$ (in the statistical mechanical limit) takes the form

$$S = \int d^d\sigma d\theta d\bar{\theta} \sqrt{-\mathfrak{g}}\, L(\mathfrak{g}_{ij}, \mathbb{B}_k, \mathbb{W}_l; \beta^i, \Lambda_\beta) \, . \tag{116}$$

An action of this sort is not only invariant under $\delta_Q$ and $\delta_{Q'}$ but also under $\frac{\partial}{\partial\theta}$ and $\theta$.

For the second solution where $K^2 = (-1)^g$, we find that $\delta_{Q'}$, which we henceforth notate as $\delta_{\overline{Q}}$ to distinguish it from the first solution, acts on superfields as

$$\delta_{\overline{Q}} \to \frac{\partial}{\partial\bar{\theta}} + i\theta\delta_\beta \, . \tag{117}$$

The Grassmann-odd objects which anticommute with $\delta_Q$ and $\delta_{\overline{Q}}$ are just $D_\theta$ and $D_{\bar{\theta}}$, which may then be interpreted as superderivatives. Thus, the most general action invariant under all of the symmetries but KMS takes the same form as in (59), but now it cannot depend on $\bar{\theta}$:

$$S = \int d^d\sigma d\theta d\bar{\theta} \sqrt{-\mathfrak{g}}\, L(\mathfrak{g}_{ij}, \mathbb{B}_k, \mathbb{W}_l; iD_\theta, D_{\bar{\theta}}; \beta^i, \Lambda_\beta) \, . \tag{118}$$

Note that actions of the type (116) are contained in (118) upon removing the dependence of the latter on the superderivatives $D_\theta$ and $D_{\bar{\theta}}$. Therefore, we may consider both

types of symmetries in what follows.[6]

The spurionic symmetry $\delta_{\overline{Q}}$ defines a Grassman-odd operator $\overline{Q}$ which only acts on the dynamical fields, and whose action on them is given by that of $\delta_{\overline{Q}}$. We find

$$[\overline{Q}, \mathbb{X}^\mu] \equiv \left(\frac{\partial}{\partial\overline{\theta}} + i\theta\delta_\beta\right)\mathbb{X}^\mu, \qquad [\overline{Q}, \mathbb{C}] \equiv \left(\frac{\partial}{\partial\overline{\theta}} + i\theta\delta_\beta\right)\mathbb{C}, \qquad (120)$$

or equivalently,

$$[\overline{Q}, X_r^\mu] = X_g^\mu, \qquad \{\overline{Q}, X_g^\mu\} = [\overline{Q}, \widetilde{X}_a^\mu] = 0, \qquad \{\overline{Q}, X_{\overline{g}}^\mu\} = -\widetilde{X}_a^\mu,$$
$$[\overline{Q}, C_r] = C_g, \qquad \{\overline{Q}, C_g\} = [\overline{Q}, \widetilde{C}_a] = 0, \qquad \{\overline{Q}, C_{\overline{g}}\} = -\widetilde{C}_a. \qquad (121)$$

The operator $\overline{Q}$ becomes a symmetry whenever the sources become aligned (i.e. the $a$-sources vanish),

$$\left.[\overline{Q}, \mathfrak{g}_{ij}]\right|_{g_{a\,ij}=0} = \left.\left(\frac{\partial}{\partial\overline{\theta}} + i\theta\delta_\beta\right)\mathfrak{g}_{ij}\right|_{g_{a\,ij}=0}, \qquad \left.[\overline{Q}, \mathbb{B}_i]\right|_{B_{a\,i}=0} = \left.\left(\frac{\partial}{\partial\overline{\theta}} + i\theta\delta_\beta\right)\mathbb{B}_i\right|_{B_{a\,i}=0}. \qquad (122)$$

The emergent Grassman-odd spurionic symmetry $\delta_{\overline{Q}}$ was first observed in [14] and later elaborated on in [19]. An emergent Grassman-odd generator $\delta_{\overline{Q}}$ which becomes a genuine symmetry once the $\tilde{a}$-fields vanish was argued for in [15] and also [18], the latter valid only in the probe limit when the $\mathbb{X}$-fields become non-dynamical. It would be interesting to better understand the interplay between these emergent symmetries.

We end this discussion with an observation. A simple computation shows that $Q$ and $\overline{Q}$ act on the tilde'd superfields by $\frac{\partial}{\partial\theta} + i\overline{\theta}\delta_\beta$ and $\frac{\partial}{\partial\overline{\theta}}$ respectively, e.g.

$$[Q, \widetilde{\mathbb{X}}^\mu] = \left(\frac{\partial}{\partial\theta} + i\overline{\theta}\delta_\beta\right)\widetilde{\mathbb{X}}^\mu, \qquad [\overline{Q}, \widetilde{\mathbb{X}}^\mu] = \frac{\partial\widetilde{\mathbb{X}}^\mu}{\partial\overline{\theta}}. \qquad (123)$$

With these definitions one may easily verify that $\delta_Q$ and $\delta_{\overline{Q}}$ intertwine as they ought according to (114): when acting on a ghost-number-0 superfield they satisfy

$$K\delta_Q = -\delta_{\overline{Q}}K, \qquad K\delta_{\overline{Q}} = \delta_Q K. \qquad (124)$$

The spurionic symmetries $\delta_Q$ and $\delta_{\overline{Q}}$ act on tilde'd superfields as

$$\delta_Q \to \frac{\partial}{\partial\theta} + i\overline{\theta}\delta_\beta, \qquad \delta_{\overline{Q}} \to \frac{\partial}{\partial\overline{\theta}}. \qquad (125)$$

Along the lines of our analysis in Subsection 2.3, we may define other superfields from $\widetilde{\mathfrak{g}}_{ij}$ and $\widetilde{\mathbb{B}}_i$. These include an inverse super-metric $\widetilde{\mathfrak{g}}^{ij}$, a super-Christoffel connection $\widetilde{\Gamma}^i{}_{jk}$, Riemann curvature $\widetilde{\mathbb{R}}^i{}_{jkl}$, flavor field strength $\widetilde{\mathbb{G}}_{ij}$, and covariant derivative $\widetilde{\mathbb{V}}_i$.

In Eq. (118) we wrote down effective actions out of the ordinary superfields which were invariant under all of the symmetries of the problem except the KMS symmetry. Here, using the tilde'd superfields, we could also write down effective actions invariant under

---

[6] For those familiar with the Schwinger-Keldysh contour we note that the action (116) exhibits no "cross-contour" terms at tree-level: after performing the superspace integral, changing basis from $r$- and $a$-fields back to 1 and 2 fields, and setting the ghosts to vanish, this action takes the form

$$\left.S\right|_{\text{ghosts}=0} = \lim_{\hbar\to 0} \frac{1}{\hbar}\left[\int d^d\sigma \left(\sqrt{-g_1}\,L(g_{1\,ij}, B_{1\,k}, \nabla_l; \beta^i, \Lambda_\beta) - \sqrt{-g_2}\,L(g_{2\,ij}, B_{2\,k}, \nabla_l; \beta^i.\Lambda_\beta)\right)\right]. \qquad (119)$$

So an action of this sort cannot be an effective action for a dissipative fluid, which exhibits cross-contour correlations by virtue of the fluctuation-dissipation theorem.

all symmetries (including $\delta_{\overline{Q}}$) but KMS. There are two Grassmann-odd objects which anticommute with $\delta_Q$ and $\delta_{\overline{Q}}$,

$$\widetilde{D}_\theta \equiv \frac{\partial}{\partial\theta}\,, \qquad \widetilde{D}_{\bar\theta} \equiv \frac{\partial}{\partial\bar\theta} - i\theta\delta_\beta\,. \tag{126}$$

We note in passing that not only does $K$ intertwine $\delta_Q$ with $\delta_{\overline{Q}}$, but the $D$'s with the $\widetilde{D}$'s as

$$KD_\theta = -\widetilde{D}_{\bar\theta}K\,, \qquad KD_{\bar\theta} = \widetilde{D}_\theta K\,, \qquad K\widetilde{D}_\theta = -D_{\bar\theta}K\,, \qquad K\widetilde{D}_{\bar\theta} = D_\theta K\,. \tag{127}$$

In Subsection 2.3 we defined a symmetry $R$ which implements the reality condition (57) on the effective action. The tilde'd superfields are invariant under $R$, as are $i\widetilde{D}_\theta$ and $\widetilde{D}_{\bar\theta}$. Then an action of the form

$$S = \int d^d\sigma d\theta d\bar\theta \,\sqrt{-\widetilde{\mathfrak{g}}}\,\widetilde{L}(\widetilde{\mathfrak{g}}_{ij}, \widetilde{\mathbb{B}}_k, \widetilde{\mathbb{W}}_l; i\widetilde{D}_\theta, \widetilde{D}_{\bar\theta}; \beta^i, \Lambda_\beta)\,, \tag{128}$$

is invariant under all but the KMS symmetry.

### 2.5.4  Imposing the KMS symmetry

We now impose a worldvolume KMS symmetry, which is the combination of $K$ and worldvolume $\vartheta^*$. As we mentioned at the beginning of Section 2.5.2, the KMS symmetry becomes an invariance of the effective action under $K$ alone. Under $K$, the various objects that can appear in the effective action are transformed as

$$\begin{aligned}
K(\mathfrak{g}_{ij}) &= \eta_i\eta_j\widetilde{\mathfrak{g}}_{ij}\,, & K(\mathbb{B}_i) &= \eta_i\widetilde{\mathbb{B}}_i\,, & K(\beta^i) &= -\eta_i\beta^i\,, & K(\Lambda_\beta) &= -\Lambda_\beta\,, \\
K(\mathbb{W}_i) &= \eta_i\widetilde{\mathbb{W}}_i\,, & K(D_\theta) &= -\widetilde{D}_{\bar\theta}\,, & K(D_{\bar\theta}) &= \widetilde{D}_\theta\,.
\end{aligned} \tag{129}$$

Here $\widetilde{\mathfrak{g}}_{ij}$ and $\widetilde{\mathbb{B}}_i$ are defined in (105), $\eta_i$ is the worldvolume CPT eigenvalue associated with derivatives (e.g., in Minkowski space in Cartesian coordinates, we have $\eta_0 = -1$, $\eta_1 = -1$ and the remaining components unity), $\widetilde{\mathbb{W}}_i$ is the covariant derivative whose connection is associated with $\widetilde{\mathfrak{g}}_{ij}$, and the tilde'd superderivatives are given in (126).

Acting with $K$ on an effective action (118) built from the ordinary superfields gives

$$\begin{aligned}
\int d^d\sigma d\theta d\bar\theta\, &\sqrt{-\mathfrak{g}}\, L(\mathfrak{g}_{ij}, \mathbb{B}_k, \mathbb{W}_l; iD_\theta, D_{\bar\theta}; \beta^i, \Lambda_\beta) \\
&\to \int d^d\sigma d\theta d\bar\theta\, \sqrt{-\widetilde{\mathfrak{g}}}\, \widetilde{L}(\widetilde{\mathfrak{g}}_{ij}, \widetilde{\mathbb{B}}_k, \widetilde{\mathbb{W}}_l; i\widetilde{D}_{\bar\theta}, \widetilde{D}_\theta; \beta^i, \Lambda_\beta)\,,
\end{aligned} \tag{130}$$

where $\widetilde{L}$ is determined by $L$ as

$$\widetilde{L}(\widetilde{\mathfrak{g}}_{ij}, \widetilde{\mathbb{B}}_k, \widetilde{\mathbb{W}}_l; i\widetilde{D}_{\bar\theta}, \widetilde{D}_\theta; \beta^i, \Lambda_\beta) = L(\eta_i\eta_j\widetilde{\mathfrak{g}}_{ij}, \eta_k\widetilde{\mathbb{B}}_k, \eta_l\widetilde{\mathbb{W}}_l - i\widetilde{D}_{\bar\theta}, \widetilde{D}_\theta; -\eta_i\beta^i, -\Lambda_\beta)\,. \tag{131}$$

Note that the KMS transformation maps an action of ordinary superfields (118) to one with tilde'd superfields (128).

We then see that KMS conjugation acts on the action by the combination of three operations: exchange superfields by their tilde'd versions (or, equivalently, superpullbacks by their tilde'd version), exchange the superderivatives $D_\theta$ and $D_{\bar\theta}$ with $-\widetilde{D}_{\bar\theta}$ and $\widetilde{D}_\theta$, and multiply the various fields by their CPT eigenvalue. It is then clear how to render the effective action invariant under worldvolume KMS. Given any action constructed from ordinary superfields as in (118), we add it to its image under $K$, which we call its KMS

partner term. In an equation, effective actions invariant under all symmetries take the form:[7]

$$
\begin{aligned}
S_{eff} = \int d^d\sigma d\theta d\bar{\theta} \Big\{ & \sqrt{-\mathfrak{g}}\, L(\mathfrak{g}_{ij}, \mathbb{B}_k, \pmb{\mathbb{W}}_l; iD_\theta, D_{\bar{\theta}}; \beta^i, \Lambda_\beta) \\
& + \sqrt{-\widetilde{\mathfrak{g}}}\, \widetilde{L}(\widetilde{\mathfrak{g}}_{ij}, \widetilde{\mathbb{B}}_k, \widetilde{\pmb{\mathbb{W}}}_l; i\widetilde{D}_{\bar{\theta}}, \widetilde{D}_\theta; \beta^i, \Lambda_\beta) \Big\},
\end{aligned}
\tag{132}
$$

where $\widetilde{L}$ was defined in (131).

Eq. (132) is the main result of this Section. It describes actions which, in the statistical mechanical limit, are invariant under the doubled symmetries, the reality condition, the Schwinger-Keldysh symmetry and a $\mathbb{Z}_2$ worldvolume KMS symmetry. (Our result is ultimately identical to that of [19,21], as we demonstrate in Appendix C.) However, we have not yet argued that a worldvolume KMS symmetry implies the target KMS symmetry (80) that we sought to impose. We conclude this Section with such an argument.

In the classical limit, the equations of motion of our effective theory are solved by some profile for the bosonic fields,

$$
X_r^\mu = X_{rc}^\mu, \qquad X_a^\mu = X_{ac}^\mu, \qquad C_r = C_{rc}, \qquad C_a = C_{ac},
\tag{133}
$$

and setting the ghosts to vanish. Plugging this solution back into the effective action (132) gives the tree-level approximation to the generating functional $W$. We use the classical solution $(X_{rc}^\mu(\sigma), C_{rc}(\sigma))$ to push forward the worldvolume $\delta_\beta$ to a target space transformation $\delta_{b'}$, and the transformation of the target space sources under $K$, to a target space CPT transformation $\Theta$. Then the worldvolume KMS symmetry implies

$$
W_{\text{tree}}[A_r, A_a; \delta_{b'}] = W_{\text{tree}}[\eta_A \Theta^* A_r, \eta_A \Theta^* \widetilde{A}_a; \delta_{b'}^{\text{CPT}}],
\tag{134}
$$

where $A_r(x) = \lim_{\hbar\to 0} \frac{1}{2}(A_1(x) + A_2(x))$ and $A_a(x) = \lim_{\hbar\to 0} \frac{A_1(x) - A_2(x)}{\hbar}$ represents all target sources. But this is nothing more than the covariant KMS symmetry (80) for the particular transformation $\delta_{b'}$ and CPT transformation $\Theta$, in the statistical mechanical limit.

This argument can be generalized to account for loop contributions to $W$. Expanding $S_{eff}$ around the classical solution and formally treating the coefficients of its non-Gaussian part as small parameters, one may in principle construct a loop expansion for a 1PI effective action $S_{1\text{PI}}$ for the $X$- and $C$-supermultiplets. Barring an anomaly, this 1PI action will also be invariant under the same symmetries of the effective action, including the $\mathbb{Z}_2$ worldvolume KMS symmetry. Recall that to go from $S_{1\text{PI}}$ to the loop-approximation to $W$, $W_{\text{loop}}$, one solves the equations of motion that follow from variation of $S_{1\text{PI}}$, and then plugs the solution back into $S_{1\text{PI}}$. Using this solution, we pushforward the worldvolume KMS symmetry to the target space, as we have done for the tree level approximation. Thus, $W_{\text{loop}}$ is also invariant under the target space KMS symmetry (80).

## 3 Summary and the relation to hydrodynamics

Let us summarize our findings so far. In the statistical mechanical limit we are working in, we assume that there exists a coordinate system where the external metric and flavor

---

[7]We caution the reader that $\delta_Q$ and $\delta_{\overline{Q}}$ act on the first term in a different way than on the second. On the first, $\delta_Q$ acts as $\frac{\partial}{\partial\theta}$ and on the second as $\frac{\partial}{\partial\theta} + i\bar{\theta}\delta_\beta$. Nevertheless the total action is invariant under $\delta_Q$ and $\delta_{\overline{Q}}$, since both terms in $S_{eff}$ are separately invariant.

fields are almost aligned,

$$g_{1\,\mu\nu}(x) = g_{r\,\mu\nu}(x) + \frac{\hbar}{2}g_{a\,\mu\nu}(x) + O(\hbar^2)\,, \qquad g_{2\,\mu\nu}(x) = g_{r\,\mu\nu}(x) - \frac{\hbar}{2}g_{a\,\mu\nu}(x) + O(\hbar^2)\,.$$

$$B_{1\,\mu}(x) = B_{r\,\mu}(x) + \frac{\hbar}{2}B_{a\,\mu}(x) + O(\hbar^2)\,, \qquad B_{2\,\mu}(x) = B_{r\,\mu}(x) - \frac{\hbar}{2}B_{a\,\mu}(x) + O(\hbar^2)\,.$$

$$(135)$$

In this limit, the Schwinger-Keldysh effective action takes the form

$$S_{eff} = \int d^d\sigma d\theta d\bar\theta \Big\{ \sqrt{-\mathfrak{g}}\, L(\mathfrak{g}_{ij}, \mathbb{B}_k, \mathbb{W}_l; iD_\theta, D_{\bar\theta}; \beta^i, \Lambda_\beta)$$
$$+ \sqrt{-\widetilde{\mathfrak{g}}}\, \widetilde{L}(\widetilde{\mathfrak{g}}_{ij}, \widetilde{\mathbb{B}}_k, \widetilde{\mathbb{W}}_l; i\widetilde{D}_{\bar\theta}, \widetilde{D}_\theta; \beta^i, \Lambda_\beta) \Big\}\,,$$

$$(136)$$

where the various terms are defined below.

The superfields $\mathfrak{g}_{ij}$ and $\mathbb{B}_i$ are referred to as superpullbacks and are given by

$$\mathfrak{g}_{ij} = g_{r\,\mathrm{ij}}(\mathbb{X}) + \bar\theta\theta g_{a\,ij}(\mathbb{X})\,, \qquad \mathbb{B}_i = B_{r\,i}(\mathbb{X},\, \mathbb{C}) + \bar\theta\theta B_{a\,i}(\mathbb{X})\,, \qquad (137)$$

with

$$g_{r\,ij}(X) = g_{r\,\mu\nu}(X)\partial_i X^\mu \partial_j X^\nu\,, \qquad g_{a\,ij}(X) = g_{a\,\mu\nu}(X)\partial_i X^\mu \partial_j X^\nu\,,$$
$$B_{r\,i}(X,\, C) = B_{r\,\mu}(X)\partial_i X^\mu + \partial_i C\,, \qquad B_{a\,i}(X) = B_{a\,\mu}(X)\partial_i X^\mu\,,$$

$$(138)$$

and

$$\mathbb{X}^\mu = X_r^\mu + \theta X_{\bar g}^\mu + \bar\theta X_g^\mu + \bar\theta\theta X_a^\mu\,, \qquad \mathbb{C} = C_r + \theta C_{\bar g} + \bar\theta C_g + \bar\theta\theta C_a\,, \qquad (139)$$

where $\theta$ and $\bar\theta$ are Grassmann-odd coordinates. The operator $\mathbb{W}_i$ is the covariant derivative taken using the Christoffel connection associated with $\mathfrak{g}_{ij}$. The superderivatives $D_\theta$ and $D_{\bar\theta}$ are given by

$$D_\theta = \frac{\partial}{\partial\theta} - i\bar\theta\delta_\beta\,, \qquad D_{\bar\theta} = \frac{\partial}{\partial\bar\theta}\,,$$

and $\beta^i$ and $\Lambda_\beta$ are external parameters associated with the thermal state of the system in the infinite past.

The tilde'd Lagrangian $\widetilde{L}$ is defined as

$$\widetilde{L}(\widetilde{\mathfrak{g}}_{ij}, \widetilde{\mathbb{B}}_k, \widetilde{\mathbb{W}}_l; iD_{\bar\theta}, \widetilde{D}_\theta; \beta^i, \Lambda_\beta) = L(\eta_i\eta_j\widetilde{\mathfrak{g}}_{ij}, \eta_k\widetilde{\mathbb{B}}_k, \eta_i\widetilde{\mathbb{W}}_i; -i\widetilde{D}_{\bar\theta}, \widetilde{D}_\theta; -\eta_i\beta^i, -\Lambda_\beta)\,, \qquad (140)$$

with tilde'd fields defined as follows. The super-pullbacks are given by

$$\widetilde{\mathfrak{g}}_{ij} = g_{r\,\mathrm{ij}}(\mathbb{X}) + \bar\theta\theta \left(g_{a\,ij}(\mathbb{X}) + i\delta_\beta g_{r\,ij}(X_r)\right)\,,$$
$$\widetilde{\mathbb{B}}_i = B_{r\,i}(\mathbb{X}) + \bar\theta\theta \left(B_{a\,i}(\mathbb{X}) + i\delta_\beta B_{r\,i}(X_r)\right) + \partial_i\mathbb{C} + \bar\theta\theta i\delta_\beta \partial_i C_r\,.$$

$$(141)$$

Tilde'd covariant derivatives $\widetilde{\mathbb{W}}_i$ are taken using the Christoffel connection generated by $\widetilde{\mathfrak{g}}$. Tilde'd superderivatives are given by

$$\widetilde{D}_\theta = \frac{\partial}{\partial\theta}\,, \qquad \widetilde{D}_{\bar\theta} = \frac{\partial}{\partial\bar\theta} - i\theta\delta_\beta\,.$$

The $\eta$'s correspond to CPT eigenvalues of the various terms. In Minkowski space they are given by

$$\eta_0 = \eta_1 = -1\,, \qquad (142)$$

with the remaining eigenvalues equal to one.

The tilde'd super-Lagrangian is simple to construct in practice. The super-Lagrangian $L$ is a worldvolume scalar, which depends on scalar superfields which may be decomposed in a basis which are either even or odd under CPT. Let $\mathbb{F}^{\pm}$ denote such a basis of superfields, where the superscript indicates the CPT-eigenvalue. Given a super-Lagrangian $L(\mathbb{F}^{+}, \mathbb{F}^{-})$, the KMS partner super-Lagrangian is simply $L(\widetilde{\mathbb{F}}^{+}, -\widetilde{\mathbb{F}}^{-})$. It is this last form that will be most useful to us when constructing actions for fluids.

The Lagrangian $L$ must also satisfy the following symmetries. It must be a scalar under worldvolume diffeomorphisms under which $\mathfrak{g}_{ij}$, $\mathbb{B}_i$, $\mathbb{W}_i$ and $\beta^i$ transform as tensors and $\Lambda_\beta$, $D_\theta$ and $D_{\bar\theta}$ transform as scalars. It must also be invariant under worldvolume gauge transformations under which $\mathbb{C}$ transforms as a phase, $\mathbb{C} \to \mathbb{C} + \Lambda$, and $\Lambda_\beta$ transforms as $\Lambda_\beta \to \Lambda_\beta - \beta^i \partial_i \Lambda$. The Lagrangian $L$ must be a real function of its arguments. We also impose an additive ghost number symmetry. Under it, we assign $(\delta_Q, X_{\bar g}^\mu, C_{\bar g}, \bar\theta, D_\theta)$ ghost number $+1$ and $(\delta_{\overline{Q}}, X_g^\mu, C_g, \theta, D_{\bar\theta})$ ghost number $-1$. So defined, the superfields $(\mathbb{X}^\mu, \mathbb{C}, \mathfrak{g}_{ij}, \mathbb{B}_k, \mathbb{W}_l, \beta^m, \Lambda_\beta)$ are all ghost number-0, and we demand that $L$ is ghost number-0.

By design, the effective action (136) is invariant under a Grassmann-odd symmetry $\delta_Q$, which enforces the Schwinger-Keldysh symmetry $Z[A, A] = 1$. It is also invariant under a worldvolume KMS symmetry, which exchanges $L$ with $\widetilde{L}$ in the action. Together, invariance under $\delta_Q$ and KMS, mandate a second Grassmann-odd symmetry $\delta_{\overline{Q}}$. For the interested reader, the action of $\delta_Q$, $\delta_{\overline{Q}}$, and the worldvolume KMS symmetry is summarized in Subsection 2.5.3.

Collectively noting the superfields which may be constructed from $\mathfrak{g}_{ij}$ and $\mathbb{B}_k$ by $\mathbb{F}_A$, with $A$ a collective index, we expand the super-Lagrangian $L$ as

$$L = \frac{1}{2}L_0 + \frac{1}{2}\sum_{n=0} i^{n+1} L^{ABC_1\ldots C_n} D_\theta \mathbb{F}_A D_{\bar\theta} \mathbb{F}_B D\mathbb{F}_{C_1} \ldots D\mathbb{F}_{C_n} + L_{\text{ghost}}, \qquad (143)$$

where the $L^{ABC_1\cdots}$ and $L'^{ABC_1\cdots}$ are in general super-differential operators constructed from $(\mathbb{F}, \mathbb{W}_i; \beta^j, \Lambda_\beta)$, and we have defined

$$D = D_\theta D_{\bar\theta}. \qquad (144)$$

The terms $L_{\text{ghost}}$ are those which vanish identically when setting the ghosts to vanish, e.g., $D_\theta \mathbb{F}_A D_{\bar\theta} \mathbb{F}_B D_\theta \mathbb{F}_C D_{\bar\theta} \mathbb{F}_D$. We call $L_0$ a scalar term, and refer to other parts of $L$ as tensor terms. We will often refer to tensor terms with $n$ powers of $D\mathbb{F}$ (or $n$ powers of $D'\mathbb{F}$) as $n+2$ order tensor terms.

For convenience let us write the KMS conjugate of the Lagrangian explicitly,

$$\widetilde{L} = \frac{1}{2}\widetilde{L}_0 + \frac{1}{2}\sum_{n=0} (-i)^{n+1} \eta_{ABC_1\ldots C_n} \widetilde{L}^{ABC_1\ldots C_n} \widetilde{D}_{\bar\theta}\widetilde{\mathbb{F}}_A \widetilde{D}_\theta\widetilde{\mathbb{F}}_B \widetilde{D}\widetilde{\mathbb{F}}_{C_1} \ldots \widetilde{D}\widetilde{\mathbb{F}}_{C_n} + \widetilde{L}_{\text{ghost}}, \quad (145)$$

where $\eta_{ABC_1\ldots C_n} = \eta_A \eta_B \eta_{C_1} \ldots \eta_{C_n}$ and $\eta_A$ is the CPT eigenvalue of $\mathbb{F}_A$ and we have defined

$$\widetilde{D} = \widetilde{D}_{\bar\theta}\widetilde{D}_\theta. \qquad (146)$$

The tilde'd components of the Lagrangian are defined as in (140).

In the remainder of this manuscript we will extract the hydrodynamic constitutive relations from effective actions of the form (136). By constitutive relations, we mean the tree-level expressions for the stress tensor $T_r^{\mu\nu}$ and flavor current $J_r^\mu$ upon setting the $a$-sources to vanish. In the absence of $a$-sources, we may consistently take the ghosts and dynamical $a$-fields to vanish, so that the only remaining dynamical fields are $X_r^\mu$ and $C_r$. As we discussed in [18], we obtain these constitutive relations as follows. We first vary

the effective action with respect to $a$-type sources, and then set the ghosts and $a$-fields to vanish. This defines a worldvolume stress tensor and flavor current via

$$T_r^{ij}(\sigma) = \frac{2}{\sqrt{-g_r(\sigma)}} \frac{\delta S_{eff}}{\delta g_{a\,ij}(\sigma)}\bigg|_{a=\text{ghosts}=0} \,, \qquad J_r^i(\sigma) = \frac{1}{\sqrt{-g_r(\sigma)}} \frac{\delta S_{eff}}{\delta B_{a\,i}(\sigma)}\bigg|_{a=\text{ghosts}=0} \,. \quad (147)$$

The constitutive relations are then obtained by pushing forward the worldvolume stress tensor and current using $X_r^\mu(\sigma)$, e.g.,

$$T^{\mu\nu}(x) = T_r^{ij}(\sigma(x))\partial_i X_r^\mu(\sigma(x))\partial_j X_r^\nu(\sigma(x)) \,. \quad (148)$$

The remaining equations of motion for $X_r^\mu(\sigma)$ and $C_r(\sigma)$ are exactly the conservation equations for $T^{\mu\nu}$ and $J^\mu$,

$$\frac{\delta S_{eff}}{\delta X_a^\mu}\bigg|_{a=\text{ghosts}=0} = -(\nabla^\nu T_{\mu\nu} - G_{\mu\nu}J^\nu) = 0 \,, \qquad \frac{\delta S_{eff}}{\delta C_a}\bigg|_{a=\text{ghosts}=0} = -\nabla_\mu J^\mu = 0 \,. \quad (149)$$

Here $\nabla_\mu$ is the covariant derivative associated with the metric $g_{\mu\nu}(x) = g_{r\,\mu\nu}(x)$ and $G_{\mu\nu}$ the field strength of $B_\mu(x) = B_{r\,\mu}(x)$. In practice, the physical stress tensor $T^{\mu\nu}$ and $J^\mu$ are given by the worldvolume stress tensor $T_r^{ij}$ and current $J_r^i$ upon replacing the worldvolume indices with target space ones.

Before closing this Section we note that one often computes the constitutive relations in a derivative expansion. To this end, we consistently assign scalings whereby $\mathfrak{g}_{ij}$, $\mathbb{B}_k$, $\beta^i$ and $\Lambda_\beta$ are zeroth order in derivatives, $\mathbb{W}_i$ is first order in derivatives, and $D_\theta$ and $D_{\bar\theta}$ are order one half in derivatives. With this scaling in mind, the expansion (143) (and its KMS conjugate) should be truncated at order $n$ if we are interested in the constitutive relations to order $n + 1$

## 4 A simple example: the ideal fluid

In this Section we work out the effective action and constitutive relations for the simplest possible example, that of ideal hydrodynamics. To leading order in derivatives, the super-Lagrangian $L$ appearing in the effective action $S_{eff}$ in (136) is merely

$$L = G(\mathbb{T}, \nu) \,, \quad (150)$$

where $\mathbb{T}$ and $\nu$ are the only zeroth order diffeomorphism and $U(1)$ invariant scalars available,

$$\mathbb{T} = \frac{1}{\sqrt{-\mathfrak{g}_{ij}\beta^i\beta^j}} \,, \qquad \nu = \beta^i\mathbb{B}_i + \Lambda_\beta \,. \quad (151)$$

They are respectively even and odd under CPT.

According to (136) the total effective action is

$$S_{eff} = \int d^d\sigma d\theta d\bar\theta \left( \sqrt{-\mathfrak{g}}\, G(\mathbb{T}, \nu) + \sqrt{-\widetilde{\mathfrak{g}}}\, G(\widetilde{\mathbb{T}}, -\widetilde{\nu}) \right) \,, \quad (152)$$

with

$$\widetilde{\mathbb{T}} = \frac{1}{\sqrt{-\beta^i\beta^j\widetilde{\mathfrak{g}}_{ij}}} \,, \qquad \widetilde{\nu} = \beta^i\widetilde{\mathbb{B}}_i + \Lambda_\beta \,. \quad (153)$$

Using that for a general superfield $\mathbb{F}$,

$$\widetilde{\mathbb{F}}(\mathbb{X}) = \mathbb{F}(\mathbb{X}) + \bar\theta\theta i\delta_\beta F_r(\mathbb{X}) \,, \quad (154)$$

the effective action can be written more simply as

$$S_{eff} = \int d^d\sigma d\theta d\bar{\theta}\sqrt{-\mathfrak{g}}\, P(\mathbb{T}, \nu)\,, \tag{155}$$

with

$$P(\mathbb{T}, \nu) = G(\mathbb{T}, \nu) + G(\mathbb{T}, -\nu)\,. \tag{156}$$

To efficiently compute $T_r^{ij}(\sigma)$ and $J_r^i(\sigma)$, we define a worldvolume stress tensor superfield, and a worldvolume current superfield by

$$\mathbb{T}^{ij} = \frac{2}{\sqrt{-\mathfrak{g}}}\frac{\delta S_{eff}}{\delta \mathfrak{g}_{ij}}\,, \qquad \mathbb{J}^i = \frac{1}{\sqrt{-\mathfrak{g}}}\frac{\delta S_{eff}}{\delta \mathbb{B}_i}\,. \tag{157}$$

Upon setting the ghosts and $a$-fields to vanish, these superfields become $T_r^{ij}$ and $J_r^i$,

$$\mathbb{T}^{ij}|_{a=\text{ghosts}=0} = T_r^{ij}\,, \qquad \mathbb{J}^i|_{a=\text{ghosts}=0} = J_r^i\,. \tag{158}$$

We easily compute the super-stress tensor and current (157) to be

$$\mathbb{T}^{ij} = \mathbb{T}^3\frac{\partial P}{\partial \mathbb{T}}\beta^i\beta^j + P\mathfrak{g}^{ij}\,, \qquad \mathbb{J}^i = \frac{\partial P}{\partial \nu}\beta^i\,. \tag{159}$$

Setting the ghosts and $a$-fields to vanish, we define

$$T \equiv \mathbb{T}|_{a=\text{ghosts}=0}\,, \qquad \nu \equiv \nu|_{a=\text{ghosts}=0}\,, \tag{160a}$$

and a normalized velocity $u^\mu$ via

$$u^\mu = T\beta^i\partial_i X_r^\mu\,. \tag{160b}$$

In terms of these, we find that the constitutive relations that follow from the action (152) are

$$\begin{aligned}
T^{\mu\nu} &= \left(-P + T\frac{\partial P}{\partial T}\right)u^\mu u^\nu + P(g^{\mu\nu} + u^\mu u^\nu)\,, \\
J^\mu &= \frac{1}{T}\frac{\partial P}{\partial \nu}u^\mu\,,
\end{aligned} \tag{161}$$

with $P = P(T, \nu)$. These are exactly the constitutive relations of an ideal fluid,

$$\begin{aligned}
T^{\mu\nu} &= \epsilon u^\mu u^\nu + P\left(g^{\mu\nu} + u^\mu u^\nu\right)\,, \\
J^\mu &= \rho u^\mu\,,
\end{aligned} \tag{162}$$

with pressure $P(T, \nu)$, local temperature $T$, velocity $u^\mu$, reduced chemical potential $\nu = \frac{\mu}{T}$ and $\epsilon$ the energy density and $\rho$ the charge density which are related to the pressure via

$$\epsilon = -P + T\left(\frac{\partial P}{\partial T}\right)_\nu\,, \qquad \rho = \frac{1}{T}\left(\frac{\partial P}{\partial \nu}\right)_T\,. \tag{163}$$

Thus, our effective action (152) describes an ideal fluid as advertised.

Following this example, in the remainder of this work we identify the local temperature $T$, reduced chemical potential $\nu$, and velocity $u^\mu$ according to (160). We regard $\mathbb{T}$ as the super-temperature, $\nu$ as the super (reduced) chemical potential and $\beta^i$ as the (unnormalized) velocity field.

## 5 The entropy current

One of the most interesting aspects of relativistic hydrodynamics is the stipulation of the existence of an entropy current $S^\mu$ whose leading order term in a derivative expansion is

$$S^\mu = su^\mu + O(\partial)\,, \tag{164}$$

with $s$ the entropy density, and such that

$$\nabla_\mu S^\mu \geq 0\,. \tag{165}$$

Recently in [25] and later in [23, 24], it was shown how to obtain the hydrodynamic entropy current from the Schwinger-Keldysh effective action. In particular, in [23] we have provided an algorithm for defining the entropy current by coupling it to an external source $\mathbb{A}_I$, which resembles a dynamical $U(1)_T$ field postulated in [26, 27]. In [23] we have applied our construction to a probe limit of the Schwinger-Keldysh effective action, valid to all orders in $\hbar$ but where charge was free to move in a fixed thermally equilibrated background. In what follows we briefly summarize the construction of [23] and adapt it to the statistical mechanical limit.

Consider first the action of a scalar field $\phi$,

$$S = \int d^d x \sqrt{-g}\, L(\phi;\, g_{\mu\nu})\,, \tag{166}$$

which depends on an external metric $g_{\mu\nu}$. The variation of the action with respect to $\phi$ and $g_{\mu\nu}$,

$$\delta S = \int d^d x \sqrt{-g}\left(E_\phi \delta\phi + \frac{1}{2} T^{\mu\nu}\delta g_{\mu\nu}\right)\,, \tag{167}$$

defines the stress tensor $T^{\mu\nu}$ and the equation of motion $E_\phi$. Consider the particular variation $\delta_\beta$, which is generated by an infinitesimal coordinate transformation $x^\mu \to x^\mu + \beta^\mu$, under which $\phi$ and $g_{\mu\nu}$ vary by a Lie derivative along $\beta^\mu$,

$$\delta_\beta \phi = \mathcal{L}_\beta \phi\,, \qquad \delta_\beta g_{\mu\nu} = \mathcal{L}_\beta g_{\mu\nu}\,. \tag{168}$$

In general, $\delta_\beta$ is not a symmetry of the action in the sense that $\delta_\beta S$ does not necessarily vanish on-shell. However, we can impose invariance of the action under a suitably "gauged" version of $\delta_\beta$ once we incorporate an appropriate connection.

Consider the transformation

$$\delta_T \phi = \Lambda_T \delta_\beta \phi\,, \qquad \delta_T g_{\mu\nu} = \Lambda_T \delta_\beta g_{\mu\nu}\,, \tag{169}$$

with a spacetime dependent parameter $\Lambda_T$. In general we will refer to transformations $\delta_T$ of a quantity $F$ as homogeneous if $\delta_T$ acts on $F$ as $\delta_T F = \Lambda_T \delta_\beta F$. Clearly, derivatives of $\phi$ and $g_{\mu\nu}$ will not transform homogeneously under $\delta_T$. However, introducing a connection $A_\mu$ and modifying the partial derivative as

$$\partial_\mu \to \partial_\mu^{(A)} = \partial_\mu + A_\mu \delta_\beta\,, \tag{170}$$

then $\partial_\mu^{(A)}\phi$ and $\partial_\mu^{(A)} g_{\nu\rho}$ transform homogeneously under $\delta_T$ provided that $A_\mu$ varies as

$$\delta_T A_\mu = \Lambda_T \delta_\beta A_\mu - A_\mu \delta_\beta \Lambda_T - \partial_\mu \Lambda_T\,. \tag{171}$$

Upon replacing $\partial_\mu \to \partial_\mu^{(A)}$ the Christoffel connection is modified as

$$\Gamma^{(A)\,\mu}{}_{\nu\rho} = \frac{1}{2} g^{\mu\sigma}\left(\partial_\nu^{(A)} g_{\rho\sigma} + \partial_\rho^{(A)} g_{\nu\sigma} - \partial_\sigma^{(A)} g_{\nu\rho}\right)\,, \tag{172}$$

which leads to a modified covariant derivative $\nabla^{(A)}_\mu$. It acts on, e.g., the metric as

$$\nabla^{(A)}_\mu g_{\nu\rho} = \partial^{(A)}_\mu g_{\nu\rho} - \Gamma^{(A)\,\sigma}{}_{\nu\mu} g_{\sigma\rho} - \Gamma^{(A)\,\sigma}{}_{\rho\mu} g_{\nu\sigma} = 0 \,. \tag{173}$$

After replacing $\partial_\mu \to \partial^{(A)}_\mu$ everywhere, the minimally coupled Lagrangian $L^{(A)}$ transforms homogeneously under $\delta_T$,

$$\delta_T L^{(A)} = \Lambda_T \delta_\beta L^{(A)} \,. \tag{174}$$

In order to make the action invariant under $\delta_T$ we note that

$$\delta_T \sqrt{-g} = \frac{1}{2} g^{\mu\nu} \Lambda_T \delta_\beta g_{\mu\nu} = \Lambda_T \partial_\mu(\sqrt{-g}\beta^\mu) \,, \tag{175}$$

and

$$\delta_T \left( \frac{1}{\beta^\mu A_\mu + 1} \right) = \delta_\beta \left( \frac{\Lambda_T}{\beta^\mu A_\mu + 1} \right) \,. \tag{176}$$

Thus, the modified action

$$S^{(A)} = \int \frac{d^d x \sqrt{-g}}{\beta^\mu A_\mu + 1} L^{(A)} \,, \tag{177}$$

is invariant under $\delta_T$,

$$\delta_T S^{(A)} = \int d^d x \, \partial_\nu \left( \frac{\sqrt{-g}\beta^\nu}{\beta^\mu A_\mu + 1} L^{(A)} \right) = 0 \,, \tag{178}$$

on a manifold without a boundary.

We note in passing that one can characterize the transformation properties of fields or sources, $F$, under $\delta_T$ by assigning them an additive "charge" $n$. A field $F^{(n)}$ with charge $n$ varies under $\delta_T$ as

$$\delta_T F^{(n)} = \Lambda_T \delta_\beta F^{(n)} - n F^{(n)} \delta_\beta \Lambda_T \,. \tag{179}$$

Clearly $F^{(n)} F^{(m)}$ will have charge $n+m$, and fields which transform homogeneously have charge 0. Using this nomenclature, we construct the Lagrangian $L^{(A)}$ so it has charge 0. Note that a field of charge $-1$ varies as a Lie derivative,

$$\delta_T F^{(-1)} = \Lambda_T \delta_\beta F^{(-1)} + F^{(-1)} \delta_\beta \Lambda_T = \delta_\beta(\Lambda_T F^{(-1)}) \,, \tag{180}$$

and so its integral $\int d^d x \sqrt{-g}\, F^{(-1)}$ is invariant. The object $\beta^\mu A_\mu + 1$ has charge $+1$, so that $\frac{L^{(A)}}{\beta^\mu A_\mu + 1}$ is just such a charge $-1$ object.

We can now define a current $S^\mu$ which couples to $A_\mu$ via

$$\delta S^{(A)} = \int d^d x \frac{\sqrt{-g}}{1 + \beta^\mu A_\mu} \left( E_\phi \delta\phi + \frac{1}{2} T^{\mu\nu} \delta g_{\mu\nu} - S^\mu \delta A_\mu \right) \,. \tag{181}$$

The invariance of $S^{(A)}$ under $\delta_T$ implies the off-shell relation

$$\nabla_\mu S^\mu \Big|_{A=0} = \frac{1}{2} T^{\mu\nu} \delta_\beta g_{\mu\nu} + E_\phi \delta_\beta \phi \,. \tag{182}$$

Indeed, if $\beta^\mu$ is a Killing vector then $S^\mu$ is the expected conserved current $S^\mu = T^{\mu\nu}\beta_\nu$.

The above construction may be adapted to the Schwinger-Keldysh effective action (136) where $\delta_\beta$ is the transformation generated by $\beta^i$ and the flavor transformation $\Lambda_\beta$. To wit, $\delta_\beta$ acts on the dynamical fields via

$$\delta_\beta \mathbb{X}^\mu = \beta^i \partial_i \mathbb{X}^\mu \,, \qquad \delta_\beta \mathbb{C} = \beta^i \partial_i \mathbb{C} + \Lambda_\beta \,, \tag{183}$$

so that (169) takes the form

$$\delta_T \mathbb{X}^\mu = \mathbb{A}_T \delta_\beta \mathbb{X}^\mu \,, \qquad \delta_T \mathbb{C} = \mathbb{A}_T \delta_\beta \mathbb{C} \,, \tag{184}$$

where $\Lambda_T$ has been upgraded to a superfield $\mathbb{A}_T$. Minimally coupling to an external field $\mathbb{A}_i$,

$$\partial_i \to \partial_i^{(A)} = \partial_i + \mathbb{A}_i \delta_\beta \,, \tag{185}$$

so that, e.g.

$$\partial_i \mathbb{X}^\mu \to \partial_i^{(A)} \mathbb{X}^\mu = (\partial_i + \mathbb{A}_i \delta_\beta) \mathbb{X}^\mu \,, \qquad \partial_i \mathbb{C} \to \partial_i^{(A)} \mathbb{C} = (\partial_i + \mathbb{A}_i \delta_\beta) \mathbb{C} \,, \tag{186}$$

and defining $\mathbb{A}_i$ to vary under $\delta_T$ as

$$\delta_T \mathbb{A}_i = \mathbb{A}_T \delta_\beta \mathbb{A}_i - \mathbb{A}_i \delta_\beta \mathbb{A}_T - \partial_i \mathbb{A}_T \,, \tag{187}$$

then derivatives of $\mathbb{X}^\mu$ and $\mathbb{C}$ transform homogeneously under $\delta_T$. Our analysis differs from that in the toy model of a scalar field $\phi$ in that the target space sources $g_{s\,\mu\nu}(x)$ and $B_{s\,\mu}(x)$ are inert under $\delta_\beta$ and therefore also under $\delta_T$. The transformation properties of, say, $g_{r\,ij}(\mathbb{X})$ under $\delta_T$ are solely due to the dependence on $\mathbb{X}^\mu$.

The transformation rules (186) imply that the fields $\mathbb{F}$ in the Lagrangian (143) should be replaced by their counterparts $\mathbb{F}^{(A)}$ where

$$\mathfrak{g}_{ij}^{(A)} = \mathfrak{g}_{kl} \left( \delta_i^k + \beta^k \mathbb{A}_i \right) \left( \delta_j^l + \beta^l \mathbb{A}_j \right) \,, \tag{188}$$

and

$$\mathbb{B}_i^{(A)} = \mathbb{B}_k \left( \delta_i^k + \beta^k \mathbb{A}_i \right) \,, \tag{189}$$

which transform homogeneously under $\delta_T$

$$\delta_T \mathfrak{g}_{ij}^{(A)} = \mathbb{A}_T \delta_\beta \mathfrak{g}_{ij}^{(A)} \,, \qquad \delta_T \mathbb{B}_i^{(A)} = \mathbb{A}_T \delta_\beta \mathbb{B}_i^{(A)} \,. \tag{190}$$

The appropriately modified covariant derivatives $\mathbb{W}^{(A)} = \partial^{(A)} + \mathbb{\Gamma}^{(A)}$ also transform homogeneously, where the connection is

$$\mathbb{\Gamma}^{(A)\,i}{}_{jk} = \frac{1}{2} \mathfrak{g}^{(A)\,im} \left( \partial_j^{(A)} \mathfrak{g}_{mk}^{(A)} + \partial_k^{(A)} \mathfrak{g}_{jm}^{(A)} - \partial_m^{(A)} \mathfrak{g}_{jk}^{(A)} \right) \,. \tag{191}$$

Recall that the Lagrangian (143) contains not only worldvolume derivatives but also superspace derivatives. In order for the superspace derivatives to transform homogeneously under $\delta_T$ we must upgrade $\mathbb{A}_i$ to a super-connection in superspace. That is, we need to introduce $\mathbb{A}_\theta$ and $\mathbb{A}_{\bar\theta}$ components such that

$$D_\theta \to D_\theta + \mathbb{A}_\theta \delta_\beta \,, \qquad D_{\bar\theta} \to D_{\bar\theta} + \mathbb{A}_{\bar\theta} \delta_\beta \,, \tag{192}$$

in addition to the transformation rule

$$\delta_T \mathbb{A}_\theta = \mathbb{A}_T \delta_\beta \mathbb{A}_\theta - \mathbb{A}_\theta \delta_\beta \mathbb{A}_T - D_\theta \mathbb{A}_T \,, \tag{193}$$

and an analogous transformation for $\mathbb{A}_{\bar\theta}$.

Finally we need to consider a modified measure similar to the discussion around (177). In our scalar field example we had to modify the measure so that the Lagrangian density carried charge $-1$. A straightforward computation shows that in our sigma model, we do not need to replace the measure at all:

$$\sqrt{-\mathfrak{g}} \to \frac{\sqrt{-\mathfrak{g}^{(A)}}}{\beta^i \mathbb{A}_i + 1} = \sqrt{-\mathfrak{g}} \,. \tag{194}$$

We have almost completed our construction of a $\delta_T$-invariant action. Recall, however, that the Schwinger-Keldysh effective action is invariant under a $\mathbb{Z}_2$ KMS symmetry which results in a KMS partner term (145). To ensure that the KMS partner Lagrangian is also invariant under $\delta_T$, we define the action of the $\mathbb{Z}_2$ symmetry $K$ on the super-connection as

$$K(\mathbb{A}_i) = -\eta_i \widetilde{\mathbb{A}}_i \,, \qquad K(\mathbb{A}_\theta) = \widetilde{\mathbb{A}}_{\bar\theta} \,, \qquad K(\mathbb{A}_{\bar\theta}) = -\widetilde{\mathbb{A}}_\theta \,, \tag{195}$$

with

$$\widetilde{\mathbb{A}}_I = \mathbb{A}_I + \bar\theta\theta\, i\delta_\beta A_{r\,I} \,. \tag{196}$$

With this sign choice, we have

$$K\left(\mathfrak{g}_{ij}^{(A)}\right) = \eta_i\eta_j\widetilde{\mathfrak{g}}_{ij}^{(\widetilde{A})} \,, \qquad \widetilde{\mathfrak{g}}_{ij}^{(\widetilde{A})} = \widetilde{\mathfrak{g}}_{kl}(\delta_i^k + \widetilde{\mathbb{A}}_i\beta^k)(\delta_j^l + \widetilde{\mathbb{A}}_j\beta^l) \,, \tag{197}$$

and similarly for the flavor field. We also have

$$K\left(\mathbb{W}_i^{(A)}\right) = \eta_i\widetilde{\mathbb{W}}_i^{(\widetilde{A})} \,, \qquad K\left(D_\theta^{(A)}\right) = -\widetilde{D}_{\bar\theta}^{(\widetilde{A})} \,, \qquad K\left(D_{\bar\theta}^{(A)}\right) = \widetilde{D}_\theta^{(\widetilde{A})} \,, \tag{198}$$

where $\widetilde{\mathbb{W}}^{(\widetilde{A})} = \partial + \widetilde{\mathbb{A}}\delta_\beta + \widetilde{\mathbb{\Gamma}}^{(\widetilde{A})}$ is the covariant derivative taken with the connection

$$\widetilde{\mathbb{\Gamma}}^{(\widetilde{A})\,i}{}_{jk} = \frac{1}{2}\widetilde{\mathfrak{g}}^{(\widetilde{A})\,il}\left(\partial_j^{(\widetilde{A})}\widetilde{\mathfrak{g}}_{kl}^{(\widetilde{A})} + \partial_k^{(\widetilde{A})}\widetilde{\mathfrak{g}}_{jl}^{(\widetilde{A})} - \partial_l^{(\widetilde{A})}\widetilde{\mathfrak{g}}_{jk}^{(\widetilde{A})}\right) \,, \tag{199}$$

and

$$\widetilde{D}_\theta^{(\widetilde{A})} = \frac{\partial}{\partial\theta} + \widetilde{\mathbb{A}}_\theta\delta_\beta \,, \qquad \widetilde{D}_{\bar\theta}^{(\widetilde{A})} = \frac{\partial}{\partial\bar\theta} - i\theta\delta_\beta + \widetilde{\mathbb{A}}_{\bar\theta}\delta_\beta \,. \tag{200}$$

Defining

$$\widetilde{\mathbb{A}}_T = \mathbb{A}_T + \bar\theta\theta i\delta_\beta\Lambda_{T\,r} \,, \tag{201}$$

it then follows from (187) and (193) that the components of $\widetilde{\mathbb{A}}_I$ vary under the transformation $\mathbb{A}_T$ as

$$\begin{aligned}
\delta_T\widetilde{\mathbb{A}}_i &= \widetilde{\mathbb{A}}_T\delta_\beta\widetilde{\mathbb{A}}_i - \widetilde{\mathbb{A}}_i\delta_\beta\widetilde{\mathbb{A}}_T - \partial_i\widetilde{\mathbb{A}}_T \,, \\
\delta_T\widetilde{\mathbb{A}}_\theta &= \widetilde{\mathbb{A}}_T\delta_\beta\widetilde{\mathbb{A}}_\theta - \widetilde{\mathbb{A}}_\theta\delta_\beta\widetilde{\mathbb{A}}_T - \widetilde{D}_\theta\widetilde{\mathbb{A}}_T \,, \\
\delta_T\widetilde{\mathbb{A}}_{\bar\theta} &= \widetilde{\mathbb{A}}_T\delta_\beta\widetilde{\mathbb{A}}_{\bar\theta} - \widetilde{\mathbb{A}}_{\bar\theta}\delta_\beta\widetilde{\mathbb{A}}_T - \widetilde{D}_{\bar\theta}\widetilde{\mathbb{A}}_T \,,
\end{aligned} \tag{202}$$

and the tilde'd super-pullbacks transform homogeneously

$$\delta_T\widetilde{\mathfrak{g}}_{ij}^{(\widetilde{A})} = \widetilde{\mathbb{A}}_T\delta_\beta\widetilde{\mathfrak{g}}_{ij}^{(\widetilde{A})} \,, \qquad \delta_T\widetilde{\mathbb{B}}_i^{(\widetilde{A})} = \widetilde{\mathbb{A}}_T\delta_\beta\widetilde{\mathbb{B}}_i^{(\widetilde{A})} \,. \tag{203}$$

Recall that the action

$$S_{eff} = \int d^d\sigma d\theta d\bar\theta\left(\sqrt{-\mathfrak{g}}\,L + \sqrt{-\widetilde{\mathfrak{g}}}\,\widetilde{L}\right) \tag{204}$$

was constructed so that it is invariant under all the symmetries of the problem. Having defined the action of $K$ on $\mathbb{A}_I$, we observe that the minimally coupled action

$$S_{eff}^{(A)} = \int d^d\sigma d\theta d\bar\theta\left\{\sqrt{-\mathfrak{g}}\,L^{(A)} + \sqrt{-\widetilde{\mathfrak{g}}}\,\widetilde{L}^{(\widetilde{A})}\right\} \tag{205}$$

is invariant under all the symmetries of the problem and under $\delta_T$.

In analogy with (182), the supercurrent

$$\mathbb{S}^I = S'^I + \theta S_{\bar g}^I + \bar\theta S_g^I + \bar\theta\theta S_t^I = -\frac{1}{\sqrt{-\mathfrak{g}}}\left.\frac{\delta S_{eff}^{(A)}}{\delta\mathbb{A}_I}\right|_{\mathbb{A}_I=0} \,, \tag{206}$$

satisfies the off-shell Ward identity

$$\mathbb{W}_i \mathbb{S}^i + D_\theta \mathbb{S}^\theta + D_{\bar\theta} \mathbb{S}^{\bar\theta} + \beta_\mu \left( \mathbb{D}_\nu \mathbb{T}^{\mu\nu} - \mathbb{G}^\mu{}_\nu \mathbb{J}^\nu \right) + \nu \, \mathbb{D}_\mu \mathbb{J}^\mu = 0 \, . \tag{207}$$

The terms proportional to $\beta_\mu$ and $\nu$ are the equations of motion for $\mathbb{X}^\mu$ and $\mathbb{C}$, where $\mathbb{T}^{\mu\nu} = \mathbb{T}^{ij} \partial_i \mathbb{X}^\mu \partial_j \mathbb{X}^\nu$ and $\mathbb{T}^{ij}$ is the super-stress tensor conjugate to $\mathfrak{g}_{ij}$ and so on. As a result, on-shell, the current $\mathbb{S}^I$ is conserved in superspace,

$$\mathbb{W}_i \mathbb{S}^i + D_\theta \mathbb{S}^\theta + D_{\bar\theta} \mathbb{S}^{\bar\theta} \big|_{\text{on-shell}} = 0 \, . \tag{208}$$

The bottom component of (208) is given by

$$\nabla_i S'^i = -S^\theta_{\bar g} - S^{\bar\theta}_g \, . \tag{209}$$

We now argue that $S'^i$ is closely related to the hydrodynamic entropy current, upon setting the $a$-fields and ghosts to vanish, and the right-hand side of (209) characterizes its non-conservation.

Let us compute $\mathbb{S}$ at zeroth order in derivatives. The appropriate action for such an analysis is given by (152) with (156). Coupling this action to $\mathbb{A}$ we find

$$S_{eff}^{(A)} = \int d^d\sigma d\theta d\bar\theta \sqrt{-\mathfrak{g}} \, P \left( \mathbb{T}^{(A)}, \nu^{(A)} \right) , \tag{210}$$

where

$$\mathbb{T}^{(A)} = \frac{1}{\sqrt{-\mathfrak{g}_{ij}^{(A)} \beta^i \beta^j}} = \frac{\mathbb{T}}{\beta^i \mathbb{A}_i + 1} \, ,$$
$$\nu^{(A)} = \nu(\beta^i \mathbb{A}_i + 1) \, , \tag{211}$$

and $P$ is an even function of $\nu^{(A)}$. Varying with respect to $\mathbb{A}_i$ we obtain

$$\mathbb{S}^i = \left( \frac{\partial P}{\partial \mathbb{T}} \mathbb{T} - \frac{\partial P}{\partial \nu} \nu \right) \beta^i , \qquad S^\theta = S^{\bar\theta} = 0 \, . \tag{212}$$

Recalling that the entropy density $s$ is related to the pressure and temperature via

$$s = \left( \frac{\partial P}{\partial T} \right)_\mu , \tag{213}$$

we find, using (160) and pushing forward $S'^i$ to a vector $S'^\mu$ in the physical space, that

$$S'^\mu = s u^\mu + O(\partial) \, . \tag{214}$$

Thus, $S'^i$ coincides with the entropy current at zeroth order in derivatives.

The entropy current $S^\mu$ has two defining properties: it must coincide with $s u^\mu$ at zeroth order in derivatives, and it must have non-negative divergence. We will now show that $S^i$ may be constructed from $S'^i$ by adding to the latter appropriate higher derivative corrections. To start, let us first deduce $S^{\bar\theta}_g$ and $S^\theta_{\bar g}$. A straightforward but tedious computation gives us

$$S^\theta_{\bar g} + S^{\bar\theta}_g = \frac{1}{2} \left( L^{AB} + \eta_{AB} \widetilde{L}^{AB} \right) \delta_\beta F_{r\,A} \delta_\beta F_{r\,B}$$
$$+ \frac{1}{2} \sum_{n=1}^\infty (-1)^n \eta_{A\ldots C_n} \widetilde{L}^{ABC_1\ldots C_n} \delta_\beta F_{r\,A} \delta_\beta F_{r\,B} \delta_\beta F_{r\,C_1} \ldots \delta_\beta F_{r\,C_n} - \nabla_i J_S^i \, , \tag{215}$$

where, with some abuse of notation, $L^{AB}$, $\widetilde{L}^{AB}$ and $\widetilde{L}^{ABC_1\ldots C_n}$ refer to the bottom components of the quantities defined in (143) and (145), and we have set all $a$-type fields and ghosts to zero. The divergence of $J_S^i$ which appears in the last line of (215) comes about as follows. Recall that the $L^{A\ldots C_n}$'s are differential operators. When varying the action with respect to $\mathbb{A}_\theta$ or $\mathbb{A}_{\bar\theta}$ we may need to integrate by parts. The term $\nabla_i J_S^i$ accounts for this procedure.

The expression in (215) may be simplified by a relabeling of the terms in the action. By making the replacement

$$
\begin{aligned}
L^{AB} &\to L^{AB} - \frac{1}{2} L^{ABC_1} \delta_\beta \mathbb{F}_{C_1} , \\
L^{ABC_1\ldots C_n} &\to L^{ABC_1\ldots C_n} - L^{ABC_1\ldots C_n C_{n+1}} \delta_\beta \mathbb{F}_{C_{n+1}} \qquad n \geq 1 ,
\end{aligned}
\tag{216}
$$

in the Lagrangian (143) (and an appropriate replacement in (145)), expression (215) simplifies to

$$
\begin{aligned}
S_{\bar g}^\theta + S_g^{\bar\theta} = & \frac{1}{2} \left( L^{AB} + \eta_{AB} \widetilde{L}^{AB} \right) \delta_\beta F_{r\,A} \delta_\beta F_{r\,B} \\
& - \frac{1}{4} \left( L^{ABC} + \eta_{ABC} \widetilde{L}^{ABC} \right) \delta_\beta F_{r\,A} \delta_\beta F_{r\,B} \delta_\beta F_{r\,C} - \nabla_i J_S^i .
\end{aligned}
\tag{217}
$$

Recall that $\text{Im}(S_{eff})$ must be non-negative due to unitarity for any field configuration. As we will see shortly, positivity of the imaginary part of the effective action leads to a positive entropy production. The imaginary part of the action is given by

$$
\begin{aligned}
\text{Im}(S_{eff}) = & -\frac{1}{2} \int d^d\sigma \sqrt{-g_r} \Bigg[ \left( L^{AB} + \eta_{AB} \widetilde{L}^{AB} \right) F_{a\,A} F_{a\,B} \\
& - \frac{1}{2} \big( L^{ABC_1} F_{a\,B} (2 F_{a\,C} \delta_\beta F_{r\,A} + F_{a\,A} \delta_\beta F_{r\,C}) \\
& + \eta_{ABC_1} \widetilde{L}^{ABC_1} F_{a\,A} (2 F_{a\,C} \delta_\beta F_{r\,B} + F_{a\,B} \delta_\beta F_{r\,C}) big) \\
& + \sum_{n=2} \Big( (-1)^{\lfloor \frac{n-1}{2} \rfloor} L^{A\ldots C_n} F_{a\,B} \ldots F_{a\,C_{n-1}} \alpha_{AC_n} \\
& + (-1)^n \eta_{A\ldots C_n} \widetilde{L}^{A\ldots C_n} F_{a\,A} F_{a\,C_n} \sum_{j=0}^{\lfloor \frac{n}{2} \rfloor} \Pi^{2j}_{B\ldots C_{n-1}} \Big) \Bigg] ,
\end{aligned}
\tag{218}
$$

where $\lfloor m \rfloor$ is the floor of $m$ and we have defined

$$
\alpha_{AC_n} = \begin{cases} F_{a\,A} F_{a\,C_n} - \delta_\beta F_{r\,A} \delta_\beta F_{r\,C_n} & n \text{ even} \\ F_{a\,A} \delta_\beta F_{r\,C_n} + \delta_\beta F_{a\,A} F_{a\,C_n} & n \text{ odd} \end{cases} ,
\tag{219}
$$

and $\Pi^{2j}_{B\,C_1,\ldots C_{n-1}}$ gives the sum over all permutations of distinct $2j$ $a$-type fields and distinct $n-2-2j$ $r$-type fields on which $\delta_\beta$ acts. For example,

$$
\Pi^0_{B\,C_1} = \delta_\beta F_{r\,B}
\tag{220}
$$

or

$$
\Pi^2_{B\,C_1\,C_2} = F_{a\,B} F_{a\,C_1} \delta_\beta F_{r\,C_2} + F_{a\,B} F_{a\,C_2} \delta_\beta F_{r\,C_1} + F_{a\,C_1} F_{a\,C_2} \delta_\beta F_{r\,B} .
\tag{221}
$$

The right-hand side of (218) must be non-negative for any field configuration. In the absence of a non-perturbative expression for the action it is difficult, if not impossible, to solve the positivity constraint $\text{Im}(S_{eff}) \geq 0$ exactly. However, by working perturbatively in derivatives we may obtain a necessary condition for positivity,

$$
\sigma^{AB} \equiv - \lim_{\partial \to 0} \left( L^{AB} + \eta_{AB} \widetilde{L}^{AB} \right) \succeq 0 .
\tag{222}
$$

Following [2,25] we may use (222) to organize the right-hand side of (217) into a quadratic form, up to total derivatives, order by order in the derivative expansion. Explicitly,

$$-S_g^\theta - S_g^{\bar\theta} = \sigma \left( \delta_\beta F_r + \frac{1}{2}\sigma^{-1} \left( Q_{(2)} + \ldots + Q_{(n-1)} \right) \right)^2 + \nabla J_S', \tag{223}$$

with $Q_{(n)}$ an $n$'th derivative vector and where we have omitted the Latin indices for brevity.[8] With these definitions we find that

$$S^i = S'^i - J_S^i - J_S'^i \tag{224}$$

satisfies

$$S^i = su^i + \mathcal{O}(\partial) \tag{225}$$

(since $J_S^i$ and $J_S'^i$ are at least second order in derivatives) and the on-shell relation

$$\nabla_i S^i \Big|_{\text{on-shell}} \geq 0. \tag{226}$$

We may now identify $S^i$ with the entropy current. In Sections 6 and 7 we will see that up to second order in derivatives, the right-hand side of $\nabla_i S^i$ takes the standard form $\frac{1}{T}\zeta\Theta^2 + \frac{1}{T}\eta\sigma^{\mu\nu}\sigma_{\mu\nu}$ where $\zeta$ and $\eta$ are the bulk and shear viscosities, $\Theta$ and $\sigma^{\mu\nu}$ are the divergence of the velocity field and shear tensor and $T$ is the temperature.

Note that the relation (222) which implies the local Second Law, is a necessary but not sufficient condition for the imaginary part of the effective action to be positive semi-definite. Looking at the full expression for (218) we find that certain transport coefficients associated with three-tensor terms (and some associated with two-tensor terms) may be constrained by positivity of the imaginary part of the effective action but not by positivity of the entropy current. We will discuss this observation in detail in the remainder of this work.

## 6 A panoply of transport coefficients

In what follows we will carry out a detailed analysis of the possible constitutive relations which can result from the Schwinger-Keldysh effective action and the constraints imposed on them. In Subsection 6.1 we will discuss the Onsager relations and CPT transformation properties of the constitutive relations and then in Subsection 6.2 we will classify the possible resulting transport coefficients. But before proceeding with a detailed analysis we pause to consider the general structure of the possible transport coefficients. In this preamble we will focus on how the transport coefficients behave under CPT and whether or not they are subject to a positivity condition. The main results are Eq. (237), where we decompose the constitutive relations according to how the various terms transform under KMS, and Eq. (244), where we compute $\text{Im}(S_{eff})$ in terms of the coefficients appearing in the constitutive relations.

Let us denote both the $U(1)$ current and the stress tensor by $J_r^A$ which is associated with the field $F_A$ such that (147) becomes

$$J_r^A = \frac{1}{\sqrt{-g_r}} \frac{\delta S_{eff}}{\delta F_{a\,A}} \Bigg|_{\substack{\mathbb{X}=X_r \\ \mathbb{C}=C_r \\ F_a=0}}. \tag{227}$$

---

[8]It is an interesting question whether the divergence of the current $S'^i$ is positive (up to a total derivative) even when the leading order terms in a derivative expansion vanish. We leave this issue open for future exploration.

Given the action (136) with (143) and (145) and the replacements (216), we may write

$$J_r^A = J_{r\,(0)}^A + J_{r\,(2)}^A + J_{r\,(3)}^A \,, \tag{228}$$

where the first term represents the contributions to $J_r^A$ coming from the scalar terms $L_0 + \widetilde{L}_0$ in the Lagrangian and the remaining terms represent contributions from 2 and 3-tensor terms. The remaining tensor terms do not contribute to the constitutive relations on account of (216).

A formal computation gives us

$$
\begin{aligned}
J_{r\,(0)}^A &= \frac{1}{2}\frac{1}{\sqrt{-g_r}}\frac{\delta}{\delta F_{r\,A}}\int d^d\sigma\sqrt{-g_r}\Big(L_0 + \widetilde{L}_0\Big)\Big|_{a=\text{ghosts}=0}\,, \\
J_{r\,(2)}^A &= \frac{1}{2}\Big(L^{BA} + \eta_{AB}\widetilde{L}^{AB}\Big)\delta_\beta F_{r\,B}\,, \\
J_{r\,(3)}^A &= -\frac{1}{4}\Big(L^{BAC} + \eta_{ABC}\widetilde{L}^{ABC}\Big)\delta_\beta F_{r\,B}\delta_\beta F_C\,.
\end{aligned}
\tag{229}
$$

We remind the reader that the $L^{AB\cdots}$ is a differential operator so that the right-hand side of (229) should be thought of as a formal expression where the corresponding term in the effective action has been integrated by parts.

To make our analysis more explicit let us write out the various tensor terms where the differential operators are spelled out, viz.,

$$
\begin{aligned}
&L^{ABC}\left(D_\theta\mathbb{F}_A\right)\left(D_{\bar\theta}\mathbb{F}_B\right)\left(D\mathbb{F}_C\right) \\
&= L^{ABC\ell_1\dots\ell_a i_1\dots i_b j_1\dots j_c}\left(\mathbb{W}_{\ell_1}\dots\mathbb{W}_{\ell_a}D_\theta\mathbb{F}_A\right)\left(\mathbb{W}_{i_1}\dots\mathbb{W}_{i_b}D_{\bar\theta}\mathbb{F}_B\right)\left(\mathbb{W}_{j_1}\dots\mathbb{W}_{j_c}D\mathbb{F}_C\right)\,,
\end{aligned}
\tag{230}
$$

where $L^{ABC}$ is a differential operator but $L^{A\dots j_c}$ is not, and there is a similar definition for $L^{AB}$. In Appendix D we show that, after some massaging, any tensor term in the Lagrangian can be made to take a form similar to (230) up to possible boundary terms. In what follows, in order to avoid cluttering our equations, we will use $L^{ABC\{i\}_{abc}}$ or $L^{AB\{i\}_{ab}}$ to denote the tensor term coefficients and $\mathbb{W}^c$ in place of $\mathbb{W}_{i_1}\dots\mathbb{W}_{i_c}$. We will treat the index $c$ to the right of $\mathbb{W}$ as counting the number of derivatives in the expression. We will switch to more explicit notation when appropriate.

We now find that (229) takes the form

$$
\begin{aligned}
J_{r\,(0)}^A &= \frac{1}{2}\frac{\delta}{\delta F_{r\,A}}\int d^d\sigma\sqrt{-g_r}\Big(L_0 + \widetilde{L}_0\Big)\Big|_{a=\text{ghosts}=0}\,, \\
J_{r\,(2)}^A &= \frac{1}{2}\nabla^{a\dagger}\left[\Big(L^{BA\{i\}_{ba}} + \eta_{AB\{i\}_{ab}}\widetilde{L}^{AB\{i\}_{ab}}\Big)\nabla^b\delta_\beta F_{r\,B}\right]\,, \\
J_{r\,(3)}^A &= -\frac{1}{4}\nabla^{a\dagger}\left[\Big(L^{BAC\{i\}_{bac}} + \eta_{ABC\{i\}_{abc}}\widetilde{L}^{ABC\{i\}_{abc}}\Big)(\nabla^b\delta_\beta F_{r\,B})(\nabla^c\delta_\beta F_{r\,C})\right]\,,
\end{aligned}
\tag{231}
$$

where $\eta_{ABC\{i\}_{abc}}$ is the CPT eigenvalue associated to $\mathbb{W}_{\ell_1}\dots\mathbb{W}_{\ell_a}\mathbb{F}_A\dots\mathbb{W}_{j_1}\dots\mathbb{W}_{j_c}\mathbb{F}_C$, $\nabla^b = \nabla_{i_1}\dots\nabla_{i_b}$, $\nabla^{a\dagger} = (-1)^a\nabla_{i_a}\dots\nabla_{i_1}$ and $\widetilde{L}_0$ and $\widetilde{L}^{A\cdots}$ are related to $L_0$ and $L^{A\cdots}$ by KMS conjugation

$$F_{r\,A} \to \eta_A F_{r\,A}\,,\quad \nabla_i \to \eta_i\nabla_i\,,\quad \beta^i \to -\eta_i\beta^i\,,\quad \Lambda_\beta \to -\Lambda_\beta\,,\quad \nu \to -\nu\,,\quad T \to T\,, \tag{232}$$

and so

$$
\begin{aligned}
\widetilde{L}_0(F_{r\,A},\nabla_i\,;\beta^i,\Lambda_\beta) &= L_0(\eta_A F_{r\,A},\eta_i\nabla_i\,;-\eta_i\beta^i,-\Lambda_\beta)\,, \\
\widetilde{L}^{A\cdots}(F_{r\,A},\nabla_i\,;\beta^i,\Lambda_\beta) &= L^{A\cdots}(\eta_A F_{r\,A},\eta_i\nabla_i\,;-\eta_i\beta^i,-\Lambda_\beta)\,,
\end{aligned}
\tag{233}
$$

where fields are merely multiplied by their eigenvalues under CPT since we have set the ghosts and $a$-fields to vanish. The expressions in (231) are schematic. Each represents a class of contributions associated with an appropriate tensor term.

In what follows, we will find it useful to characterize the transformation properties of the various transport coefficients under KMS conjugation. To this end, consider

$$
\begin{aligned}
L_\pm^{AB\{i\}_{ab}} &= L^{AB\{i\}_{ab}} \mp \eta_{AB\{i\}_{ab}} \widetilde{L}^{AB\{i\}_{ab}} , \\
L_\pm^{ABC\{i\}_{abc}} &= L^{ABC\{i\}_{abc}} \pm \eta_{ABC\{i\}_{abc}} \widetilde{L}^{ABC\{i\}_{abc}} .
\end{aligned}
\tag{234}
$$

Under KMS conjugation the term involving $L_\pm^{A\cdots}$ in the current transforms as

$$
\nabla^{a\dagger} \left( L_\pm^{A\ldots C\{i\}_{a\ldots c}} \nabla^b \delta_\beta F_{r\,B} \nabla^c \delta_\beta F_{r\,C} \right) \to \pm \eta_A \nabla^{a\dagger} \left( L_\pm^{ABC\{i\}_{abc}} \nabla^b \delta_\beta F_{r\,B} \nabla^c \delta_\beta F_{r\,C} \right)
\tag{235}
$$

(and a similar equation for $L^{AB\{i\}_{ab}}$). In obtaining (235) we have used $\eta^2 = 1$ and the fact that $K(\delta_\beta) = -\delta_\beta$. We refer to the $L_+^{A\cdots}$ terms in the currents as KMS-even, and the $L_-^{A\cdots}$ as KMS-odd. The KMS-even terms transform in same way as the currents, while the KMS-odd terms transform in the opposite way.

Let us pause to discuss two consequences of KMS conjugation and CPT invariance which will become important in what follows. First, although the original effective action is invariant under KMS, the thermal expectation value of the currents may receive contributions which are both KMS-even and KMS-odd. The KMS-odd terms arise only from the tensor terms in the effective action. Second, the tensor terms lead to transport coefficients whose behavior under CPT is fully determined by the KMS symmetry. The same holds for the transport coefficients coming from a scalar Lagrangian, as we discuss in the next Subsection. Given a transport coefficient which multiplies some tensor structure in the constitutive relations, the CPT-eigenvalue of the coefficient is just the product of the KMS-parity of the whole term and the KMS-eigenvalue of the tensor structure. Because we have in mind theories where CPT is only broken by sources, CPT-even transport coefficients are even functions of chemical potential, while CPT-odd coefficients are odd functions of chemical potential.

Let us further define the quantities

$$
\begin{aligned}
L_+^{[AB\{i\}_{ab}]} &= \frac{1}{2} \left( L_+^{AB\{i\}_{ab}} - L_+^{BA\{i\}_{ba}} \right) , \\
L_-^{(AB\{i\}_{ab})} &= \frac{1}{2} \left( L_-^{AB\{i\}_{ab}} + L_-^{BA\{i\}_{ba}} \right) , \\
N_-^{[AB\{i\}_{ab}]} &= \frac{1}{2} \left( L_-^{ABC\{i\}_{abc}} - L_-^{BAC\{i\}_{bac}} \right) \nabla^c \delta_\beta F_{r\,C} , \\
N_+^{(AB\{i\}_{ab})} &= \frac{1}{2} \left( L_+^{ABC\{i\}_{abc}} + L_+^{BAC\{i\}_{bac}} \right) \nabla^c \delta_\beta F_{r\,C} .
\end{aligned}
\tag{236}
$$

The first two lines of (236) correspond to the KMS-even and KMS-odd parts of the 2-tensor terms, and the last two to the KMS-even and KMS-odd parts of the 3-tensor terms. In terms of these quantities the constitutive relations for the tensor terms take the somewhat simple form

$$
J_{\text{tensor}}^A = \frac{1}{2} \nabla^{a\dagger} \left[ \left( -L_+^{[AB\{i\}_{ab}]} + L_-^{(AB\{i\}_{ab})} - \frac{1}{2} N_+^{(AB\{i\}_{ab})} + \frac{1}{2} N_-^{[AB\{i\}_{ab}]} \right) \nabla^b \delta_\beta F_{r\,B} \right] .
\tag{237}
$$

Parity of the various transport coefficients under an exchange of indices or under KMS serve as the basis for the Onsager relations, as we demonstrate in Section 6.1. Eq. (237) is our first main result. We have organized the constitutive relations according to their KMS-parity as well as their index structure.

Let us make a comment on notation which we will use extensively throughout this Section and the remainder of this work. We will find it convenient to decompose tensors into components whose symmetry under an exchange of $A$ and $B$ or $a$ and $b$ is well defined. For a general tensor $T^{AB\{i\}ab}$ we will define circular or square brackets on pairs of indices to denote symmetrization or antisymmetrization, e.g.,

$$T^{[AB]\{i\}ab} = \frac{1}{2}\left(T^{AB\{i\}ab} - T^{BA\{i\}ab}\right) \quad \text{or} \quad T^{AB\{i\}(ab)} = \frac{1}{2}\left(T^{AB\{i\}ab} + T^{AB\{i\}ba}\right),$$
(238)

while a square or circular bracket around all four indices implies symmetry under a simultaneous exchange of $A$ with $B$ and of $a$ with $b$. Thus, for instance,

$$T^{[[AB]\{i\}(ab)]} = \frac{1}{2}\left(T^{[AB]\{i\}(ab)} - T^{[BA]\{i\}(ba)}\right) = T^{[AB]\{i\}(ab)}.$$
(239)

With this notation in mind we have, for example,

$$T^{[AB\{i\}ab]} = T^{[[AB]\{i\}(ab)]} + T^{[(AB)\{i\}[ab]]} = T^{[AB]\{i\}(ab)} + T^{(AB)\{i\}[ab]}.$$
(240)

Note that $T^{[(AB)\{i\}(ab)]} = T^{[[AB]\{i\}[ab]]} = 0$.

With our explicit notation, entropy production is given by

$$\nabla_i S^i = -\frac{1}{2}\left(L_-^{(AB\{i\}ab)} - \frac{1}{2}N_+^{(AB\{i\}ab)}\right)(\nabla^a \delta_\beta F_{r\,A})(\nabla^b \delta_\beta F_{r\,B}) + \nabla_i J_S^i\Big|_{\text{on-shell}},$$
(241)

positivity of entropy production (226) is guaranteed by

$$-\frac{1}{2}\int d^d\sigma \sqrt{-g_r}\, L_-^{(AB\{i\}ab)} \nabla^a \delta_\beta F_{r\,A} \nabla^b \delta_\beta F_{r\,B}\bigg|_{\text{on-shell}} \geq 0,$$
(242)

and its positivity is guaranteed perturbatively by

$$-\lim_{\partial\to 0}\frac{1}{2}L_-^{(AB\{i\}ab)} \succeq 0.$$
(243)

See (222). Schwinger-Keldysh positivity imposes the more stringent constraint

$$\begin{aligned} \text{Im}(S_{eff}) = -\frac{1}{2}\int d^d\sigma \sqrt{-g_r}\Big[&L_-^{(AB\{i\}ab)} \nabla^a F_{a\,A} \nabla^b F_{a\,B} \\ &-\frac{1}{2}\nabla^a F_{a\,A} \nabla^b F_{a\,B} N_+^{(AB\{i\}ab)} \\ &-\nabla^a F_{a\,B} \nabla^b \delta_\beta F_{r\,A}\left(N_-^{[AB\{i\}ab]} + N_+^{(AB\{i\}ab)}\right)\Big|_{\delta_\beta F_{r\,C}\to F_{a\,C}} \\ &+\left(\text{higher tensor} \atop \text{terms}\right).\end{aligned}$$
(244)

In comparing (244) with (242) we find that the on-shell value of $L_-^{(AB\{i\}ab)}$ and $N_+^{(AB\{i\}ab)}$ is constrained by the positivity of entropy production and that their off-shell value is constrained by the Schwinger-Keldysh positivity condition. In classifying the various transport we will find it useful to further separate $L_-^{(AB\{i\}ab)}$ and $N_+^{(AB\{i\}ab)}$ into two disjoint sets,

$$P_-^{(AB\{i\}ab)} = \left\{L_-^{(AB\{i\}ab)} \,\Big|\, L_-^{(AB\{i\}ab)} \nabla^a \delta_\beta F_{r\,A} \nabla^b \delta_\beta F_{r\,B}\Big|_{\text{on-shell}} = 0\right\},$$
(245)

such that

$$L_-^{(AB\{i\}ab)} = M_-^{(AB\{i\}ab)} + P_-^{(AB\{i\}ab)},$$
(246)

and

$$P_+^{(AB\{i\}ab)} = \left\{ N_+^{(AB\{i\}ab)} \,\middle|\, N_+^{(AB\{i\}ab)} \nabla^a \delta_\beta F_{r\,A} \nabla^b \delta_\beta F_{r\,B} \Big|_{\text{on-shell}} = 0 \right\} , \qquad (247)$$

such that

$$N_+^{(AB\{i\}ab)} = M_+^{(AB\{i\}ab)} + P_+^{(AB\{i\}ab)} . \qquad (248)$$

Eq. (244) is the second main result of this Section. Note that only the 2- and 3-tensor terms contribute to the constitutive relations, but all of the tensor terms contribute to $\text{Im}(S_{eff})$. It is now straightforward to classify the transport coefficients appearing in (237) according to how they contribute to entropy production (242), or to Schwinger-Keldysh positivity (244). We divide the tensor terms into four classes: we call the $M_\pm^{(AB\{i\}ab)}$ terms dissipative (as they determine the entropy production), the $P_\pm^{(AB\{i\}ab)}$ pseudo-dissipative (as they do not), the $L_+^{[AB\{i\}ab]}$ terms non-dissipative since they do not contribute to entropy production and they are unconstrained by the Schwinger-Keldysh positivity condition, and $N_-^{[AB\{i\}ab]}$ as exceptional (since they are constrained by (244) but do not produce entropy).

In the remainder of this Section we will first expand on the interplay between CPT and KMS transformation properties of the constitutive relations where we will also see how the underlying KMS symmetry implies the Onsager reciprocity relations. We will then turn our attention to the various classes of transport as described above and work out some simple examples of each.

## 6.1 The Onsager reciprocity relations and CPT

In the context of hydrodynamics, the Onsager reciprocity relations [4, 5] imply certain correlations between transport coefficients. These relations follow from transformation properties of correlation functions under CPT and are independent of the constraints generated by positing the existence of an entropy current.

The Schwinger-Keldysh effective action is not invariant under CPT. It is, however, invariant under the $\mathbb{Z}_2$ KMS symmetry, which includes a CPT-flip. For this reason the KMS symmetry implies that certain transport coefficients are even under CPT, and others odd, but the map is not immediate. In the previous Subsection we saw that the CPT-eigenvalue of transport coefficients is equal to the KMS parity of the term it appears in, times the KMS-eigenvalue of the tensor structure it multiplies. CPT-even transport coefficients are even functions of chemical potential, and CPT-odd coefficients are odd functions of chemical potential. In this way the Onsager relations are enforced by the structure of the Schwinger-Keldysh effective action. In the remainder of this Subsection we illustrate this interplay between KMS and CPT in more detail.

We begin with the scalar terms. From the formal expression (229) for the currents, it is manifest that the scalar part of the action is CPT-even. This generalizes an earlier result that the action for ideal hydrodynamics is a pressure term $\int d^d\sigma d\theta d\bar\theta \sqrt{-\mathbb{g}}\, P$ with $P$ an even function of chemical potential. Beyond ideal hydrodynamics, transport coefficients which multiply CPT-even scalars in the action are themselves even under CPT, while transport coefficients which multiply CPT-odd scalars are themselves odd under CPT. Moreover, the contribution of the scalar terms to the current, which we called $J_{r(0)}^A$, transforms under KMS (232) as

$$J_{r(0)}^A(\nabla, F_r) = \eta_A J_{r(0)}^A(\eta\nabla, \eta F_r) , \qquad (249)$$

and so in the language we used in the last Subsection, the scalar contribution to the current is KMS-even.

Next consider the 2-tensor terms. Recall that their contribution to the constitutive relations is given by (237), which we may write as

$$J^A_{r\,(2)} = J^A_{(2)+} + J^A_{(2)-}\,, \tag{250}$$

with

$$J^A_{(2)+} = -\frac{1}{2}\nabla^{a\dagger}\left(L_+^{[AB\{i\}_{ab}]}\nabla^b\delta_\beta F_{r\,B}\right)\,, \qquad J^A_{(2)-} = \frac{1}{2}\nabla^{a\dagger}\left(L_-^{(AB\{i\}_{ab})}\nabla^b\delta_\beta F_{r\,B}\right)\,, \tag{251}$$

and $L_\pm^{A\cdots}$ were defined in (234) and (236). Using (235) we obtain

$$J^A_{(2)\pm}(\nabla, F_r) = \pm\eta_A J^A_{(2)\pm}(\eta\nabla, \eta F_r)\,. \tag{252}$$

So the 2-tensor terms in the current contain both KMS-even and KMS-odd transport. From (242) and (244) we see that the former is dissipationless, while the latter includes dissipative contributions.

Note that in the above example, $L_+^{[AB\{i\}_{ab}]}$ is odd under a joint exchange of $A$ and $a$ with $B$ and $b$ respectively, while $L_-^{(AB\{i\}_{ab})}$ is even under such an exchange. This statement, together with (252), encapsulates the Onsager relations. To see how the Onsager relations emerge in a more familiar form, let us consider the example of the conductivities for a parity-violating fluid in two spatial dimensions [9]. Consider a fluid with several $U(1)$ currents labeled by $\alpha$. There are two conductivity matrices, the longitudinal conductivity $\sigma^{\alpha\beta}$ and the Hall conductivity $\bar{\sigma}^{\alpha\beta}$, which appear in the constitutive relations as

$$J^{\mu\alpha} = \ldots + \sigma^{\alpha\beta}V^\mu_\beta + \bar{\sigma}^{\alpha\beta}\epsilon^{\mu\nu\rho}u_\nu V_{\beta\,\rho}\,, \tag{253}$$

where

$$P^{\mu\nu} = g^{\mu\nu} + u^\mu u^\nu\,, \qquad V_{\alpha\,\mu} = E_{\alpha\,\mu} - TP_{\mu\nu}\partial^\nu\nu_\alpha = -TP^\nu_\mu\delta_\beta B_{\alpha\,\nu}\,, \qquad E_{\alpha\,\mu} = G_{\alpha\,\mu\nu}u^\nu\,. \tag{254}$$

They govern the retarded two-point functions of currents at low frequency $\omega$ and zero wavenumber, e.g.

$$\langle J^{x\alpha}(\omega)J^{x\beta}(-\omega)\rangle = i\omega\sigma^{\alpha\beta} + O(\omega^2)\,, \qquad \langle J^{x\alpha}(\omega)J^{y\beta}(-\omega)\rangle = i\omega\bar{\sigma}^{\alpha\beta} + O(\omega^2)\,. \tag{255}$$

In this example, the symmetric part of $\sigma^{\alpha\beta}$ and the antisymmetric part of $\bar{\sigma}^{\alpha\beta}$ are dissipative. The second Law implies that the symmetric part of $\sigma^{\alpha\beta}$ is a non-negative matrix and that the antisymmetric part of $\bar{\sigma}^{\alpha\beta}$ vanishes. The Onsager relations imply that the symmetric parts of $\sigma^{\alpha\beta}$ and $\bar{\sigma}^{\alpha\beta}$ are CPT-even, and the antisymmetric part of $\sigma^{\alpha\beta}$ is CPT-odd [37–39].

Let us see how these statements arise from the effective action. The tensor term in the action which accounts for this transport is

$$L = \ldots - \frac{i}{2}\mathbb{T}\left(\Sigma^{\beta\alpha}\mathbb{P}^{ij} + \overline{\Sigma}^{\beta\alpha}\mathbb{e}^{kji}\mathbb{u}_k\right)D_\theta\mathbb{B}_{\alpha\,i}D_{\bar{\theta}}\mathbb{B}_{\beta\,j} + \ldots\,, \tag{256}$$

with $\mathbb{P}^{ij} = \mathbb{g}^{ij} + \mathbb{u}^i\mathbb{u}^j$ with $\mathbb{u}^i = \mathbb{T}\beta^i$ the super-velocity, and $\mathbb{e}^{ijk}$ the super-Levi-Civita tensor in three dimensions. This Lagrangian leads to the currents (253) with

$$\sigma^{\alpha\beta} = \frac{1}{2}\left(\Sigma^{\alpha\beta} + \widetilde{\Sigma}^{\beta\alpha}\right)\,, \qquad \bar{\sigma}^{\alpha\beta} = \frac{1}{2}\left(\overline{\Sigma}^{\alpha\beta} + \widetilde{\overline{\Sigma}}^{\beta\alpha}\right)\,. \tag{257}$$

Clearly the symmetric parts of $\sigma^{\alpha\beta}$ and $\bar{\sigma}^{\alpha\beta}$ are CPT-even, while the antisymmetric parts are CPT-odd. A quick computation shows that the symmetric part of $\sigma^{\alpha\beta}$ corresponds to

a KMS-odd term and its antisymmetric part to a KMS-even term. This is consistent with our earlier result that the CPT-eigenvalue of a transport coefficient is the product of the KMS parity of the term it appears in with the KMS-eigenvalue of the tensor structure it multiplies, as $\sigma^{\alpha\beta}$ multiples a KMS-odd tensor structure in (253). Similarly, the symmetric part of $\bar{\sigma}^{\alpha\beta}$ corresponds to a KMS-even term and the antisymmetric part to a KMS-odd term. As for Schwinger-Keldysh positivity, it implies that the symmetric part of $\sigma^{\alpha\beta}$ is non-negative and that the antisymmetric part of $\bar{\sigma}^{\alpha\beta}$ vanishes.

In this example the KMS-even transport is non-dissipative and the KMS-odd transport is dissipative. This is not quite the case when one goes beyond first-order hydrodynamics. While the KMS-even terms are always dissipationless, KMS-even transport coming from higher tensor terms contributes to $\text{Im}(S_{eff})$ as we saw in (244). Furthermore, while all KMS-odd transport contributes to $\text{Im}(S_{eff})$, not all of it is dissipative.

The 3-tensor terms can be similarly decomposed into KMS-even and KMS-odd parts. We decompose them as

$$J^A_{r\,(3)} = J^A_{+\,(3)} + J^A_{-\,(3)} \,, \tag{258}$$

where

$$J^A_{(3)\,+} = -\frac{1}{4}\nabla^{a\dagger}\left(N_+^{(AB\{i\}_{ab})}\nabla^b\delta_\beta F_{r\,B}\right)\,, \qquad J^A_{(3)\,-} = \frac{1}{4}\nabla^{a\dagger}\left(N_-^{[AB\{i\}_{ab}]}\nabla^b\delta_\beta F_{r\,B}\right)\,, \tag{259}$$

and the $N_\pm$ were defined in (236). Using (235), we find

$$J^A_{(3)\,\pm}(\nabla, F_r) = \pm\eta_A J^A_{(3)\,\pm}(\eta\nabla, \eta F_r)\,. \tag{260}$$

Note that $N_+^{(AB\{i\}_{ab})}$ is symmetric under an exchange of the $AB$ and $ab$ indices, is KMS-even, and contributes to dissipation, while $N_-^{[AB\{i\}_{ab}]}$ is antisymmetric under the same exchange of indices, is KMS-odd, does not contribute to entropy production, but is nevertheless constrained by Schwinger-Keldysh positivity.

## 6.2 Classification of hydrodynamic transport

A full classification of all possible transport coefficients according to their role in entropy production was carried out in [26, 27] using an off-shell reformulation of the second Law. The eight classes of transport described in [26, 27] include two types of scalars (hydrodynamic and hydrostatic), two types of transport coefficients which are associated with anomalies (referred to as anomalous transport terms and hydrostatic flux vectors), Berry-like transport, Hydrodynamic flux vectors, conserved entropy terms and dissipative terms.[9] Of all the classes, only the dissipative terms lead to entropy production. In what follows we will offer a complementary viewpoint on the classification of [26, 27] and show how different classes of transport arise naturally from the structure of the Schwinger-Keldysh effective action. Our classification separates the various transport coefficients into scalar terms, non-dissipative terms, dissipative terms, pseudo-dissipative terms and exceptional terms.

### 6.2.1 Scalar terms

Observe that all transport coefficients which arise from the scalar Lagrangian $L_0$ do not contribute to entropy production (242), nor to the imaginary part of $S_{eff}$. These terms

---

[9]The authors of [26, 27] also included a class of hydrostatically forbidden terms in their classification. This class is comprised of expressions which will not appear in the constitutive relations. The absence of such terms is naturally incorporated in the effective action formulation since the Schwinger-Keldysh partition function reduces to the hydrostatic partition function [6, 7] in the appropriate limit.

are referred to as scalar terms in the classification of [26,27]. The authors of [26,27] further separate these into two classes which they refer to as $H_S$ and $\bar{H}_S$, differing in whether they contribute when the system is in hydrostatic equilibrium. Hydrostatic equilibrium is characterized by the existence of a timelike Killing vector [6, 7]. In the current context, the hydrostatic limit is obtained by identifying the timelike Killing vector with $\beta^i$, taking the $a$-fields to vanish, and taking the $r$-sources to be time-independent. Thus, terms in $L_0$ which do not vanish when $\delta_\beta = 0$ are $H_S$ and terms which do vanish are $\bar{H}_S$.

The simplest scalar term is the action for an ideal fluid given in Section 4, which we briefly reproduce here for convenience. The most general scalar Lagrangian at zeroth order in derivatives is

$$\frac{1}{2}\left(L_0 + \widetilde{L}_0\right) = G(T, \nu) + G(T, -\nu) = P(T, \nu). \tag{261}$$

The constitutive relations resulting from this Lagrangian are the ideal stress-energy tensor and $U(1)$ current

$$
\begin{aligned}
T_r^{ij} &= P\left(g_r^{ij} + u^i u^j\right) + \left(\left(\frac{\partial P}{\partial T}\right)_\mu T + \left(\frac{\partial P}{\partial \mu}\right)_T \mu - P\right) u^i u^j, \\
J_r^i &= \left(\frac{\partial P}{\partial \mu}\right)_T u^i,
\end{aligned}
\tag{262}
$$

where, as usual, we have defined the rescaled velocity $u^i = T\beta^i$. The entropy current is given by

$$S^i = \left(\frac{\partial P}{\partial T}\right)_\mu u^i \tag{263}$$

and satisfies

$$\nabla_i S^i = 0 \tag{264}$$

on-shell.

### 6.2.2 Non-dissipative tensor terms

Recall from (244) that $\mathrm{Im}(S_{eff})$ is determined by the KMS-odd part of the 2-tensor terms, as well as both the KMS-even and KMS-odd parts of the 3-tensor and higher-tensor terms. Moreover, the entropy production (242) arises from KMS-odd 2-tensor transport and from KMS-even 3-tensor terms. Thus the KMS-even part of 2-tensor terms,

$$J_{r\,(2)}^A = -\frac{1}{2}\nabla^{a\dagger}\left(L_+^{[AB\{i\}ab]}\nabla^b \delta_\beta F_{r\,B}\right), \qquad L_+^{[AB\{i\}ab]} = \frac{1}{2}\left(L_+^{AB\{i\}ab} - L_+^{BA\{i\}ba}\right), \tag{265}$$

is the unique part of the tensor terms which is not subject to the Schwinger-Keldysh positivity constraint $\mathrm{Im}(S_{eff}) \geq 0$. We refer to these terms as non-dissipative tensor terms.

Using (240), we may write

$$L_+^{[AB\{i\}ab]} = L_+^{[AB]\{i\}(ab)} + L_+^{(AB)\{i\}[ab]}. \tag{266}$$

In the language of [26, 27], Berry-type terms are closest to $L_+^{[AB]\{i\}(ab)}$ and $\bar{H}_V$-terms are closest to $L_+^{(AB)\{i\}[ab]}$ terms. Note that both of these terms are non-dissipative and KMS-even, similar to the scalar terms described in the previous Subsection. We will refer to terms of the form $L_+^{[AB]\{i\}(ab)}$ as non-dissipative antisymmetric and to terms of the form $L_+^{(AB)\{i\}[ab]}$ as non-dissipative symmetric.

The most general constitutive relations for the $L_+^{[AB]\{i\}(ab)}$ type terms take the form

$$J_{r\,(2)}^A = -\frac{1}{4}\nabla^{a\dagger}\left(L_+^{[AB]\{i\}ab}\nabla^b\delta_\beta F_{r\,B}\right) - \frac{1}{4}\nabla^{a\dagger}\left(L_+^{[AB]\{i\}ba}\nabla^b\delta_\beta F_{r\,B}\right). \tag{267}$$

These relations can be simplified for special configurations. In the case where $a = b = 0$, i.e., there are no derivatives acting on either $D_\theta \mathbb{F}_A$ or on $D_{\bar\theta}\mathbb{F}_B$ in the action, we find

$$J_{r\,(2)}^A = -\frac{1}{2}L_+^{[AB]}\delta_\beta F_{r\,B}. \tag{268}$$

If $a + b = 1$ we obtain

$$J_{r\,(2)}^A = \frac{1}{4}(\nabla_i L_+^{[AB]i})\delta_\beta F_{r\,B}. \tag{269}$$

Other values of $a$ and $b$ do not seem to take a particularly simple form.

The constitutive relations for the $L_+^{(AB)\{i\}[ab]}$ type terms take a form similar to (267)

$$J_{r\,(2)}^A = -\frac{1}{4}\nabla^{a\dagger}\left(L_+^{(AB)\{i\}ab}\nabla^b\delta_\beta F_{r\,B}\right) + \frac{1}{4}\nabla^{a\dagger}\left(L_+^{(AB)\{i\}ba}\nabla^b\delta_\beta F_{r\,B}\right). \tag{270}$$

This expression vanishes when $a = b = 0$ and takes the form

$$J_{r\,(2)}^A = \frac{1}{4}(\nabla_i L_+^{(AB)i})\delta_\beta F_{r\,B} + \frac{1}{2}L_+^{(AB)i}\nabla_i\delta_\beta F_{r\,B} \tag{271}$$

for $a + b = 1$. Otherwise, we have not been able to bring (270) to a particularly simple form.

We presented two examples of antisymmetric non-dissipative transport in Subsection 6.1 in the form of the antisymmetric part of the ordinary conductivity $\sigma^{\alpha\beta}$ as well as the matrix of Hall conductivities. We briefly reprise the Hall conductivity term for a single $U(1)$ current. This transport arises from the Lagrangian

$$L = \ldots - \frac{i}{2}\overline{\Sigma}(\mathbb{T},\nu)\,\mathfrak{e}^{kji}\mathfrak{u}_k D_\theta\mathbb{B}_i D_{\bar\theta}\mathbb{B}_j, \tag{272}$$

leading to

$$L_+^{[ij]} = -2\bar\sigma^+\epsilon^{ijk}u_k, \qquad \bar\sigma^+ = \frac{1}{2}\left(\overline{\Sigma}(T,\nu) + \overline{\Sigma}(T,-\nu)\right), \tag{273}$$

which yields

$$J_{r\,(2)}^i = \bar\sigma^+\epsilon^{ijk}u_j V_k, \tag{274}$$

where $V_k$ was defined in (254). The entropy current is modified as

$$S^i = \ldots - \frac{\mu}{T}J_{r\,(2)}^i, \tag{275}$$

but its divergence is unmodified.

An example of symmetric non-dissipative transport is

$$L = \ldots \frac{i}{2}\chi(\mathbb{T},\nu)\mathbb{P}^{ij}\mathfrak{u}^k\left(\mathbb{W}_k D_\theta\mathbb{B}_i D_{\bar\theta}\mathbb{B}_j - D_\theta\mathbb{B}_i \mathbb{W}_k D_{\bar\theta}\mathbb{B}_j\right)\ldots, \tag{276}$$

where $\mathbb{P}^{ij} = \mathfrak{g}^{ij} + \mathfrak{u}^i\mathfrak{u}^j$, which yields

$$L_+^{(ij)k} = 2\chi^+ P^{ij}u^k, \tag{277}$$

with $\chi^+(T,\nu) = \frac{1}{2}\left(\chi(T,\nu) + \chi(T,-\nu)\right)$. The corresponding current is

$$J_{r\,(2)}^i = -\frac{1}{2T}\nabla_k(\chi^+ P^{ij}u^k)\left(E_j - T\partial_j\nu\right) - \chi^+ P^{ij}u^k\nabla_k\left(\frac{E_j}{T} - \partial_j\nu\right), \tag{278}$$

and the entropy current is modified to be

$$S^i = \ldots - \frac{\mu}{T}J_{r\,(2)}^i + \frac{1}{2}\chi^+ u^i\left(\frac{E_j}{T} - \partial_j\nu\right)^2. \tag{279}$$

One may check that $\chi^+$ does not contribute to entropy production.

### 6.2.3 Dissipative terms

Recall that the divergence of the entropy current (242) is given by

$$\nabla_i S^i = -\frac{1}{2}\left(M_-^{(AB\{i\}ab)} - \frac{1}{2}M_+^{(AB\{i\}ab)}\right)(\nabla^a \delta_\beta F_{rA})(\nabla^b \delta_\beta F_{rB})\Big|_{\text{on-shell}} \geq 0\,, \qquad (280)$$

where we have used the decompositions (246), (248) and the definitions (236). Using (240), we may write

$$M_\pm^{(AB\{i\}ab)} = M_\pm^{(AB)\{i\}(ab)} + M_\pm^{[AB]\{i\}[ab]}\,. \qquad (281)$$

We will refer to $M_\pm^{(AB)\{i\}(ab)}$ and $M_\pm^{[AB]\{i\}[ab]}$ as dissipative symmetric or dissipative antisymmetric respectively. The former corresponds to the dissipative transport defined in [26, 27]. Notice also that at first order in derivatives, only KMS-odd dissipative transport $M_-^{(AB\{i\}ab)}$ is allowed. KMS-even dissipative contributions appear at second order in derivatives and onwards.

The constitutive relations for $M_-^{(AB)\{i\}(ab)}$ and $M_-^{[AB]\{i\}[ab]}$ are similar in structure to those of KMS-even terms, viz.,

$$J_{r(2)}^A = \frac{1}{4}\nabla^{a\dagger}\left(M_-^{(AB)\{i\}ab}\nabla^b \delta_\beta F_{rB}\right) + \frac{1}{4}\nabla^{a\dagger}\left(M_-^{(AB)\{i\}ba}\nabla^b \delta_\beta F_{rB}\right)\,, \qquad (282)$$

and

$$J_{r(2)}^A = \frac{1}{4}\nabla^{a\dagger}\left(M_-^{[AB]\{i\}ab}\nabla^b \delta_\beta F_{rB}\right) - \frac{1}{4}\nabla^{a\dagger}\left(M_-^{[AB]\{i\}ba}\nabla^b \delta_\beta F_{rB}\right)\,. \qquad (283)$$

If $a = b = 0$ the former simplifies to

$$J_{r(2)}^A = \frac{1}{2}M_-^{(AB)}\delta_\beta F_{rB}\,, \qquad (284)$$

and if $a + b = 1$ we obtain

$$J_{r(2)}^A = -\frac{1}{4}(\nabla_i M_-^{(AB)i})\delta_\beta F_{rB}\,, \qquad (285)$$

or

$$J_{r(2)}^A = -\frac{1}{4}(\nabla_i M_-^{[AB]i})\delta_\beta F_{rB} - \frac{1}{2}M_-^{[AB]i}\nabla_i \delta_\beta F_{rB}\,. \qquad (286)$$

Similar expressions arise for $M_+^{(AB)\{i\}(ab)}$ and $M_+^{[AB]\{i\}[ab]}$. For example, for $a = b = 0$, we have

$$J_{r(3)}^A = -\frac{1}{4}M_+^{(AB\{i\}ab)}\delta_\beta F_{rB}\,. \qquad (287)$$

In Subsection 6.1 we gave an example of symmetric dissipative transport, namely the ordinary conductivity of a charged fluid. We reprise it here. The Lagrangian is

$$L = \ldots - \frac{i}{2}\Sigma(\mathbb{T}, \nu)\mathbb{T}\mathbb{P}^{ij}D_\theta \mathbb{B}_i D_{\bar\theta}\mathbb{B}_j\,, \qquad (288)$$

which leads to

$$M_-^{(ij)} = -2\sigma^+ T P^{ij}\,, \qquad \sigma^+ = \frac{1}{2}\left(\Sigma(T, \nu) + \Sigma(T, -\nu)\right)\,, \qquad (289)$$

and

$$J_{r(2)}^i = \sigma^+ V^i\,, \qquad (290)$$

where $V^i$ was defined in (254). The entropy current becomes

$$S^i = \ldots - \frac{\mu}{T} J^i_{r(2)}\,, \tag{291}$$

and its on-shell production is

$$\nabla_i S^i = \ldots + \frac{\sigma^+}{T} V^2\,. \tag{292}$$

Schwinger-Keldysh positivity and positivity of entropy production both imply $\sigma^+ \geq 0$.

We also give an example of antisymmetric dissipative transport, which appears at second order and higher in the gradient expansion. Consider a parity-violating theory with a single $U(1)$ current in two spatial dimensions, and a Lagrangian which, apart from the ideal pressure term includes the 2-tensor term

$$L = \ldots + \frac{i}{2}\xi(\mathbb{T}, \nu)\mathbb{e}^{ijk}\mathbb{u}_k u^l \left(\mathbb{W}_l D_\theta \mathbb{B}_i D_{\bar\theta}\mathbb{B}_j - D_\theta \mathbb{B}_i \mathbb{W}_l D_{\bar\theta}\mathbb{B}_j\right)\,, \tag{293}$$

where $\mathbb{u}^i = \mathbb{T}\beta^i$ is the velocity. This gives

$$M^{[ij]l}_- = 2\xi^+ \epsilon^{ijk} u_k u^l\,, \tag{294}$$

with $\xi^+(T, \nu) = \frac{1}{2}\left(\xi(T, \nu) + \xi(T, -\nu)\right)$, and the corresponding constitutive relations

$$J^i_{r(2)} = \frac{1}{2T}\nabla_l\left(\xi^+ \epsilon^{ijk} u_k u^l\right)(E_j - T\partial_j\nu) + \xi^+ \epsilon^{ijk} u_k u^l \nabla_l\left(\frac{E_j}{T} - \partial_j\nu\right)\,. \tag{295}$$

The entropy current is

$$S^i = su^i - \nu\left(\frac{1}{2T}\nabla_l\left(\xi^+\,\epsilon^{ijk} u_k u^l\right)(E_j - T\partial_j\nu) + \xi^+\,\epsilon^{ijk} u_k u^l \nabla_l\left(\frac{E_j}{T} - \partial_j\nu\right)\right)\,, \tag{296}$$

and its divergence is

$$\nabla_i S^i = -\frac{1}{2}\xi^+\,\epsilon^{ijk} u_k u^l \left(\nabla_l \delta_\beta B_{r\,i}\,\delta_\beta B_{r\,j} - \delta_\beta B_{r\,i}\,\nabla_l \delta_\beta B_{r\,j}\right) \tag{297}$$

on-shell. If there were no other contributions to the Lagrangian, i.e. assuming that the ordinary conductivity vanishes, positivity of entropy production implies that $\xi^+$ must vanish. If the conductivity is nonzero, the modified right-hand side of (297) may be perturbatively organized in a quadratic form. Within the gradient expansion, the positivity of entropy production is ensured by the positivity of the ordinary conductivity, with no constraint on $\xi^+$. See e.g. [23, 25, 29].

### 6.2.4 Pseudo-dissipative and exceptional terms

The remaining transport coefficients in our classification are $P^{(AB\{i\}_{ab})}_{\pm}$, and the KMS-odd 3-tensor terms $N^{[AB\{i\}_{ab}]}_-$. Both types of transport do not contribute to entropy production. We call the former pseudo-dissipative, since its index structure is identical to dissipative transport, and the latter exceptional.

As an example of exceptional transport consider a Lagrangian which has the contribution

$$\begin{aligned}
L = \ldots &- \frac{i}{4}\gamma(\mathbb{T}, \nu)\left(\mathbb{P}^{m(i}\mathbb{P}^{j)n}\mathbb{P}^{kl} - \mathbb{P}^{ij}\mathbb{P}^{k(m}\mathbb{P}^{n)l}\right)\delta_\beta\mathfrak{g}_{mn} D_\theta\mathfrak{g}_{ij} D_{\bar\theta}\mathfrak{g}_{kl} \\
&- \frac{1}{2}\gamma(\mathbb{T}, \nu)\left(\mathbb{P}^{m(i}\mathbb{P}^{j)n}\mathbb{P}^{kl} - \mathbb{P}^{ij}\mathbb{P}^{k(m}\mathbb{P}^{n)l}\right)D_\theta\mathfrak{g}_{ij} D_{\bar\theta}\mathfrak{g}_{kl} D\mathfrak{g}_{mn} + \ldots\,.
\end{aligned} \tag{298}$$

A simple computation yields

$$N_-^{[(ij)(kl)]} = 2\gamma^- \left( P^{m(i}P^{j)n}P^{kl} - P^{ij}P^{k(m}P^{n)l} \right) \delta_\beta g_{r\,mn} \,, \tag{299}$$

so that the constitutive relations are

$$T_r^{ij} = \ldots - \frac{\gamma^-}{T^2}(P^{ij}\sigma^2 - 2\Theta\sigma^{ij}) \,, \tag{300}$$

where $\sigma^{ij}$ and $\Theta$ are the shear and expansion,

$$\sigma^{ij} = P^{im}P^{jn}\left( \nabla_m u_n + \nabla_n u_m - \frac{2}{(d-1)}\Theta g_{mn} \right) \,, \qquad \Theta = \nabla_i u^i \,, \tag{301}$$

and $\sigma^2 = \sigma_{ij}\sigma^{ij}$.

A straightforward computation shows that $\gamma^-$ contributes neither to the entropy current nor to entropy production. Thus the second Law does not constrain it. However, Schwinger-Keldysh positivity (244) does. If the fluid Lagrangian was given by a pressure term, the highlighted terms in (298), and nothing else (i.e. if the viscosities vanish), then Schwinger-Keldysh positivity would become the statement

$$\text{Im}(S_{eff}) = +\frac{1}{2}\int d^d\sigma\sqrt{-g_r}\, g_{a\,ij}\delta_\beta g_{r\,kl}N_-^{[(ij)(kl)]}\bigg|_{\delta_\beta g_{r\,mn}\to g_{a\,mn}} \geq 0 \,, \tag{302}$$

or more explicitly

$$-\int d^d\sigma\sqrt{-g_r}\frac{\gamma^-}{T}\left( P^{ij}\sigma^{mn} + 2\Theta\left( \frac{1}{(d-1)}P^{ij}P^{mn} - P^{m(i}P^{j)n} \right) \right)g_{a\,ij}g_{a\,mn} \geq 0 \,. \tag{303}$$

Since the quantity in (303) does not have a definite sign, the only way the inequality (303) can be satisfied is to set $\gamma^- = 0$.

Our example serves as a demonstration that the Schwinger-Keldysh positivity condition can enforce constraints on transport which the entropy current is indifferent to. Since we have set the shear and bulk viscosities in this example to zero (or, at least, to be perturbatively small) it probes the very edges of allowed parameter space. Recall that within the gradient expansion, once the divergence of the entropy current is arranged into a complete square, then its positivity is ensured by the positivity of the aforementioned viscosities. If we restrict ourselves to configurations which satisfy the equations of motion and have small gradients, then the latter condition is necessary and sufficient to ensure that the entropy production will be non negative. In contrast, the Schwinger-Keldysh positivity condition must hold for any field configuration, so while a solution which is valid in a perturbative gradient expansion is a necessary condition for positivity, it is not a sufficient one. Regardless of this distinction, it seems that unless the derivative expansion truncates it is impractical to attempt to solve the Schwinger-Keldysh positivity condition in its entirety.

As an example of pseudo-dissipative transport let us consider a Lagrangian which includes the pressure term and the 2-tensor term

$$L = \ldots + \frac{i}{2}\zeta(\mathbb{T},\nu)\left( \mathbb{u}^i\mathbb{c}^{jkl}\mathbb{u}_l + \mathbb{u}^j\mathbb{c}^{ikl}\mathbb{u}_l \right)\delta_\beta\mathbb{B}_k D_\theta\mathbb{B}_i D_{\bar\theta}\mathbb{B}_j + \ldots \,, \tag{304}$$

which leads to a KMS-odd contribution to the constitutive relations

$$P_-^{(ij)} = 2\zeta^-\left( u^i\epsilon^{jkl}u_l + u^j\epsilon^{ikl}u_l \right)\delta_\beta B_{r\,k} \,, \qquad \zeta^- = \frac{1}{2}(\zeta(T,\nu) - \zeta(T,-\nu)) \,. \tag{305}$$

This contribution is pseudo-dissipative, on account of

$$P_-^{ij}\delta_\beta B_{r\,i}\delta_\beta B_{r\,j} = 0\,, \tag{306}$$

and so $\zeta^-$ does not contribute to entropy production. The resulting flavor current is

$$J_{r(2)}^i = \frac{1}{2}P_-^{(ij)}\delta_\beta B_{r\,j} = \zeta^-(u^l\partial_l\nu)\epsilon^{ijk}u_jV_k\,, \tag{307}$$

where $V_i$ was defined in (254). While $\zeta^-$ does not contribute to entropy production, it is constrained by Schwinger-Keldysh positivity, which in the absence of any other tensor terms in the Lagrangian reads

$$\text{Im}(S_{eff}) = -\int d^d\sigma\,\zeta^-\left(u^i\epsilon^{jkl}u_l + u^j\epsilon^{ikl}u_l\right)\delta_\beta B_{r\,k}B_{a\,i}B_{a\,j} \geq 0\,. \tag{308}$$

As in our example of exceptional transport, the integrand does not have a definite sign and the only way to satisfy the inequality (308) is to set $\zeta^- = 0$.

### 6.2.5 Additional conserved currents

By definition, we may always add to the entropy current trivially conserved currents $J_C^i$ which appear at least at first order in derivatives. The modified entropy current will still have non-negative divergence and will be proportional to the entropy density at zeroth order in derivatives. In the effective action these contributions to the entropy current may be generated in the following way.

Consider a trivially conserved super-current $\mathbb{J}_C^i$ whose bottom component is $J_C^i$. An example of such a current in $2+1$-dimensions is $\mathbb{J}_C^i = \mathfrak{e}^{ijk}\mathbb{G}_{jk}$. More generally such a current will always locally take the form $\mathbb{J}_C^i = \mathfrak{e}^{ii_1\dots i_{d-1}}\partial_{i_1}\mathbb{V}_{i_2\dots i_{d-1}}$. Consider the redefined action

$$\int d^d\sigma d\theta d\bar\theta\sqrt{-\mathfrak{g}}\,L^{(A)} \to \int d^d\sigma d\theta d\bar\theta\sqrt{-\mathfrak{g}}\left(L^{(A)} - \frac{1}{2}\mathbb{A}_i\mathbb{J}_C^i + O(\mathbb{A}^2)\right). \tag{309}$$

The redefined action is invariant under $\delta_T$ when setting $\mathbb{A}_i$ to vanish due to the conservation of $\mathbb{J}_C^i$. If $\mathbb{A}_i\mathbb{J}_C^i$ is KMS-odd, meaning $K(\mathbb{A}_i\mathbb{J}_C^i) = -\widetilde{\mathbb{A}}_i\widetilde{\mathbb{J}}_C^I$ (as is $\mathfrak{e}^{ijk}\mathbb{A}_i\mathbb{G}_{jk}$), then this redefinition disappears after adding the KMS partner term. If, however, it is KMS-even, then the KMS partner term contributes in the same way as the original and the full effective action is redefined as

$$S_{eff} \to S_{eff} - \int d^d\sigma d\theta d\bar\theta\sqrt{-\mathfrak{g}}\,\mathbb{A}_i\mathbb{J}_C^i + O(\mathbb{A}^2)\,. \tag{310}$$

This redefinition does not affect the constitutive relations of the currents since the latter are evaluated at $\mathbb{A}_i = 0$. On the other hand the bottom component of the entropy current is redefined as

$$S^i \to S^i + J_C^i\,, \tag{311}$$

which is the modification we were after.

If the current $\mathbb{J}_C^i$ can be written as $\mathbb{J}_C^i = \mathfrak{e}^{ii_1\dots i_{d-1}}\partial_{i_1}\mathbb{V}_{i_2\dots i_{d-1}}$ globally, then this modification to the entropy current is trivial. The total entropy is unmodified. If however $\mathbb{V}$ can only be written this way locally, as for $\mathbb{J}_C^i = \mathfrak{e}^{ijk}\mathbb{G}_{jk}$ when space is compact and there is a net flux through it, then the total entropy is modified by this term. This latter, globally non-trivial, redefinition of the entropy current is class $C$ transport in the nomenclature of [26, 27]. The physical interpretation given there is that this transport quantifies topological shifts to ground state degeneracy, as one finds in fractional quantum Hall states.

### 6.2.6 Anomaly-induced transport

The two remaining classes of transport in the "eight-fold way" of [26, 27] are related to
't Hooft anomalies. In their nomenclature, class $A$ transport refers to those transport
coefficients which are directly governed by 't Hooft anomalies for continuous symmetries,
in that the Second Law fully determines class $A$ transport in terms of anomaly coefficients.
Class $H_V$ transport, like the $T^2$ contribution to the chiral vortical effect in four dimensions,
is described by those transport coefficients which are not tied to anomalies by the Second
Law, but which are nevertheless governed by anomalies for many systems [28, 40–43].

A proper discussion of anomalies and anomaly-induced transport would take us some-
what far afield. However, building upon the results of [36,44], it is straightforward using the
inflow mechanism to construct effective actions for anomaly-induced transport. We write
down the effective action for class $A$ transport in Appendix F for any 't Hooft anomaly,
and using [45], those results can be easily generalized to account for $H_V$ transport as well.

## 7 More examples

In Section 4 we worked out the effective action and constitutive relations for an ideal fluid.
In this Section we consider two more examples. The first is the first-order hydrodynam-
ics of a parity-violating fluid in 2+1-dimensions [9], and the second is the second-order
hydrodynamics of a neutral, parity-preserving fluid in any dimension [29].

To proceed, we must construct the most general possible expressions for $L_0$, $L^{AB}$,
and $L^{ABC}$ which is compatible with the symmetries of the problem (e.g., coordinate/$U(1)$
invariance, and possibly parity). While there are few such terms when working at a
low order in the derivative expansion, the number of possible expressions grows with
the number of derivatives, turning the classification of allowed terms in the Lagrangian
into a formidable task. Luckily, there are a few simplifying considerations which we
can use in order to minimize the number of independent terms in the Lagrangian. These
considerations are not new and should be familiar to practitioners of hydrodynamics. They
include using the equations of motion at lower order in the gradient expansion to simplify
the higher order constitutive relations, and utilizing frame transformations in order to
remove ambiguities associated with out of equilibrium definitions of thermodynamic fields.
Let us spell these out in some detail.

In the context of hydrodynamics we can use the equations of motion at lower order
in the derivative expansion to show that some tensor structures are equivalent on-shell.
Consider the hydrodynamic equations of motion,

$$\nabla_\nu T^{\mu\nu} = G^\mu{}_\nu J^\nu\,, \qquad \nabla_\mu J^\mu = 0\,, \tag{312}$$

with $G_{\mu\nu}$ the field strength associated with the flavor field conjugate to the $U(1)$ current
$J^\mu$. At leading order in derivatives the constitutive relations for the stress tensor and
current are those of ideal hydrodynamics, given in (162). Inserting these into the equations
of motion, we find, e.g.,

$$0 = \nabla_\mu J^\mu = u^\mu \partial_\mu \rho + \rho \nabla_\mu u^\mu + O(\partial^2)\,, \tag{313}$$

and similar equations when considering the constitutive relations for the stress tensor.

With some slight manipulations they can be put into the form

$$\left(\frac{\partial\epsilon}{\partial T}\right)_{\nu}(u^{\mu}\partial_{\mu}T)+\left(\frac{\partial\epsilon}{\partial\nu}\right)_{T}(u^{\mu}\partial_{\mu}\nu)-(\epsilon+P)\nabla_{\mu}u^{\mu}=O(\partial^{2})\,,$$

$$\left(\frac{\partial\rho}{\partial T}\right)_{\nu}(u^{\mu}\partial_{\mu}T)+\left(\frac{\partial\rho}{\partial\nu}\right)_{T}(u^{\mu}\partial_{\mu}\nu)+\rho\nabla_{\mu}u^{\mu}=O(\partial^{2})\,, \qquad (314)$$

$$a^{\mu}+\frac{P^{\mu\nu}\partial_{\nu}T}{T}-\frac{\rho}{\epsilon+P}V^{\mu}=O(\partial^{2})\,,$$

where $V_{\mu}$ was defined in (254) ($V_{\mu}=E_{\mu}-TP_{\mu}^{\nu}\partial_{\nu}\nu$ with $E_{\mu}=G_{\mu\nu}u^{\nu}$), and $a^{\mu}=u^{\nu}\nabla_{\nu}u^{\mu}$. These relations imply that, on-shell, only one of the three a priori independent one-derivative scalars ($u^{\mu}\partial_{\mu}T, u^{\nu}\partial_{\nu}\nu, \nabla_{\rho}u^{\rho}$) are independent of each other, and correspondingly that the vector $Ta^{\mu}+P^{\mu\nu}\partial_{\nu}T$ may be eliminated in favor of $V^{\mu}$.

Next, we can use field redefinitions to remove some terms from the constitutive relations. Recall that the constitutive relations are inherently ambiguous. Within the derivative expansion one may use a field redefinition of the hydrodynamic variables, e.g.,

$$T \to T + O(\partial)\,, \qquad (315)$$

which leaves the leading order constitutive relations invariant, but modifies them at first and higher order in the derivative expansion. Like all field redefinitions, this change modifies the equations of motion but not physical observables such as correlation functions of the stress tensor or current. The field redefinition ambiguity is well-known in the context of hydrodynamics and is often referred to as a choice of frame [1]. Perhaps the most commonly used frame is the Landau frame, where the hydrodynamic variables are fixed by the conditions

$$u^{\nu}T_{\mu\nu}=-\epsilon\, u_{\mu}\,, \qquad u^{\mu}J_{\mu}=-\rho\,, \qquad (316)$$

where $\epsilon$ is the thermodynamic energy density satisfying $\epsilon=-P+T\left(\frac{\partial\rho}{\partial T}\right)_{\mu}+\mu\left(\frac{\partial P}{\partial\mu}\right)_{T}$ with $P$ the pressure and the charge density is $\rho=\left(\frac{\partial P}{\partial\mu}\right)_{T}$. (See (163).)

More operatively, we can begin with the most general frame and decompose the stress tensor and current into components parallel to, and transverse to the velocity,

$$J^{\mu}=\mathcal{N}u^{\mu}+\nu^{\mu}\,,$$
$$T^{\mu\nu}=\mathcal{E}u^{\mu}u^{\nu}+\mathcal{P}P^{\mu\nu}+u^{\mu}q^{\nu}+u^{\nu}q^{\mu}+\tau^{\mu\nu}\,, \qquad (317)$$

where $\mathcal{N}=\rho+O(\partial), \mathcal{E}=\epsilon+O(\partial), \mathcal{P}=P+O(\partial), u^{\mu}q_{\mu}=u^{\mu}\nu_{\mu}=P^{\mu\nu}\tau_{\mu\nu}=0, u^{\mu}\tau_{\mu\nu}=0$, and $(\nu^{\mu}, q^{\nu}, \tau^{\rho\sigma})$ are all at least first order in derivatives. Redefining the hydrodynamic variables as

$$\mu \to \mu+\delta\mu\,, \qquad T \to T+\delta T\,, \qquad u^{\mu} \to u^{\mu}+\delta u^{\mu}\,, \qquad (318)$$

with

$$\begin{pmatrix} \partial_{\nu}\rho & \partial_{T}\rho \\ \partial_{\nu}\epsilon & \partial_{T}\epsilon \end{pmatrix} \begin{pmatrix} \delta\mu \\ \delta T \end{pmatrix} = -\begin{pmatrix} \mathcal{N}-\rho \\ \mathcal{E}-\epsilon \end{pmatrix}\,, \qquad (319)$$

and

$$\delta u^{\mu}=-q^{\mu} \qquad (320)$$

will bring us into the Landau frame (i.e. set the quantities $\mathcal{N}-\rho, \mathcal{E}-\epsilon$, and $q^{\mu}$ to vanish at first order in derivatives),

$$u^{\nu}T_{\mu\nu}=-\epsilon\, u_{\mu}+O(\partial^{2})\,, \qquad u^{\mu}J_{\mu}=-\rho+O(\partial^{2})\,. \qquad (321)$$

Once the constitutive relations have been brought to Landau frame at first order in derivatives, reiterating this procedure allows one to enforce the Landau frame condition (316) to all orders in gradients.

In our Schwinger-Keldysh effective theory we may also use the leading order equations of motion to eliminate unphysical terms from the action and super-constitutive relations. Recall that deforming the action by a term proportional to the equations of motion does not affect any observable. The super-equations of motion are

$$\mathbb{W}_j \mathbb{T}^{ij} = \mathbb{G}^i{}_j \mathbb{J}^j \,, \qquad \mathbb{W}_i \mathbb{J}^i = 0 \,. \tag{322}$$

In Section 4 we found the super-stress tensor and super-current at leading order in derivatives

$$\mathbb{T}^{ij} = \epsilon \mathbb{u}^i \mathbb{u}^j + P \mathbb{P}^{ij} + O(\partial) \,, \qquad \mathbb{J}^i = \rho \mathbb{u}^i + O(\partial) \,. \tag{323}$$

The equations of motion at leading order in derivatives are then supersymmetrized versions of the ordinary equations of ideal hydrodynamics (314),

$$\left(\frac{\partial \epsilon}{\partial \mathbb{T}}\right)_\nu (\mathbb{u}^i \partial_i \mathbb{T}) + \left(\frac{\partial \epsilon}{\partial \nu}\right)_\mathbb{T} (\mathbb{u}^i \partial_i \nu) - (\epsilon + P)\mathbb{W}_i \mathbb{u}^i = O(\partial^2) \,,$$
$$\left(\frac{\partial \rho}{\partial \mathbb{T}}\right)_\nu (\mathbb{u}^i \partial_i \mathbb{T}) + \left(\frac{\partial \rho}{\partial \nu}\right)_\mathbb{T} (\mathbb{u}^i \partial_i \nu) - \rho \mathbb{W}_i \mathbb{u}^i = O(\partial^2) \,, \tag{324}$$
$$\mathbb{a}^i + \frac{\mathbb{P}^{ij}\partial_j \mathbb{T}}{\mathbb{T}} - \frac{\rho}{\epsilon + P}\mathbb{V}^i = O(\partial^2) \,,$$

with

$$\mathbb{a}^i = \mathbb{u}^j \mathbb{W}_j \mathbb{u}^i \,, \qquad \mathbb{V}_i = \mathbb{G}_{ij}\mathbb{u}^j - \mathbb{T}\mathbb{P}_i{}^j \partial_j \nu \,, \tag{325}$$

and we remind the reader that $\epsilon$, $P$ and $\rho$ are functions of $\mathbb{T}$ and $\nu$. Within the action we may then eliminate the a priori independent scalars $\mathbb{u}^i \partial_i \mathbb{T}$ and $\mathbb{u}^i \partial_i \nu$ in favor of $\mathbb{W}_i \mathbb{u}^i$ and so on.

One might also be tempted to use superfield redefinitions to eliminate couplings from the effective action. Redefining

$$\mathbb{X}^\mu \to \mathbb{X}^\mu + \delta\mathbb{X}^\mu \,, \qquad \mathbb{C} \to \mathbb{C} + \delta\mathbb{C} \,, \tag{326}$$

where $\delta\mathbb{X}^\mu$ and $\delta\mathbb{C}$ are at least zeroth order in derivatives, we find that the change in the action is of the form

$$S_{eff} \to S_{eff} - \int d^d\sigma d\theta d\bar\theta \sqrt{-\mathfrak{g}} \Big\{ (\mathbb{W}_\nu \mathbb{T}^{\mu\nu} - \mathbb{G}^\mu{}_\nu \mathbb{J}^\nu)\,\delta\mathbb{X}_\mu + (\mathbb{W}_i \mathbb{J}^i)\delta\mathbb{C} \Big\} \,, \tag{327}$$

where $\mathbb{T}^{\mu\nu} = \mathbb{T}^{ij}\partial_i \mathbb{X}^\mu \partial_j \mathbb{X}^\nu$, $\mathbb{W}_\mu = (\partial_i \mathbb{X}^\mu)^{-1}\mathbb{W}_i$, and so on. However, as we will see shortly, it seems that these redefinitions are not useful—shifting the action by a term proportional to the equation of motion and then using the equation of motions to eliminate redundant tensor data will bring us back to our starting point.

Let us attempt to put these tools to work. Consider for definiteness the most general scalar Lagrangian for a parity-preserving fluid at first order in derivatives,

$$L_0 = G + p_1(\mathbb{u}^i \partial_i \mathbb{T}) + p_2(\mathbb{u}^i \partial_i \nu) + p_3 \mathbb{W}_i \mathbb{u}^i + O(\partial^2) \,, \tag{328}$$

where the $p_i$ are functions of $\mathbb{T}$ and $\nu$. The leading order equations of motion (324) may be used to eliminate the two scalars $\mathbb{u}^i \partial_i \nu$ and $\mathbb{W}_i \mathbb{u}^i$ in favor of $\mathbb{u}^i \partial_i \mathbb{T}$. The most general scalar Lagrangian becomes

$$L_0 = G + p_1(\mathbb{u}^i \partial_i \mathbb{T}) + O(\partial^2) \,, \tag{329}$$

and the scalar part of the effective action becomes

$$S_{eff} = \int d^d\sigma d\theta d\bar{\theta}\sqrt{-\mathbb{g}}\left(P + p_1^-(\mathbb{u}^i\partial_i\mathbb{T}) + (\text{tensor terms}) + O(\partial^2)\right),\qquad(330)$$

with $p_1^- = \frac{1}{2}(p_1(\mathbb{T},\nu) - p_1(\mathbb{T},-\nu))$. Consider the field redefinition

$$\mathbb{X}^\mu \to \mathbb{X}^\mu + \alpha\mathbb{u}^\mu,\qquad(331)$$

where $\alpha$ is a function of $\mathbb{T}$ and $\nu$. According to (327), the action (330) is modified as

$$S_{eff} \to S_{eff} + \int d^d\sigma d\theta d\bar{\theta}\sqrt{-\mathbb{g}}\left(\mathbb{u}^i\partial_i\epsilon + (\epsilon + P)\mathbb{\nabla}_i\mathbb{u}^i\right)\alpha + O(\partial^2).\qquad(332)$$

Using (324) to replace the scalars $\mathbb{u}^i\partial_i\nu$ and $\mathbb{\nabla}_i\mathbb{u}^i$ in favor of $\mathbb{u}^i\partial_i\mathbb{T}$ the new term in (332) simply vanishes and the action is left invariant. While we cannot remove the one-derivative term $p_1^-$ from the action, it can be removed from the super-constitutive relations by a suitable redefinition of $\mathbb{T}$, $\mathbb{u}^i$, and $\nu$, which one might call a super-frame transformation. We leave such a study for future work.

In the examples to follow, we use the ideal equations of motion (324) to remove various couplings from the effective action. We then compute the constitutive relations, and use a frame transformation to put the stress tensor and flavor current into Landau frame.

## 7.1 Parity-violating first-order hydrodynamics in 2+1 dimensions

Let us consider a $2+1$ dimensional, parity violating, charged fluid to first order in derivatives. The most general Lagrangian for such a fluid is given by

$$
\begin{aligned}
L ={}& \frac{1}{2}L_0(\mathbb{g},\mathbb{B};\mathbb{\nabla}) + \frac{i}{2}L^{ij}(\mathbb{g},\mathbb{B};\mathbb{\nabla})D_\theta\mathbb{B}_iD_{\bar{\theta}}\mathbb{B}_j + \frac{i}{2}L^{ijkl}(\mathbb{g},\mathbb{B};\mathbb{\nabla})D_\theta\mathbb{g}_{ij}D_{\bar{\theta}}\mathbb{g}_{kl} \\
&+ \frac{i}{2}L^{ijk}(\mathbb{g},\mathbb{B};\mathbb{\nabla})D_\theta\mathbb{B}_iD_{\bar{\theta}}\mathbb{g}_{jk} + \frac{i}{2}\bar{L}^{ijk}(\mathbb{g},\mathbb{B};\mathbb{\nabla})D_\theta\mathbb{g}_{ij}D_{\bar{\theta}}\mathbb{B}_k,
\end{aligned}
\qquad(333)
$$

and we have suppressed the dependence on $\mathbb{T}$ and $\nu$. We consider the most general $L_0$ to first order in derivatives and the two-tensor terms at zeroth order in derivatives. In Table 1 we have listed all possible tensor and scalar structures which may contribute to these terms, where $\mathbb{e}^{ijk}$ is the super-Levi-Civita tensor,

$$\mathbb{e}^{012} = \frac{1}{\sqrt{-\mathbb{g}}},\qquad(334)$$

and we have defined the super-projector

$$\mathbb{P}^{ij} = \mathbb{g}^{ij} + \mathbb{u}^i\mathbb{u}^j,\qquad(335)$$

where $\mathbb{u}^i = \mathbb{T}\beta^i$ is the normalized super-velocity and $\mathbb{T} = \frac{1}{\sqrt{-\mathbb{g}_{ij}\beta^i\beta^j}}$ is the super-temperature. We raise and lower indices using $\mathbb{g}_{ij}$ and its inverse. For example, $\mathbb{\beta}_i = \mathbb{g}_{ij}\beta^j$.

Let us start with the scalar action. The most general action we may write down is composed of a leading term which we have seen in Section 4 contributes to the pressure and five additional terms coming from the five scalar quantities appearing in table 1. Of these, one can be removed via integration by parts, and another by using the equations of motion, so we end up with

$$\frac{1}{2}\left(L_0 + \widetilde{L}_0\right) = P + p^-\beta^i\partial_i\mathbb{T} + M_\Omega^+\mathbb{e}^{ijk}\mathbb{\beta}_i\partial_j\mathbb{\beta}_k + M_B^-\mathbb{e}^{ijk}\mathbb{\beta}_i\partial_j\mathbb{B}_k\qquad(336)$$

Table 1: Possible contributions to the tensor structures specified by (333). In writing down the entries we have omitted possible permutations of indices. The super Levi-Civita tensor $\mathfrak{e}^{ijk}$ is defined in (334) and the super projection $\mathbb{P}^{ij}$ is defined in (335). Terms which do not involve the Levi-Civita term are present in an expansion action to first order in derivatives in any spacetime dimension. Those terms involving $\mathfrak{e}^{ijk}$ are specific to $2+1$ dimensional fluids.

| $L_0$ | $L^{ij}$ | $L^{ijk}$ | $L^{ijkl}$ |
|---|---|---|---|
| $\mathbb{W}_i\beta^i$ | $\mathbb{P}^{ij}$ | $\beta^i\beta^j\beta^k$ | $\beta^i\beta^j\beta^k\beta^l$ |
| $\beta^i\partial_i\mathbb{T}$ | $\beta^i\beta^j$ | $\mathbb{P}^{ij}\beta^k$ | $\mathbb{P}^{ij}\beta^k\beta^l$ |
| $\beta^i\partial_i\mathbb{\nu}$ | $\mathfrak{e}^{ijm}\mathbb{\beta}_m$ | $\mathfrak{e}^{ijm}\mathbb{P}_m{}^k$ | $\mathbb{P}^{ij}\mathbb{P}^{kl}$ |
| $\mathfrak{e}^{ijk}\mathbb{\beta}_i\mathbb{W}_j\mathbb{\beta}_k$ | | $\mathfrak{e}^{ijm}\mathbb{\beta}_m\beta^k$ | $\mathfrak{e}^{ikm}\mathbb{\beta}_m\mathbb{P}^{jl}$ |
| $\mathfrak{e}_m{}^{jk}\beta^m\mathbb{W}_j\mathbb{B}_k$ | | | $\epsilon^{ikm}\mathbb{\beta}_m\beta^j\beta^l$ |

up to a total derivative. The KMS partner terms imply that $P$ and $M_\Omega^+$ are even functions of $\nu$ while $M_B^-$ and $p^-$ are odd functions of $\nu$.

Carrying out the variation with respect to $B_{a\,i}$ and $g_{a\,ij}$ we find that

$$
\begin{aligned}
J_{r\,(0)}^k =& \frac{\partial P}{\partial\nu}\beta^k + \left(\frac{\partial p^-}{\partial\nu}\beta\cdot\partial T + \frac{\partial M_\Omega^+}{\partial\nu}\epsilon^{ijm}\beta_i\partial_j\beta_m + \frac{\partial M_B^-}{\partial\nu}\epsilon^{ijm}\beta_i\partial_j B_m\right)\beta^k \\
& - \left(\frac{\partial M_B^-}{\partial\nu}\partial_j\nu + \frac{\partial M_B^-}{\partial T}\partial_j T\right)\epsilon^{ijk}\beta_i - M_B^-\epsilon^{ijk}\partial_j\beta_i\,, \\
T_{r\,(0)}^{ij} =& g^{ij}P + \frac{\partial P}{\partial T}T^3\beta^i\beta^j + g^{ij}p^-\beta\cdot\partial T - \left(p^-\partial\cdot\beta + \frac{\partial p^-}{\partial\nu}\beta\cdot\partial\nu\right)T^3\beta^i\beta^j \\
& + M_\Omega^+\epsilon^{nk(i}\beta^{j)}\partial_n\beta_k + M_\Omega^+\epsilon^{mn(i}\beta_m\partial_n\beta^{j)} + M_B^-\epsilon^{nk(i}\beta^{j)}\partial_n B_k\,,
\end{aligned}
\tag{337}
$$

where indices are lowered and raised with $g_{r\,ij}$ and its inverse.

To understand the structure of the tensor terms it is convenient to recast (333) in the form

$$
L = \frac{1}{2}L_0(\mathbb{g},\mathbb{B};\mathbb{W}) + \frac{i}{2}D_\theta\begin{pmatrix}\mathbb{B}\\\mathbb{g}\end{pmatrix}L^{AB}D_{\bar\theta}\begin{pmatrix}\mathbb{B}\\\mathbb{g}\end{pmatrix}\,,
\tag{338}
$$

with

$$
L^{AB} = \begin{pmatrix} L^{i\ell} & L^{imn} \\ \bar{L}^{jk\ell} & L^{jkmn} \end{pmatrix}\,.
\tag{339}
$$

The constitutive relations for the tensor terms can be read off of (237) and are given by

$$
\begin{pmatrix} J_{(2)}^i \\ \frac{1}{2}T_{(2)}^{jk} \end{pmatrix} = \frac{1}{2}\left(L_- - L_+\right)\begin{pmatrix} \partial_\ell\nu - G_{\ell p}\beta^p \\ \nabla_m\beta_n + \nabla_n\beta_m \end{pmatrix}\,,
\tag{340}
$$

where we have defined the matrices

$$
L_- = \begin{pmatrix} L_-^{(i\ell)} & L_-^{imn} \\ L_-^{T\,jk\ell} & L_-^{((jk)(mn))} \end{pmatrix}\,, \qquad L_+ = \begin{pmatrix} L_+^{[i\ell]} & L_+^{imn} \\ -L_+^{T\,jk\ell} & L_+^{[(jk)(mn)]} \end{pmatrix}\,,
\tag{341}
$$

with

$$L_\pm^{i\ell} = L^{i\ell} \mp \eta_i \eta_\ell \widetilde{L}^{i\ell} \,,$$

$$L_\pm^{imn} = \frac{1}{2}\left( L^{imn} \mp \eta_i \eta_m \eta_n \widetilde{L}^{imn}\right) \mp \frac{1}{2}\left( \bar{L}^{mni} \mp \eta_i \eta_m \eta_n \widetilde{\bar{L}}^{mni}\right)\,,$$

$$L_\pm^{T\,jk\ell} = L_\pm^{\ell jk}\,,$$

$$L_\pm^{(ij)(mn)} = L^{(ij)(mn)} \mp \eta_i \eta_j \eta_m \eta_n \widetilde{L}^{(ij)(mn)}\,. \tag{342}$$

To compute these expressions explicitly, let us denote

$$L^{ij} = - s_1 \mathbb{P}^{ij} - s_2 \beta^i \beta^j - s_3 \mathfrak{e}^{ijk}\beta_k\,,$$

$$L^{ijk} = - q_1 \mathbb{P}^{i(j}\beta^{k)} - q_2 \beta^i \mathbb{P}^{jk} - q_3 \beta^i \beta^j \beta^k - q_4 \beta_n \mathfrak{e}^{ni(j}\beta^{k)}\,,$$

$$\bar{L}^{jki} = - \bar{q}_1 \mathbb{P}^{i(j}\beta^{k)} - \bar{q}_2 \beta^i \mathbb{P}^{jk} - \bar{q}_3 \beta^i \beta^j \beta^k - \bar{q}_4 \beta_n \mathfrak{e}^{ni(j}\beta^{k)}\,,$$

$$L^{(ij)(kl)} = - r_1 \beta^i \beta^j \beta^k \beta^l - r_2 \left( \mathbb{P}^{ij}\beta^k \beta^l + \mathbb{P}^{kl}\beta^i \beta^j\right) - r_3 \left( \mathbb{P}^{ij}\beta^k \beta^l - \mathbb{P}^{kl}\beta^i \beta^j\right) - r_4 \beta^{(i}\mathbb{P}^{j)(k}\beta^{\ell)}$$
$$- r_5 \mathbb{P}^{ij}\mathbb{P}^{kl} - r_6 \mathbb{P}^{i(k}\mathbb{P}^{\ell)j} - r_7 \beta_n \left( \mathfrak{e}^{ni(k}\mathbb{P}^{\ell)j} + \mathfrak{e}^{nj(k}\mathbb{P}^{\ell)i}\right) - r_8 \beta_n \beta^{(i}\mathfrak{e}^{j)n(k}\beta^{\ell)}\,. \tag{343}$$

Rewriting (340) in a more traditional form,

$$J_{r\,(2)}^i = \eta^{ij}\left( \partial_j \nu - G_{jk}\beta^k\right) + \eta^{ijk}\left( \nabla_j \beta_k + \nabla_k \beta_j\right)\,,$$

$$\frac{1}{2}T_{r\,(2)}^{jk} = \bar{\eta}^{jki}\left( \partial_i \nu - G_{il}\beta^l\right) + 2\eta^{jkmn}(\nabla_m \beta_n + \nabla_n \beta_m)\,, \tag{344}$$

we find

$$2\eta^{ij} = - s_1^+ P^{ij} - s_2^+ \beta^i \beta^j + s_3^+ \epsilon^{ijk}\beta_k\,,$$

$$4\eta^{ijk} = - (\theta_1^- - \bar{\theta}_1^+) P^{i(j}\beta^{k)} - (\theta_2^- - \bar{\theta}_2^+)\beta^i P^{jk} - (\theta_3^- - \bar{\theta}_3^+)\beta^i \beta^j \beta^k - (\theta_4^+ - \bar{\theta}_4^-)\beta_n \epsilon^{ni(j}\beta^{k)}\,,$$

$$4\bar{\eta}^{jki} = - (\theta_1^- + \bar{\theta}_1^+) P^{i(j}\beta^{k)} - (\theta_2^- + \bar{\theta}_2^+)\beta^i P^{jk} - (\theta_3^- + \bar{\theta}_3^+)\beta^i \beta^j \beta^k - (\theta_4^+ + \bar{\theta}_4^-)\beta_n \epsilon^{ni(j}\beta^{k)}\,,$$

$$2\eta^{ijkl} = - r_1^+ \beta^i \beta^j \beta^k \beta^l - r_2^+ \left( P^{ij}\beta^k \beta^l + P^{kl}\beta^i \beta^j\right) - r_4^+ \beta^{(i}P^{j)(k}\beta^{\ell)} - r_5^+ P^{ij}P^{kl} - r_6^+ P^{i(k}P^{\ell)j}$$
$$- r_3^- \left( P^{ij}\beta^k \beta^l - P^{kl}\beta^i \beta^j\right) - r_7^+ \beta_n \left( \epsilon^{ni(k}P^{\ell)j} + \epsilon^{nj(k}P^{\ell)i}\right) - r_8^+ \beta_n \beta^{(i}\epsilon^{j)n(k}\beta^{\ell)}\,, \tag{345}$$

where the $\pm$ superscript on the coefficients $s_i$, and $r_i$ specifies whether it is even or odd under CPT,

$$s_i^\pm = s_i(T, \nu) \pm s_i(T, -\nu)\,, \qquad r_i^\pm = r_i(T, \nu) \pm r_i(T, -\nu)\,, \tag{346}$$

and

$$\theta_i^\pm = (q_i(T, \nu) \pm q_i(T, -\nu)) + (\bar{q}_i(T, -\nu) \pm \bar{q}_i(T, -\nu))\,,$$

$$\bar{\theta}_i^\pm = (q_i(T, \nu) \pm q_i(T, -\nu)) - (\bar{q}_i(T, -\nu) \pm \bar{q}_i(T, -\nu))\,. \tag{347}$$

Note that terms associated with the coefficients $r_1^+$, $r_2^+$, $r_4^+$, $r_5^+$, $r_6^+$, $\theta_i^-$, $\theta_4^+$, $s_1^+$ and $s_2^+$ contribute to $L_-^{AB}$ whereas those associated with $r_3^-$, $\bar{\theta}_i^+$, $\bar{\theta}_4^-$, $r_7^+$, $r_8^+$ and $s_3^+$ contributes to $L_+^{AB}$.

To unpackage (337) and (345) let us decompose the fields into scalars vectors and tensors of the $SO(2)$ symmetry which is preserved under rotations around the directions of $\beta^i$. Traceless symmetric representations of $SO(2)$ are given by the combination

$$P^i_{\ \ell}P^j_{\ k}T^{\ell k} - \frac{1}{2}P^{ij}P_{\ell k}T^{\ell k} = -\eta \sigma^{ij} + \tilde{\eta}\tilde{\sigma}^{ij}\,, \tag{348}$$

with

$$\eta = \frac{r_6^+}{T}, \qquad \tilde{\eta} = \frac{1}{T^2}(M_\Omega^+ - 2r_7^+),$$

(349)

where we have defined

$$\nabla_k \beta_q \left( P^{iq} P^{jk} + P^{ik} P^{jq} - P^{ij} P^{kq} \right) = T^{-1} \sigma^{ij},$$

$$\beta_n \nabla_k \beta_q \left( P^{jq} \epsilon^{nik} + P^{jk} \epsilon^{niq} + P^{iq} \epsilon^{njk} + P^{ik} \epsilon^{njq} \right) = -2T^{-2} \tilde{\sigma}^{ij}.$$

(350)

We identify $\eta$ with the shear viscosity and $\tilde{\eta}$ with the Hall viscosity.

There are two $SO(2)$ invariant vectors

$$P^j{}_i J^i = s_1^+ \frac{V^j}{2T} - \frac{1}{4}(\theta_1^- - \bar{\theta}_1^+) P^{jk} \left( \frac{\partial_k T}{T^3} + \beta^i \partial_i \beta_k \right) + \left( \frac{s_3^+}{2} + \frac{\partial M_B^-}{\partial \nu} \right) \frac{\tilde{V}^j}{T^2} - \frac{1}{T^2} \frac{\partial M_B^-}{\partial \nu} \tilde{E}^j$$

$$+ \left( \frac{1}{4} \left( \theta_4^+ + \bar{\theta}_4^- \right) - T^2 M_B^- \right) \epsilon^{jmk} \beta_m \left( \frac{\partial_k T}{T^3} + \beta^i \partial_i \beta_k \right)$$

$$+ \left( \frac{2M_B^-}{T} - \frac{\partial M_B^-}{\partial T} \right) \epsilon^{jmk} \beta_m \partial_k T,$$

(351)

and

$$P^j{}_i T^{ik} \beta_k = -\frac{2}{T^3} \left( \theta_1^- - \bar{\theta}_1^+ \right) V^j + \frac{r_+}{2T^2} P^{jk} \left( \frac{\partial_k T}{T^3} + \beta^i \partial_i \beta_k \right) - \frac{1}{4T^4} \left( \theta_4^+ + \bar{\theta}_4^- \right) \tilde{V}^j$$

$$- \frac{M_B^-}{T^2} \tilde{E}^j + \left( M_\Omega^+ - \frac{r_8^+}{2T^2} \right) \epsilon^{jnk} \beta_n \left( \frac{\partial_k T}{T^3} + \beta^i \partial_i \beta_k \right) - M_\Omega^+ \epsilon^{jmk} \beta_m \frac{\partial_k T}{T^3},$$

(352)

where we have used

$$\epsilon^{iml} \beta_m \beta^k \nabla_k \beta_l = \epsilon^{kml} \beta_m \beta^i \nabla_k \beta_l + \epsilon^{ikl} \beta_m \beta^m \nabla_k \beta_l + \epsilon^{imk} \beta_m \beta^l \nabla_k \beta_l,$$

(353)

and

$$\beta^l \nabla_k \beta_l = \partial_k T / T^3,$$

(354)

and the definitions

$$V^i = E^i - T P^{ij} \partial_j \nu, \qquad \tilde{V}^i = T \epsilon^{ijk} \beta_j V_k, \qquad \tilde{E}^i = T \epsilon^{ijk} \beta_j E_k.$$

(355)

The three $SO(3)$ invariant scalars are given by

$$\beta_i T^{ij} \beta_j = \frac{\epsilon}{T^2} - \frac{2r_1^+}{T^7} \beta \cdot \partial T + \frac{\Sigma - \left( \theta_3^- - \bar{\theta}_3^+ \right)}{2T^4} \beta \cdot \partial \nu$$

$$- \frac{(2(r_2^+ + r_3^-) + p^- T^3)}{T^5} \Theta + \frac{2M_B^-}{T^3} B - \frac{2M_\Omega^+}{T^4} \Omega,$$

$$\beta^i J_i = \frac{1}{2T^3} \left( \theta_2^- - \bar{\theta}_2^+ \right) \Theta + \frac{\Sigma + (\theta_3^- - \bar{\theta}_3^+)}{2T^5} \beta \cdot \partial T + \frac{s_2^+}{2T^2} \beta \cdot \partial \nu - \frac{1}{T^2} \frac{\partial M_B^-}{\partial \nu} B$$

$$+ \frac{1}{T^4} \left( M_B^- T^2 - \frac{\partial M_\Omega^+}{\partial \nu} \right) \Omega,$$

$$P_{ij} T^{ij} = 2P - 4 \left( \theta_2^- - \bar{\theta}_2^+ \right) \beta \cdot \partial \nu - \frac{8}{T} \left( r_5^+ + r_6^+ \right) \Theta - \frac{2}{T^3} \left( 4(r_2^+ + r_3^-) - p^- T^3 \right) \beta \cdot \partial T$$

$$+ \frac{2M_\Omega^-}{T^2} \Omega,$$

(356)

with

$$\frac{\Sigma}{2} = \frac{\partial p^+}{\partial T}T^3 - p^+T^2 - \frac{\partial p^-}{\partial \nu}T^3 \,, \tag{357}$$

and where we have defined

$$\Omega = T^2\epsilon^{ijk}\beta_i\nabla_j\beta_k \,, \qquad B = -T\epsilon^{ijk}\beta_i\partial_j B_k \,, \qquad \Theta = TP^{ij}\nabla_i\beta_j \,. \tag{358}$$

We are interested in the on-shell constitutive relations. Since we are working to first order in derivatives, we may use the zeroth order equations of motion to obtain on-shell relations between first order scalars and vectors. To this end, we recast (314) in the form

$$\beta\cdot\partial T = -\left(\frac{\partial P}{\partial \epsilon}\right)_\rho\Theta, \qquad \beta\cdot\partial \nu = -\frac{1}{T^2}\left(\frac{\partial P}{\partial \rho}\right)_\epsilon\Theta\,, \qquad \frac{1}{T^3}P^{ij}\partial_j T + P^{ij}\beta^k\nabla_k\beta_j = -\frac{R_0}{T}V^i\,, \tag{359}$$

where $R_0 = \rho/(\epsilon + P)$ and $\rho$ and $\epsilon$ were defined in (163). As should be clear from direct inspection, the vectors and scalars slightly simplify under (359).

Further simplification of the constitutive relations can be obtained by switching to frame invariant variables. In [46] it was shown that, to first order in the derivative expansion, the vector combination $P_{ij}J^j + R_0 T P_{ij}T^{jk}\beta_k$ is frame invariant as is the scalar combination $\frac{1}{2}\left(P_{ij}T^{ij} - 2P\right) - \left(\frac{\partial P}{\partial \epsilon}\right)_\rho T^2\left(\beta_i T^{ij}\beta_j - \frac{\epsilon}{T^2}\right) + \left(\frac{\partial P}{\partial \rho}\right)_\epsilon T\left(\beta_i J^i + \frac{\rho}{T}\right)$. We find

$$P_{ij}J^j + R_0 T P_{ij}T^{jk}\beta_k = \sigma V_i + \tilde{\sigma}\tilde{V}^i + \tilde{\chi}_E\tilde{E}^i + \tilde{\chi}_T T\epsilon^{ijk}\beta_j\partial_k T\,, \tag{360}$$

with

$$\begin{aligned}
\sigma &= \frac{R_0^2 r_4^+}{2T^3} - \frac{R_0\theta_1^-}{2T^2} + \frac{s_1^+}{2T}\,, \\
\tilde{\chi}_E &= -\frac{R_0 M_B^-}{T} - \frac{1}{T^2}\frac{\partial M_B^-}{\partial \nu}\,, \\
\tilde{\chi}_T &= -\frac{R_0 M_\Omega^+}{T^3} + \frac{2M_B^-}{T^2} - \frac{1}{T}\frac{\partial M_B^-}{\partial T}\,, \\
\tilde{\sigma} &= \frac{1}{T}\left(R_0^2\left(\frac{M_\Omega^+}{T} - \frac{r_8^+}{2T^3}\right) - R_0\left(M_B^- + \frac{\bar{\theta}_4^-}{2T^2}\right) + \frac{\partial M_B^-}{\partial \nu}\frac{1}{T} + \frac{s_3^+}{T}\right)\,,
\end{aligned} \tag{361}$$

and

$$\begin{aligned}
\frac{1}{2}\left(P_{ij}T^{ij} - 2P\right) &- \left(\frac{\partial P}{\partial \epsilon}\right)_\rho T^2\left(\beta_i T^{ij}\beta_j - \frac{\epsilon}{T^2}\right) + \left(\frac{\partial P}{\partial \rho}\right)_\epsilon T\left(\beta_i J^i + \frac{\rho}{T}\right) \\
&= -\tilde{\chi}_B B - \tilde{\chi}_\Omega \Omega - \zeta\Theta\,,
\end{aligned} \tag{362}$$

with

$$\begin{aligned}
\tilde{\chi}_B &= \frac{1}{T}\left(\left(\frac{\partial P}{\partial \rho}\right)_\epsilon\frac{\partial M_B^-}{\partial \nu} + 2\left(\frac{\partial P}{\partial \epsilon}\right)_\rho M_B^-\right)\,, \\
\tilde{\chi}_\Omega &= -\frac{1}{T}\left(\left(\frac{\partial P}{\partial \rho}\right)_\epsilon\left(M_B^- - \frac{1}{T^2}\frac{\partial M_\Omega^+}{\partial \nu}\right) + \frac{\left(1 + 2\left(\frac{\partial P}{\partial \epsilon}\right)_\rho\right)M_\Omega^+}{T}\right)\,, \\
\zeta &= \frac{1}{T}\left(2r_5^+ + r_6^+ + \frac{2}{T^4}\left(\frac{\partial P}{\partial \epsilon}\right)_\rho^2 r_1^+ - \frac{4}{T^2}\left(\frac{\partial P}{\partial \epsilon}\right)_\rho r_2^+ + \frac{1}{2T^2}\left(\frac{\partial P}{\partial \rho}\right)_\epsilon^2 s_2^+\right. \\
&\qquad \left. - \frac{1}{T^3}\left(\frac{\partial P}{\partial \rho}\right)_\epsilon\left(T^2\theta_2^- - \left(\frac{\partial P}{\partial \epsilon}\right)_\rho\theta_3^-\right)\right)\,.
\end{aligned} \tag{363}$$

It is straightforward to enforce positivity of the divergence of the entropy current (242) which implies, via (340) and

$$\partial_j \nu - G_{jk}\beta^k = -\frac{1}{T}V_j + \left(\frac{\partial P}{\partial \rho}\right)_\epsilon \beta_j \Theta\,,$$

$$\nabla_i \beta_j + \nabla_j \beta_i = \left(\frac{P_{ij}}{T} - 2T\left(\frac{\partial P}{\partial \epsilon}\right)_\rho \beta_i \beta_j\right)\Theta - R_0\left(V_j\beta_i + V_i\beta_j\right) + \frac{1}{T}\sigma_{ij}\,,$$

(364)

that

$$\frac{1}{T}\left(\frac{1}{2}P_{ij}T^{ij}_{(2)-} - T^2\left(\frac{\partial P}{\partial \epsilon}\right)_\rho \beta_i\beta_j T^{ij}_{(2)-} + T\left(\frac{\partial P}{\partial \rho}\right)_\epsilon \beta_i J^i_{(2)-}\right)\Theta$$

$$- \frac{V_k}{T}P_{ki}\left(J^i_{(2)-} + R_0 T^{ij}_{(2)-}\beta_j\right) + \frac{1}{T}\sigma_{ij}T^{ij} \le 0\,. \quad (365)$$

Here, $T^{ij}_{(2)-}$ and $J^{ij}_{(2)-}$ represent contributions to the stress tensor and current coming from $L_-$ in (341) or, more specifically, the contribution of terms containing $s_1^+$, $s_2^+$, $\theta_1^-$, $\theta_2^-$, $\theta_3^-$, $\theta_4^+$, $r_1^+$, $r_2^+$, $r_4^+$, $r_5^+$ and $r_6^+$. Inserting (348), (360) and (362) into (365) we find

$$-\frac{\zeta\Theta^2}{T} - \frac{V_k V^k \sigma}{T} - \frac{4r_6^+\sigma_{ij}\sigma^{ij}}{T^2} \le 0\,, \quad (366)$$

implying

$$\zeta \ge 0\,, \qquad \sigma \ge 0\,, \qquad \eta \ge 0\,, \quad (367)$$

reproducing the expected positivity of the bulk viscosity, conductivity and shear viscosity.

The on-shell condition (365) is necessary but not sufficient to ensure positivity of the imaginary part of the effective action. Indeed, to complete our analysis of transport it remains to solve the Schwinger-Keldysh positivity condition $L_- \preceq 0$ which in the current context takes the form

$$-\begin{pmatrix} B_a & g_a \end{pmatrix} L_- \begin{pmatrix} B_a \\ g_a \end{pmatrix} \ge 0 \quad (368)$$

for any $B_{a\,i}$ and $g_{a\,ij}$ and with $L_-$ given in (341). Decomposing

$$g_a^{ij} = gT^4\beta^i\beta^j - T^2 g^i\beta^j - T^2 g^j\beta^i + \gamma^{ij}\,, \qquad B_a^i = -T^2 b\beta^i + b^i\,, \quad (369)$$

where $g^i\beta_i = 0$, $b^i\beta_i = 0$ and $\gamma^{ij}\beta_j = \gamma_i\tau^{ij} = 0$, we find

$$\gamma^{ij}\gamma_{ij}r_6^+ + \beta_n b_i g_j \epsilon^{nij}\theta_4^+ + \sum_i \begin{pmatrix} g_i & b_i \end{pmatrix} V\begin{pmatrix} g^i \\ b^i \end{pmatrix} + \begin{pmatrix} g & \gamma_i^i & b \end{pmatrix} S\begin{pmatrix} g \\ \gamma_i^i \\ b \end{pmatrix} > 0\,, \quad (370)$$

with

$$V = \begin{pmatrix} r_4^+ & \frac{\theta_1^-}{2} \\ \frac{\theta_1^-}{2} & s_1^+ \end{pmatrix}\,, \qquad S = \begin{pmatrix} r_1^+ & r_2^+ & \frac{\theta_3^-}{2} \\ r_2^+ & r_5^+ & \frac{\theta_2^-}{2} \\ \frac{\theta_3^-}{2} & \frac{\theta_2^-}{2} & s_2^+ \end{pmatrix}\,. \quad (371)$$

Thus, Schwinger-Keldysh positivity implies that

$$r_6^+ \ge 0\,, \qquad \theta_4^+ \ge 0\,, \qquad V \succeq 0\,, \qquad S \succeq 0\,. \quad (372)$$

The last two conditions constrain the transport coefficients to lie in a convex subspace of parameter space. For example, $V \succ 0$ implies

$$r_4^+ > 0\,, \qquad s_1^+ - \frac{\left(\theta_1^-\right)^2}{r_4^+} \ge 0\,. \quad (373)$$

Let us collect our results. The constitutive relations for the stress tensor and current of a $2+1$ dimensional parity violating fluid in $2+1$ dimensions satisfies (348), (360) and (362) which is identical to what was found in [9], but with additional information about the CPT transformation properties of the transport coefficients: $\tilde{\chi}_\Omega$, $\zeta$, $\sigma$, $\tilde{\sigma}$, $\tilde{\chi}_E$, $\eta$ and $\tilde{\eta}$ are CPT even while $\tilde{\chi}_B$, $\tilde{\chi}_T$ are CPT odd.

The terms $\tilde{\chi}_E$, $\tilde{\chi}_B$, $\tilde{\chi}_\Omega$ and $\tilde{\chi}_T$ depend only on $M_\Omega^+$ and $M_B^-$ so that these four coefficients are interdependent. This is not surprising. These relations were observed already in [6,7] using the equilibrium partition function. Indeed, in our classification scheme $M_B^-$ and $M_\Omega^+$ are scalar terms which survive in the hydrostatic limit where $\delta_\beta \mathbb{F} = 0$. The other two coefficients $p^-$ and $p^+$ are not hydrostatic but do not contribute to transport; they vanish when we switch to on-shell frame invariant variables.

We also note that even though we started off with 19 independent coefficients in the tensor sector, we found only three dissipative transport coefficients, $\sigma$, $\zeta$ and $\eta$ and two non dissipative transport coefficients, $\tilde{\eta}$ and $\tilde{\sigma}$ all of which are CPT even. This reduction may have also been argued for by carrying out a frame transformation to, say, the Landau frame. The structure (344) implies that after such a transformation 11 of the coefficients, $s_2^+$, $\theta_1^- - \bar{\theta}_1^+$, $\theta_2^- + \bar{\theta}_2^+$, $\theta_3^-$, $\bar{\theta}_3^+$, $\theta_4^+ - \bar{\theta}_4^-$, $r_1^+$, $r_2^+$, $r_3^-$, $r_4^+$ and $r_8^+$, may be reabsorbed into a redefinition of the other coefficients and are therefore redundant. In addition, the on-shell relations (364) together with (344) and (345) imply (among other things) that the tensor structure associated with $\theta_1^- + \bar{\theta}_1^+$ is the same as that of $s_1^+$, that of $\theta_2^- - \bar{\theta}_2^+$ is the same as $r_5^+$ and that of $\theta_4^+ + \bar{\theta}_4^-$ is the same as $s_3^+$. Thus, had we been interested only in the on-shell constitutive relations it would have been sufficient to use $s_1^+$, $r_5^+$ and $r_6^+$ as representatives of transport coefficients associated with dissipation and $s_3^+$ and $r_7^+$ as representatives of transport coefficients associated with terms in the constitutive relations which do not generate dissipation.

While it seems that the number of coefficients in our action is overly redundant, we remind the reader that, our main goal in this work was to study the constitutive relations for the on-shell stress tensor, expanded perturbatively in derivatives. The Schwinger-Keldysh effective action is capable of reproducing the hydrodynamic stress tensor but also contains information on the off-shell stress tensor and on stochastic noise associated with $a$-type fields. The multitude of coefficients in the effective action encode this extra information to which the hydrodynamic stress tensor is oblivious.

### 7.2 Second order neutral fluid in $d$ dimensions

Having dealt with the parity breaking $2+1$ dimensional charged fluid at the one derivative level, let us consider transport coefficients at the two derivative level. Since the number of independent transport coefficients increases significantly as the number of derivatives increases (for instance, there are 38 transport coefficients for a parity breaking charged conformal fluid in $3+1$ dimensions [47]) we will focus in this Subsection on parity preserving uncharged fluids at second order in derivatives. The Lagrangian in this case takes the form

$$L = \frac{1}{2}L_0 + \frac{i}{2}L^{(ij)(kl)}D_\theta \mathfrak{g}_{ij}D_{\bar{\theta}}\mathfrak{g}_{kl} + \frac{i}{2}L_\ell^{(ij)(kl)m}\mathbb{W}_m D_\theta \mathfrak{g}_{ij}D_{\bar{\theta}}\mathfrak{g}_{kl} + \frac{i}{2}L_r^{(ij)(kl)m}D_\theta \mathfrak{g}_{ij}\mathbb{W}_m D_{\bar{\theta}}\mathfrak{g}_{kl}$$
$$- \frac{i}{4}L^{(ij)(kl)(mn)}\delta_\beta \mathfrak{g}_{mn}D_\theta \mathfrak{g}_{ij}D_{\bar{\theta}}\mathfrak{g}_{kl} - \frac{1}{2}L^{(ij)(kl)(mn)}D_\theta \mathfrak{g}_{ij}D_{\bar{\theta}}\mathfrak{g}_{kl}D\mathfrak{g}_{mn}, \quad (374)$$

where $D = D_{\bar{\theta}}D_\theta$.

The most general contribution to the scalar terms can be parameterized as follows

$$
\begin{aligned}
\frac{1}{2}\left(L_0 + \tilde{L}_0\right) =& P + p_2 \left(\mathbb{W}_i \mathbb{T} \mathbb{W}^i \mathbb{T} - \mathbb{T}^6 \left(\beta \cdot \mathbb{W}\beta_n\right)\left(\beta \cdot \mathbb{W}\beta^n\right)\right) \\
& + p_3 \left(\beta \cdot \mathbb{W}\beta^j \mathbb{W}_j \mathbb{T} + \frac{1}{\mathbb{T}^3}\mathbb{W}_i \mathbb{T} \mathbb{W}^i \mathbb{T} + 2\mathbb{T}\left(\frac{P'}{P' + \mathbb{T}P''}\right)^2 \left(\mathbb{W}\cdot\beta\right)^2\right) \\
& + p_4 \left(\beta^i \mathbb{W}_i \mathbb{T} + \frac{\mathbb{T}P'}{P' + \mathbb{T}P''}\mathbb{W}_i\beta^i\right)^2 + p_5(\mathbb{W}_i\beta^i)^2 \\
& + p_6(\mathbb{W}_i\beta^i)\left(\beta^i \mathbb{W}_i \mathbb{T} + \frac{\mathbb{T}P'}{P' + \mathbb{T}P''}\mathbb{W}_i\beta^i\right) + p_7\mathbb{R} + p_8\beta^i\beta^j\mathbb{R}_{ij} \\
& + p_9 \left(\beta \cdot \mathbb{W}\beta_n\right)\left(\beta \cdot \mathbb{W}\beta^n\right) + p_{10}(\mathbb{W}_m\beta^p)(\mathbb{W}^m\beta_p)\,,
\end{aligned}
\tag{375}
$$

where the coefficients $P, p_1, \ldots p_9$ are general functions of the super-temperature $\mathbb{T}$ and a $'$ denotes a derivative with respect to $\mathbb{T}$. At leading order in derivatives the single contribution to the Lagrangian is the pressure term $P$. At first order in derivatives there are no contributions to the scalar part of the action since CPT-odd transport coefficients must vanish. At second order in derivatives there are 9 independent scalar terms up to total derivatives. To derive the hydrodynamic constitutive relations, it is sufficient to consider configurations which are inequivalent on-shell as we did in the previous Section. We have conveniently organized our Lagrangian so that the terms in parenthesis on the right-hand side of (375) vanish under the equations of motion. Thus, for the purpose of computing the on-shell constitutive relations it is sufficient to keep track of only five of the nine second order terms $p_5$, $p_7$, $p_8$, $p_9$ and $p_{10}$.

Contributions from the tensor terms to the transport coefficients have the general structure

$$
\begin{aligned}
\frac{1}{2}T^{ij}_{(2)} =& \frac{1}{2}\left(L_-^{((ij)(kl))} - L_+^{[(ij)(kl)]}\right)\left(\nabla_k\beta_l + \nabla_l\beta_k\right) \\
& + \frac{1}{4}\nabla_m\left(-L_-^{((ij)(kl))m} - L_-^{[(ij)(kl)]m} + L_+^{[(ij)(kl)]m} + L_+^{((ij)(kl))m}\right)\left(\nabla_k\beta_l + \nabla_l\beta_k\right) \\
& + \frac{1}{2}\left(-L_-^{[(ij)(kl)]m} + L_+^{((ij)(kl))m}\right)\nabla_m\left(\nabla_k\beta_l + \nabla_l\beta_k\right) \\
& + \frac{1}{4}\left(N_-^{[(ij)(kl)]} - N_+^{((ij)(kl))}\right)\left(\nabla_k\beta_l + \nabla_l\beta_k\right)\,,
\end{aligned}
\tag{376}
$$

with

$$
\begin{aligned}
L_+^{[(ij)(kl)]} &= \left(L^{[(ij)(kl)]} - \eta_i\eta_j\eta_k\eta_l\widetilde{L}^{[(ij)(kl)]}\right)\,, \\
L_-^{((ij)(kl))} &= \left(L^{((ij)(kl))} + \eta_i\eta_j\eta_k\eta_l\widetilde{L}^{((ij)(kl))}\right)\,, \\
L_+^{((ij)(kl))m} &= \frac{1}{2}\left(L_\ell^{((ij)(kl))m} - L_r^{((ij)(kl))m}\right) - \frac{1}{2}\eta_i\eta_j\eta_k\eta_l\eta_m\left(\widetilde{L}_\ell^{((ij)(kl))m} - \widetilde{L}_r^{((ij)(kl))m}\right)\,, \\
L_+^{[(ij)(kl)]m} &= \frac{1}{2}\left(L_\ell^{[(ij)(kl)]m} + L_r^{[(ij)(kl)]m}\right) - \frac{1}{2}\eta_i\eta_j\eta_k\eta_l\eta_m\left(\widetilde{L}_\ell^{[(ij)(kl)]m} + \widetilde{L}_r^{[(ij)(kl)]m}\right)\,, \\
L_-^{((ij)(kl))m} &= \frac{1}{2}\left(L_\ell^{((ij)(kl))m} + L_r^{((ij)(kl))m}\right) + \frac{1}{2}\eta_i\eta_j\eta_k\eta_l\eta_m\left(\widetilde{L}_\ell^{((ij)(kl))m} + \widetilde{L}_r^{((ij)(kl))m}\right)\,, \\
L_-^{[(ij)(kl)]m} &= \frac{1}{2}\left(L_\ell^{[(ij)(kl)]m} - L_r^{[(ij)(kl)]m}\right) + \frac{1}{2}\eta_i\eta_j\eta_k\eta_l\eta_m\left(\widetilde{L}_\ell^{[(ij)(kl)]m} - \widetilde{L}_r^{[(ij)(kl)]m}\right)\,, \\
N_+^{((ij)(kl))} &= \left(L^{((ij)(kl))(mn)} + \eta_i\eta_j\eta_k\eta_l\eta_m\eta_n\widetilde{L}^{((ij)(kl))(mn)}\right)\left(\nabla_m\beta_n + \nabla_n\beta_m\right)\,, \\
N_-^{[(ij)(kl)]} &= \left(L^{[(ij)(kl)](mn)} - \eta_i\eta_j\eta_k\eta_l\eta_m\eta_n\widetilde{L}^{[(ij)(kl)](mn)}\right)\left(\nabla_m\beta_n + \nabla_n\beta_m\right)\,.
\end{aligned}
\tag{377}
$$

Let us decompose the two-tensor terms, $L^{(ij)(kl)}$, into zeroth order and first order terms in derivatives which contribute to first order and second order constitutive relations respectively,

$$L^{(ij)(kl)} = L_1^{(ij)(kl)} + L_s^{(ij)(kl)} + L_v^{(ij)(kl)} + L_t^{(ij)(kl)} . \tag{378}$$

The zeroth order terms are given by

$$2L_1^{(ij)(kl)} = - r_1\beta^i\beta^j\beta^k\beta^l - r_2\left(P^{ij}\beta^k\beta^l + P^{kl}\beta^i\beta^j\right)$$
$$- r_4\beta^{(i}P^{j)(k}\beta^{l)} - r_5 P^{ij}P^{kl} - r_6 P^{i(k}P^{l)j} + \dots , \tag{379}$$

where the $\dots$ include terms which may appear in $L_1^{(ij)(kl)}$ but will drop out of the action once we add to it the KMS partner of $L_1^{(ij)(kl)}$, in this particular case it would be a term of the form $r_3(T)\left(P^{ij}\beta^k\beta^l - P^{kl}\beta^i\beta^j\right)$.

The first order terms are given by

$$2L_s^{(ij)(kl)} = -\left(s_1\Theta + s_2\left(\beta \cdot \partial T + \left(\frac{\partial P}{\partial \epsilon}\right)\Theta\right)\right)\left(P^{ij}\beta^k\beta^l - P^{kl}\beta^i\beta^j\right) + \dots . \tag{380a}$$

Note that the $s_2$ contribution vanishes on-shell so if our goal is to obtain the on-shell constitutive relations then we may omit this term from the Lagrangian. We also have

$$2L_v^{(ij)(kl)} = - v_1\left(a^{(i}\beta^{j)}\beta^k\beta^l - a^{(k}\beta^{l)}\beta^i\beta^j\right) - v_2\left(a^{(i}\beta^{j)}P^{kl} - a^{(k}\beta^{l)}P^{ij}\right)$$
$$- v_3\left(a^{(i}P^{j)(k}\beta^{l)} - a^{(k}P^{l)(i}\beta^{j)}\right) + \dots , \tag{380b}$$

where terms which vanish on-shell have been omitted, and we have defined

$$a^i = P^i{}_k T^2 \beta^j \nabla_j \beta^k . \tag{380c}$$

The tensorial contributions to the first order two-tensor terms are

$$2L_t^{(ij)(kl)} = -t_1\left(P^{ij}\sigma^{kl} - P^{kl}\sigma^{ij}\right) - t_2\left(\beta^{(i}\omega^{j)(k}\beta^{l)} - \beta^{(k}\omega^{l)(i}\beta^{j)}\right)$$
$$- t_3\left(\omega^{i(k}P^{l)j} + \omega^{j(k}P^{l)i} - \omega^{k(i}P^{j)l} - \omega^{l(i}P^{j)k}\right) + \dots , \tag{380d}$$

with

$$\begin{aligned}
\sigma^{ij} &= T P^{ik}P^{jl}\left(\nabla_k\beta_l + \nabla_l\beta_k\right) - \frac{2}{d-1}T P^{ij}P^{kl}\nabla_k\beta_l , \\
\omega^{ij} &= \frac{1}{2}T P^{ik}P^{jl}\left(\nabla_k\beta_l - \nabla_l\beta_k\right) ,
\end{aligned} \tag{381}$$

and the $\dots$ in (380) refer to expressions which will vanish once the KMS partner Lagrangian is added. Likewise, we have

$$-L_\ell^{(ij)(kl)m} = - d_{\ell 1}\left(P^{ij}\beta^k\beta^l - P^{kl}\beta^i\beta^j\right)\beta^m - d_{\ell 2}\left(\beta^i\beta^j\beta^{(k}P^{l)m} - \beta^k\beta^l\beta^{(i}P^{j)m}\right)$$
$$- d_{\ell 3}\left(\beta^{(i}P^{j)(k}P^{l)m} - \beta^{(k}P^{l)(i}P^{j)m}\right) - d_{\ell 4}\beta^i\beta^j\beta^k\beta^l\beta^m$$
$$- d_{\ell 5}\left(P^{ij}\beta^k\beta^l + P^{kl}\beta^i\beta^j\right)\beta^m - d_{\ell 6}\beta^{(i}P^{j)(k}\beta^{l)}\beta^m - d_{\ell 7}P^{ij}P^{kl}\beta^m$$
$$- d_{\ell 8}P^{i(k}P^{l)j}\beta^m - d_{\ell 9}\left(\beta^i\beta^j\beta^{(k}P^{l)m} + \beta^k\beta^l\beta^{(i}P^{j)m}\right)$$
$$- d_{\ell 10}\left(\beta^{(i}P^{j)(k}P^{l)m} + \beta^{(k}P^{l)(i}P^{j)m}\right) + \dots , \tag{382}$$

and a corresponding term for $L_r^{(ij)(kl)m}$ (which differs from (382) only through its coefficients which we denote by $d_{r\,i}$) and the three-tensor terms

$$
\begin{aligned}
2L^{(ij)(kl)(mn)} = &+ m_1 \beta^i \beta^j \beta^k \beta^l \beta^m \beta^n + m_2 \beta^i \beta^j \beta^k \beta^l P^{mn} \\
&+ m_3 \left( P^{ij} \beta^k \beta^l + \beta^i \beta^j P^{kl} \right) \beta^m \beta^n \\
&+ m_4 \left( \beta^i \beta^j \beta^{(k} P^{l)(m} \beta^{n)} + \beta^k \beta^l \beta^{(i} P^{j)(m} \beta^{n)} \right) + m_5 \beta^{(i} P^{j)(k} \beta^{l)} \beta^m \beta^n \\
&+ m_6 \left( \beta^i \beta^j P^{kl} + P^{ij} \beta^k \beta^l \right) P^{mn} + m_7 \left( \beta^i \beta^j P^{k(m} P^{n)l} + \beta^k \beta^l P^{i(m} P^{n)j} \right) \\
&+ m_8 \left( P^{ij} \beta^{(k} P^{l)(m} \beta^{n)} + P^{kl} \beta^{(i} P^{j)(m} \beta^{n)} \right) \\
&+ m_9 \left( \beta^{(i} P^{j)(k} P^{l)(m} \beta^{n)} + \beta^{(k} P^{l)(i} P^{j)(m} \beta^{n)} \right) \\
&+ m_{10} P^{ij} P^{kl} \beta^m \beta^n + m_{11} \beta^{(i} P^{j)(k} \beta^{l)} P^{mn} + m_{12} P^{l(i} P^{j)k} \beta^m \beta^n \\
&+ m_{13} \beta^{(i} P^{j)(m} P^{n)(k} \beta^{l)} + m_{14} P^{ij} P^{kl} P^{mn} \\
&+ m_{15} \left( P^{ij} P^{k(m} P^{n)l} + P^{kl} P^{i(m} P^{n)j} \right) \\
&+ m_{16} P^{l(i} P^{j)k} P^{nm} + m_{17} \left( P^{n(i} P^{j)(k} P^{l)m} + P^{m(i} P^{j)(k} P^{l)n} \right) + \cdots ,
\end{aligned}
\tag{383}
$$

where . . . denote terms which do not contribute to the constitutive relations.

Inserting (379), (380), (382) and (383) into (377) we find that $L_1^{(ij)(kl)}$ are the only terms which contribute to $L_-^{((ij)(kl))}$, $L_s^{(ij)(kl)}$, $L_v^{(ij)(kl)}$ and $L_t^{(ij)(kl)}$ contribute to $L_+^{[(ij)(kl)]}$, $L_\ell^{(ij)(kl)}$ and $L_r^{(ij)(kl)}$ contribute to $L_+^{[(ij)(kl)]m}$ and $L^{(ij)(kl)(mn)}$ contribute only to $N_+^{((ij)(kl))(mn)}$. The other terms which appear on the left-hand side of (377) vanish.

Placing the theory on-shell and shifting to the Landau frame we find that the stress tensor takes the form

$$
T_r^{ij} = \epsilon u^i u^j + P P^{ij} + \tau_{(1)}^{ij} + \tau_{(2)}^{ij} ,
\tag{384}
$$

where $\tau_{(1)}^{ij}$, $\tau_{(2)}^{ij}$ are first and second order in derivative contributions to the stress tensor given by

$$
\begin{aligned}
\tau_{(1)}^{ij} = &- \eta \sigma^{ij} - \zeta P^{ij} \Theta , \\
\tau_{(2)}^{ij} = &\left[ \tau T^{\langle} \beta \cdot \nabla \sigma^{ij\rangle} + \kappa_1 R^{\langle ij\rangle} - \kappa_2 T^2 \beta^k \beta^l R_k{}^{\langle ij\rangle}{}_l + \lambda_0 \Theta \sigma^{ij} \right. \\
&\left. + \lambda_1 \sigma^{l\langle i} \sigma_l^{j\rangle} + \lambda_2 \omega^{l\langle i} \sigma_l^{j\rangle} + \lambda_3 \omega^{l\langle i} \omega_l^{j\rangle} + \lambda_4 a^{\langle i} a^{j\rangle} \right] \\
&+ \left[ \zeta_1 (T\beta \cdot \nabla)\Theta + \zeta_2 R + \zeta_3 T^2 R_{ij} \beta^i \beta^j + \xi_1 \Theta^2 + \xi_2 \sigma^2 + \xi_3 \omega^2 + \xi_4 a^2 \right] P^{ij} ,
\end{aligned}
\tag{385}
$$

where we have defined the symmetric traceless combination

$$
A^{\langle i} B^{j\rangle} = \frac{1}{2} P^{ik} P^{jl} \left( A_k B_l + A_j B_l \right) - \frac{1}{d-1} P^{ij} A^k B_k ,
\tag{386}
$$

and $\omega^2 = \omega^{ij} \omega_{ij}$. The first order transport coefficients are given by

$$
\begin{aligned}
\zeta = &\frac{2}{T} \left( r_5 + \frac{r_6}{d-1} + \frac{P'}{T^6 (P'')^2} \left( r_1 P' - 2r_2 T^3 P'' \right) \right) , \\
\eta = &\frac{r_6}{T} .
\end{aligned}
\tag{387}
$$

Second order transport coefficients associated with the traceless part of the constitutive relations are given by

$$
\begin{aligned}
\tau =& \frac{d_{8-}}{2T^2} + \frac{p_8}{T^2} - \frac{p_{10}}{T^2} - Tp_7' \,, \\
\kappa_1 =& -2p_7 \,, \\
\kappa_2 =& -2Tp_7' \,, \\
\lambda_0 =& \lambda_0(d_{2+}, d_{3+}, d_{8-}, d_{9-}, d_{10-}, p_7, p_8, p_{10}, t_1, m_7, m_{12}, m_{15}, m_{16}, m_{17}, m_{18}) \,, \\
\lambda_1 =& -\frac{1}{2}Tp_7' - \frac{m_{17}}{T^2} - \frac{m_{18}}{T^2} \,, \\
\lambda_2 =& -\frac{d_{3+}}{T^2} + \frac{d_{10-}}{T^2} - \frac{2p_8}{T^2} + \frac{2p_{10}}{T^2} + \frac{2t_3}{T} \,, \\
\lambda_3 =& \frac{4p_8}{T^2} + \frac{4p_{10}}{T^2} - 2Tp_7' \,, \\
\lambda_4 =& -\frac{2p_9}{T^4} + 2T^2 p_7'' + \frac{4p_8}{T^2} + \frac{4p_{10}}{T^2} + 4Tp_7' - \frac{2p_8'}{T} \,,
\end{aligned}
\tag{388}
$$

and transport which contributes to the trace of the stress tensor is given by

$$
\begin{aligned}
\zeta_1 =& \frac{2d_{4-}(P')^2}{T^7 (P'')^2} - \frac{4d_{5-}P'}{T^4 P''} + \frac{2d_{7-}}{T} + \frac{d_{8-}}{(d-1)T} \\
& + \frac{2p_{10}\left(-\frac{T^2}{d-1} + \frac{2TP'}{P''} - \frac{(P')^2}{(P'')^2}\right)}{T^3} + \frac{2(d-2)T^2 p_7'}{d-1} \\
& - \frac{2(d-2)p_8}{(d-1)T} + \frac{2p_9 P'(P' - TP'')}{T^5 (P'')^2} - \frac{2p_5(TP'' + P')^2}{T^3 (P'')^2} \,, \\
\zeta_2 =& p_7\left(\frac{d-3}{d-1} + \frac{P'}{TP''}\right) - \frac{p_7' P'}{P''} \,, \\
\zeta_3 =& p_7'\left(\frac{2(d-2)T}{d-1} + \frac{2P'}{P''}\right) + p_7\left(\frac{2P'}{TP''} - \frac{2}{d-1}\right) - \frac{2p_9 P'}{T^5 P''} + \frac{4p_8 P'}{T^3 P''} + \frac{4p_{10} P'}{T^3 P''} - \frac{2p_8' P'}{T^2 P''} \,, \\
\xi_1 =& \xi_1(d_{4-}, d_{5-}, d_{7-}, d_{8-}, p_5, p_7, \ldots, \\
& p_{10}, r_1, r_2, r_5, r_6, m_1, m_2, m_3, m_6, m_7, m_{10}, m_{12}, m_{14}, \ldots, m_{17}) \,, \\
\xi_2 =& \frac{d_{2+}P'}{4T^5 P''} + \frac{d_{9-}P'}{4T^5 P''} + \frac{d_{3+}P'}{4T^3 P''} - \frac{d_{10-}P'}{4T^3 P''} + \frac{1}{2}p_7'\left(\frac{(d-2)T}{d-1} + \frac{P'}{P''}\right) \\
& + \frac{t_1}{T} + \frac{m_7 P'}{2T^5 P''} - \frac{m_{15}}{2T^2} - \frac{m_{17}}{T^2(d-1)} - \frac{p_9 P'}{2T^5 P''} - \frac{p_8\left(T - \frac{3P'}{P''}\right)}{4T^3} \\
& + \frac{p_{10}(TP'' + 5P')}{4T^3 P''} - \frac{p_8' P'}{4T^2 P''} - \frac{p_{10}' P'}{4T^2 P''} + \eta\left(\frac{P' r_1}{T^8 (P'')^2} - \frac{r_2}{T^5 P''}\right) \,,
\end{aligned}
$$

$$
\begin{aligned}
\xi_3 =& \frac{p_8\left(\frac{(d-5)T}{d-1} + \frac{P'}{P''}\right)}{T^3} + \frac{p_{10}\left(\frac{(d-5)T}{d-1} + \frac{P'}{P''}\right)}{T^3} \\
& - p_7'\left(\frac{2(d-2)T}{d-1} + \frac{2P'}{P''}\right) + \frac{2p_9 P'}{T^5 P''} + \frac{p_8' P'}{T^2 P''} - \frac{p_{10}' P'}{T^2 P''} \,,
\end{aligned}
\tag{389a}
$$

and

$$\xi_4 = \frac{p_9 \left( \frac{(d-3)T}{d-1} - \frac{3P'}{P''} \right)}{T^5} + \frac{2p_8 \left( \frac{P'}{P''} - \frac{(d-3)T}{d-1} \right)}{T^3} + \frac{2p_{10} \left( \frac{P'}{P''} - \frac{(d-3)T}{d-1} \right)}{T^3} + \frac{p_8' \left( \frac{(d-3)T}{d-1} - \frac{2P'}{P''} \right)}{T^2}$$
$$+ p_7'' \left( -\frac{2(d-2)T^2}{d-1} - \frac{2TP'}{P''} \right) + p_7' \left( -\frac{2(d-3)T}{d-1} - \frac{4P'}{P''} \right) + \frac{p_9'P'}{T^4P''} - \frac{2p_{10}'P'}{T^2P''} + \frac{p_8''P'}{TP''} ,$$
$$\tag{389b}$$

where $d_{\pm i} = d_{\ell i} \pm d_{r i}$. The expressions for $\xi_1$ and $\lambda_0$ are exceptionally long and have been relegated to Appendix E.

An analysis almost identical to the one in the previous Subsection implies that positivity of the imaginary part of the effective action is ensured perturbatively in derivatives as long as

$$r_6 \geq 0, \qquad r_4 \geq 0, \qquad \begin{pmatrix} r_1 & r_2 \\ r_2 & r_5 \end{pmatrix} \succeq 0. \tag{390}$$

The last inequality implies that $\zeta \geq 0$ whereas the first one implies that $\eta \geq 0$.

At second order, as noticed by [27, 29], the coefficients $\kappa_1$, $\kappa_2$, $\lambda_3$, $\lambda_4$, $\zeta_2$, $\zeta_3$, $\xi_3$ and $\xi_4$ are completely determined in terms of $p_7$, $p_8$, $p_9$ and $p_{10}$ and are therefore not independent. In fact, five of these transport coefficients can be determined in terms of the other three,

$$\kappa_2 = T\kappa_1' ,$$
$$\zeta_2 = \kappa_1 \left( -\frac{d-3}{2(d-1)} - \frac{P'}{2TP''} \right) + \frac{\kappa_1'P'}{2P''} ,$$
$$\zeta_3 = \kappa_1' \left( \frac{P'}{P''} - \frac{(d-2)T}{d-1} \right) + \kappa_1 \left( \frac{1}{d-1} - \frac{P'}{TP''} \right) + \frac{T\kappa_1''P'}{P''} + \frac{\lambda_4 P'}{TP''} ,$$
$$\xi_3 = \frac{1}{4}\lambda_3 \left( \frac{d-5}{d-1} + \frac{3P'}{TP''} \right) - \frac{\lambda_3'P'}{4P''} - \frac{3T\kappa_1''P'}{4P''} + \frac{3}{4}\kappa_1' \left( T - \frac{2P'}{P''} \right) - \frac{\lambda_4 P'}{TP''} ,$$
$$\xi_4 = -\lambda_4 \left( \frac{d-3}{2(d-1)} + \frac{P'}{2TP''} \right) - \frac{\lambda_4'P'}{2P''} - \frac{\kappa_1'''T^2P'}{2P''} + \frac{1}{2}T\kappa_1'' \left( T - \frac{3P'}{P''} \right) . \tag{391}$$

We have not found any other relations between transport coefficients.

# 8 Summary and discussion

In this work, we have classified the possible constitutive relations according to their role in entropy production and whether they are constrained by an additional unitarity condition which we refer to as Schwinger-Keldysh positivity. We find that certain transport coefficients which do not generate entropy are nevertheless constrained to be positive semi-definite due to the latter condition. This is somewhat surprising since it implies that the set of phenomenological constraints usually imposed on the constitutive relations is necessary but not sufficient to constrain the transport coefficients of the hydrodynamic theory. In what follows we will briefly summarize our findings and discuss their implications.

## 8.1 Summary

Our findings can be summarized as follows. The constitutive relations of the conserved currents may be classified into two main classes: scalar and tensor terms. Scalar terms are of the form

$$J^A = \frac{1}{2} \frac{1}{\sqrt{-g}} \frac{\delta}{\delta F_A} \int d^d x \sqrt{-g} \left( L_0 + \widetilde{L}_0 \right) , \tag{392}$$

with the following definitions. The index $A$ is a multi index. If $J^A$ is a conserved charge current then $A$ specifies a single spacetime index. If $J^A$ is the stress tensor then $A$ specifies a pair of symmetrized spacetime indices. The Lagrangian term $L_0$ is a function of the metric $g_{\mu\nu}$ and possibly an external $U(1)$ field $B_\mu$ which we collectively denote by $F_A$. In addition $L_0$ depends on a temperature field $T$, a chemical potential $\mu$, a velocity field $u^\mu$ and derivatives thereof. In what follows we will use a rescaled velocity field and chemical potential

$$\beta^\mu = \frac{u^\mu}{T}, \qquad \nu = \frac{\mu}{T}. \tag{393}$$

We refer to $\widetilde{L}_0$ as the KMS-partner Lagrangian. It is obtained from $L_0$ by KMS conjugation

$$F_A \to \eta_A F_A, \qquad \beta^\mu \to -\eta_\mu \beta^\mu, \qquad \nabla_\mu \to \eta_\mu \nabla_\mu, \qquad T \to T, \qquad \nu \to -\nu, \tag{394}$$

where $\eta_X$ denotes the CPT eigenvalue of $X$. The variation with respect to $F_A$ acts on $(T, \nu, \beta^\mu)$ as

$$\frac{\partial T}{\partial g_{\mu\nu}} = \frac{1}{2} T^3 \beta^\mu \beta^\nu, \qquad \frac{\partial \nu}{\partial B_\mu} = \beta^\mu, \tag{395}$$

and other variations of the temperature, chemical potential or velocity field with respect to $F_A$ are zero. The scalar contributions to the transport coefficients do not produce entropy and they coincide with what [26, 27] refer to as scalar terms.

We find it convenient to further characterize transport according to its transformation properties under KMS conjugation (394). We will refer to currents which transform as

$$J^A(\nabla, F, \beta, T, \nu) = \eta_A J^A(\eta\nabla, \eta F, -\eta\beta, T, -\nu), \tag{396}$$

as having KMS-even parity and ones that transform as

$$J^A(\nabla, F, \beta, T, \nu) = -\eta_A J^A(\eta\nabla, \eta F, -\eta\beta, T, -\nu) \tag{397}$$

as having KMS-odd parity. The constitutive relations for the scalar terms are such that the currents constructed from them are always KMS-even. In the language of Section 6, we say that the scalar terms are KMS-even.

Tensor terms, which may be decomposed into KMS-even terms or KMS-odd terms, have the structure

$$
\begin{aligned}
J^A = & \frac{(-1)^a}{4} \nabla_{\mu_a} \dots \nabla_{\mu_1} \left( \sigma_\pm^{AB\mu_1\dots\mu_a\nu_1\dots\nu_b} \nabla_{\nu_1} \dots \nabla_{\nu_b} (\delta_\beta F_B) \right) \\
& \pm \frac{(-1)^b}{4} \nabla_{\nu_b} \dots \nabla_{\nu_1} \left( \sigma_\pm^{AB\mu_1\dots\mu_a\nu_1\dots\nu_b} \nabla_{\mu_1} \dots \nabla_{\mu_a} (\delta_\beta F_B) \right),
\end{aligned}
\tag{398}
$$

where the $\pm$ subscript on $\sigma$ specifies the $\pm$ sign in the second line of (398) and

$$\delta_\beta B_\mu = \nabla_\mu \nu - \frac{E_\mu}{T}, \qquad \delta_\beta g_{\mu\nu} = \nabla_\mu \left( \frac{u_\nu}{T} \right) + \nabla_\nu \left( \frac{u_\mu}{T} \right), \tag{399}$$

where $E_\mu = G_{\mu\nu} u^\nu$ is the local electric field with $G_{\mu\nu}$ the field strength associated to $B_\mu$. The various classes of transport are determined according to the KMS parity and index structure of $\sigma$. We stress that even though up until now we have used a $\pm$ subscript to denote KMS-parity of tensor terms, the $\pm$ subscript on $\sigma$ is not associated with KMS-parity, but rather, with symmetry properties of the indices of $\sigma$, as we now explain. Non-dissipative transport is characterized by $\sigma_\pm^{AB\mu_1\dots\nu_b} = \mp \sigma_\pm^{BA\mu_1\dots\nu_b}$ and is KMS-even. Exceptional transport is also characterized by $\sigma_\pm^{AB\mu_1\dots\nu_b} = \mp \sigma_\pm^{BA\mu_1\dots\nu_b}$ but is KMS-odd. Both

dissipative and pseudo-dissipative transport are characterized by $\sigma_\pm^{AB\mu_1...\nu_b} = \pm\sigma_\pm^{BA\mu_1...\nu_b}$ and has indefinite KMS-parity, but pseudo-dissipative transport satisfies the additional constraint

$$\sigma_\pm^{AB\mu_1...\mu_a\nu_1...\nu_b}(\nabla_{\nu_1}\ldots\nabla_{\nu_b}\delta_\beta F_B)(\nabla_{\mu_a}\ldots\nabla_{\mu_1}\delta_\beta F_a) = 0\,. \tag{400}$$

We summarize the various possible transport coefficients in the first two entries of Table 2.

Table 2: A classification of all possible tensor terms in the constitutive relations and some of their properties. Here $\sigma$ refers to the tensor structure appearing in the constitutive relations, KMS parity to the KMS parity of the tensor structure of the constitutive relations, $\Delta S$ to whether it contributes to entropy production, SK to whether it is constrained by the Schwinger-Keldysh positivity condition $\text{Im}(S_{eff}) \geq 0$ and Label, to the label of these coefficients in the main text. Pseudo-dissipative terms are also constrained by (400).

| Type | $\sigma$ symmetry | KMS parity | $\Delta S$ | SK | Label |
|---|---|---|---|---|---|
| Non-dissipative | $\sigma_\pm^{AB...} = \mp\sigma_\pm^{BA...}$ | $+$ | $0$ | ✗ | $L_+^{[A...]}$ |
| Dissipative | $\sigma_\pm^{AB...} = \pm\sigma_\pm^{BA...}$ | indefinite | $\geq 0$ | ✓ | $M_\pm^{(A...)}$ |
| Pseudo-dissipative | $\sigma_\pm^{AB...} = \pm\sigma_\pm^{BA...}$ | indefinite | $0$ | ✓ | $P_\pm^{(A...)}$ |
| Exceptional | $\sigma_\pm^{AB...} = \mp\sigma_\pm^{BA...}$ | $-$ | $0$ | ✓ | $N_-^{[A...]}$ |

The symmetry structure of $\sigma$, which appears in the second column of Table 2 specifies the transformation properties of $\sigma_\pm$ under a swap of its first two indices, e.g., symmetric dissipative, or antisymmetric exceptional. In the main text this has allowed us to further decompose transport into symmetric and antisymmetric subclasses. In the third column we have noted the KMS parity of the various terms in the constitutive relations. The KMS parity follows from the underlying KMS symmetry of the action which also leads to the Onsager reciprocity relations. From the KMS parity we can determine how the various transport coefficients transform under CPT. The CPT-eigenvalue of a transport coefficient is simply the KMS parity of the term it appears in, times the KMS parity of the tensor structure it multiplies. CPT-even coefficients are even functions of chemical potential, and CPT-odd coefficients are odd functions of chemical potential.

The fourth and fifth column of Table 2 refer to the constraints imposed on the various transport coefficients. The classification of transport has been carried out with respect to these constraints. Only dissipative terms contribute to entropy production and are constrained such that entropy production is positive. All but non-dissipative terms are constrained by positivity of the imaginary part of the effective action,

$$\text{Im}(S_{eff}) \geq 0, \tag{401}$$

which we have termed Schwinger-Keldysh positivity. Exceptional terms are not constrained by the entropy production condition but are nevertheless constrained by the Schwinger-Keldysh positivity condition. Finally, pseudo-dissipative terms are very similar in their structure to dissipative terms but do not contribute to entropy production. They are constrained by the Schwinger-Keldysh positivity condition and satisfy, in addition, (400).

For ease of reference, we have included in the last column of Table 2 the labels used in the main text for the various types of transport. Non-dissipative terms were discussed

in detail in 6.2.2, dissipative terms in 6.2.3 and pseudo-dissipative and exceptional terms in 6.2.4. We computed the entropy production in (242) and the imaginary part of the effective action in (244).

The canonical examples of symmetric dissipative terms are the shear viscosity, bulk viscosity, and conductivity. As far as we know antisymmetric dissipative terms have not been studied or classified. A relatively simple example of an antisymmetric dissipative term appears in the second-order hydrodynamics of a charged, parity-violating fluid in three dimensions. It is given by

$$J^\mu = -\frac{1}{2}\nabla_\sigma\left[\xi^+ \epsilon^{\mu\nu\rho}u_\rho u^\sigma\right]\left(\partial_\nu\left(\frac{\mu}{T}\right)-\frac{E_\nu}{T}\right)-\xi^+ \epsilon^{\mu\nu\rho}u_\rho u^\sigma\nabla_\sigma\left(\partial_\nu\left(\frac{\mu}{T}\right)-\frac{E_\nu}{T}\right), \quad (402)$$

where $P_{\mu\nu} = g_{\mu\nu}+u_\mu u_\nu$ is the projection operator orthogonal to the velocity field, and $\epsilon^{\mu\nu\rho}$ is the Levi-Civita tensor. The superscript on the transport coefficient $\xi^+$ indicates that $\xi^+(T,\nu) = \xi^+(T,-\nu)$. An example of a pseudo-dissipative term can be found in Section 6.2.4. The contribution to the $U(1)$ current,

$$J^\mu = \frac{\zeta^-}{T}(u^\alpha\partial_\alpha\nu)\epsilon^{\mu\nu\rho}u_\nu\left(E_\rho - T\partial_\rho\left(\frac{\mu}{T}\right)\right), \quad (403)$$

with $\zeta^-$ CPT-odd does not generate dissipation but is nevertheless constrained by Schwinger-Keldysh positivity. This contribution is symmetric pseudo-dissipative. In the particular example given in Section 6.2.4, where the ordinary conductivity vanishes, Schwinger-Keldysh positivity sets $\zeta^- = 0$.

It is interesting to note that there are KMS-odd dissipative terms, as well as KMS-even ones. At leading order in derivatives, all dissipative terms are KMS-odd. This well-known fact is usually attributed to a breaking of time-reversal invariance by dissipation. However, note that at higher order in derivatives dissipative terms of either KMS-parity are allowed by the symmetries of the problem. Moreover, our entire analysis did not require any input on the CPT transformation properties of the currents. Rather, it required a certain $\mathbb{Z}_2$ symmetry which is a combination of the KMS condition and CPT covariance of the Schwinger-Keldysh generating functional.

Non-dissipative terms do not generate entropy and are unconstrained by the Schwinger-Keldysh positivity condition. In the language of [26, 27] the symmetric non-dissipative terms are similar to $\bar{H}_V$ and the antisymmetric non-dissipative are similar to Berry-type. An example of a symmetric non-dissipative term is

$$J^\mu = \frac{1}{2}\nabla_\lambda(\chi^+ P^{\mu\nu}u^\lambda)\left(\partial_\nu\left(\frac{\mu}{T}\right)-\frac{E_\nu}{T}\right)+\chi^+ P^{\mu\nu}u^\lambda\nabla_\lambda\left(\partial_\nu\left(\frac{\mu}{T}\right)-\frac{E_\nu}{T}\right), \quad (404)$$

where the superscript on $\chi^+$ indicates that it is CPT-even.

The exceptional terms are KMS-odd. These terms do not contribute to the entropy production but are nevertheless constrained by the Schwinger-Keldysh positivity condition. An example of an antisymmetric exceptional term is

$$T^{\mu\nu} = \ldots + \gamma^-\left(P^{\mu\nu}\sigma^2 - 2\Theta\sigma^{\mu\nu}\right), \quad (405)$$

with $\gamma^-$ a CPT-odd tranport coefficient. In the particular setup described in 6.2.4, where the ordinary viscosities vanish, the Schwinger-Keldysh positivity condition enforces $\gamma^- = 0$.

## 8.2 Discussion

The classification scheme we have presented in the previous Subsection seems to have some overlap with that of [26, 27]. The scalar terms nicely match the scalar terms of [26, 27], the non-dissipative terms are somewhat similar to the $\bar{H}_V$ and $B$ type transport of [26, 27] and dissipative terms are similarly defined both here and in the work of [26, 27]. To the best of our knowledge, a discussion of antisymmetric dissipative terms has not appeared prior to this work. An additional difference between the current work and earlier ones is that, in addition to the tensor structure of the constitutive relations, we have also characterized transport according to its transformation properties under CPT and KMS. The constitutive relations characterized as scalar or non-dissipative are KMS-even while the dissipative terms can be KMS-even or KMS-odd.

In addition to scalar, dissipative and non-dissipative classes we have demonstrated that there are two novel classes, which we have referred to as pseudo-dissipative terms and exceptional terms. Pseudo-dissipative transport can be KMS-even or odd while exceptional terms are KMS-odd. However, unlike dissipative terms, these expressions do not produce entropy. Nevertheless, they are constrained by unitarity. This is the first instance where positivity of the divergence of the entropy current is not sufficient to determine the sign of transport coefficients. Our analysis so far is very preliminary. We have shown that in extreme circumstances wherein the ordinary conductivity or viscosities vanish, the Schwinger-Keldysh positivity condition constrains exceptional transport coefficients to vanish. It is not yet clear to us whether there exists an example of transport which is constrained by the Schwinger-Keldysh positivity condition to have a semi-definite sign (as opposed to being strictly zero). Though, we can demonstrate that certain terms which contribute to stochastic noise (essentially the $F_a$ type terms which we have not discussed in this work) are constrained to be sign semi-definite under the Schwinger-Keldysh positivity condition. We hope to report on progress on this front in the near future.

One cannot help speculate about the realization of the Schwinger-Keldysh positivity condition in a hydrodynamic context. Perhaps one needs to consider all four components of the entropy current discussed in Section 5 in order to capture all constraints associated with unitarity. Likewise, it is not clear how the Schwinger-Keldysh positivity condition will be realized in holography. There, entropy production is associated with area increase theorems of the event horizon. While transport is guaranteed to satisfy the Schwinger-Keldysh positivity condition by unitarity of the dual CFT, it is an open question whether or not this condition can be geometrized in the gravity dual.

Our treatment of transport, which follows from the Schwinger-Keldysh generating functional, allows for a direct connection between the standard phenomenological model of hydrodynamics and the Schwinger-Keldysh effective action. Be that as it may, the Schwinger-Keldysh effective action provides much more information on the dynamics of the system than captured by hydrodynamics. Apart from an off-shell formulation, and a self-consistent incorporation of stochastic noise, it also allows one to study quantum effects associated with hydrodynamics. The latter requires an effective action which is valid beyond the statistical mechanical limit discussed in this paper (see however [14, 18]).

Given that we have worked in the limit of small $\hbar$, one should query the validity of the statistical mechanical limit for conformal field theories (CFTs). The statistical mechanical limit assumes a separation of scales where the inverse temperature (in units of $\hbar$) is much smaller than the mean free path which is much smaller than the size of the system. The separation of scales is needed in order to allow for a derivative expansion (whose control parameter is the mean free path) after setting $\hbar$ small. In CFTs the mean free path is controlled by $\hbar$ and the hydrodynamical variables. This implies that there is no separation of scales which implies, in turn, that one can not implement a consistent

derivative expansion. Once again, this raises the question of the relation between the formulation presented in this work and a hydrodynamic description of large $N$ gauge theories with finite $\hbar$, which in some cases can be computed holographically [52]. Naively, one may hope that in these instances large $N$ may replace small $\hbar$ in order to generate a co-aligned limit of sources as described in Section 2.2. In this case, the obstruction for generalizing the analysis from this paper will come from thermal translations associated with factors of $e^{i\hbar\delta_\beta}$ which do not become infinitesimal.

Since the effective action includes more information about the dynamics of the system than that captured by the classical hydrodynamics, the number of free parameters in the Lagrangian exceeds the number of transport coefficients in hydrodynamics at the same order in the derivative expansion. For instance, in the example presented in Section 7.1 for the first-order hydrodynamics of a $2+1$ dimensional parity-violating fluid, the Lagrangian contained 23 parameters but only 9 transport coefficients. Parity-preserving Lagrangians describing neutral fluids at second order in the derivative expansion, described in Section 7.2 are characterized by over 40 parameters which should be compared to the 12 independent transport coefficients of the hydrodynamic theory. It would certainly be of value to be able to identify the parameters of the Lagrangian which contribute to transport. A partial discussion of such an analysis was carried out in Section 7. A more robust analysis is certainly called for.

## Acknowledgments

We would like to thank A. Abanov, J. Cotler, P. Glorioso, S. Hartnoll and H. Liu for many enlightening conversations. The work of KJ was supported in part by the US Department of Energy under grant number DE-SC0013682. The work of NPF in KU Leuven was supported in part by the National Science Foundation of Belgium (FWO) grant G.001.12 Odysseus and by the European Research Council grant no. ERC-2013-CoG 616732 HoloQosmos. The work of NPF was also supported in part by the Israel Science Foundation under grant 504/13 and in part at the Technion by a fellowship from the Lady Davis Foundation. The work of AY and RM was supported in part by the Israeli Science Foundation under an ISF-UGC grant 630/14, an ISF excellence center grant 1989/14 and a BSF grant 2016324.

## A  Constraints on the imaginary part of the effective action

The Schwinger-Keldysh partition function satisfies the inequality (8a),

$$|Z| \leq 1 \,. \tag{406}$$

Since this equation plays a central role in our work we reproduce its proof which, as far as we are aware of, was first carried out in [25].

Consider the quantity

$$A = \text{Tr}\left(U^\dagger \rho V\right) \,, \tag{407}$$

where $U$ and $V$ are unitary operators and $\rho$ is a density matrix. We write $\rho$ in its eigenbasis, $\rho = \sum_n r_n |n\rangle\langle n|$. Then

$$A = \sum_{m,n} r_n \langle n|V|m\rangle\langle m|U^\dagger|n\rangle \,. \tag{408}$$

Using the Cauchy-Schwarz inequality we find

$$|A|^2 \leq \left( \sum_m r_m \left| \langle m|VV^\dagger|m \rangle \right|^2 \right) \left( \sum_m r_m \left| \langle m|U^\dagger U|m \rangle \right|^2 \right) = 1 \,, \tag{409}$$

where the second equality follows from unitarity of $U$ and $V$ and $\sum_n r_n = 1$.

Given (1),

$$Z[A_1, A_2] = \text{Tr}\left( U[A_1] \rho_{-\infty} U^\dagger[A_2] \right) \,, \tag{410}$$

it follows by the above lemma that

$$|Z| \leq 1 \,, \tag{411}$$

which is what we set out to prove.

## B  Diffeomorphisms and the action of $Q$

In this Appendix we obtain the transformation laws for the ghosts $X_{\bar{g}}^\mu$, $X_g^\mu$, $C_{\bar{g}}$ and $C_g$ under target space diffeomorphisms and $U(1)$ transformations, as well as the modified transformation laws of the $a$-type fields due to the ghosts. We assume that $Q$ commutes with $r$-type diffeomorphisms and $U(1)$ transformations, whose action we denote by $\delta_r$, but not $a$-type transformations, which we denote by $\delta_a$.

We begin with the $X_{\bar{g}}^\mu$-ghosts. Recall that the action of $Q$ on the $X$-supermultiplet is given by

$$[Q, X_r^\mu] = X_{\bar{g}}^\mu \,, \qquad \{Q, X_{\bar{g}}^\mu\} = [Q, X_a^\mu] = 0 \,, \qquad \{Q, X_g^\mu\} = X_a^\mu \,,$$

and that the action of $r$-type transformations on $X_r^\mu$,

$$\delta_r X_r^\mu(\sigma) = -\xi_r^\mu(X_r(\sigma)) \,.$$

Acting with $\delta_r$ on the first commutator, we find

$$\delta_r X_{\bar{g}}^\mu = \delta_r[Q, X_r^\mu] = [Q, \delta_r X_r^\mu] = [Q, -\xi_r^\mu(X_r)] = -X_{\bar{g}}^\nu \partial_\nu \xi_r^\mu(X_r(\sigma)) \,, \tag{412}$$

where in the second equality we used the assumption that $Q$ is inert under $\delta_r$.

Let us now turn our attention to the transformation laws for $X_a^\mu$ under $\delta_r$. By assumption is given by

$$\delta_r X_a^\mu = -X_a^\nu \partial_\nu \xi_r^\mu(X_r(\sigma)) + \text{ghosts} \,. \tag{413}$$

By assumption, $\delta_r$ commutes with $Q$ and so $[Q, \delta_r X_a] = 0$. A straightforward computation yields

$$\begin{aligned} [Q, -X_a^\nu \partial_\nu \xi_r^\mu(X_r(\sigma))] &= -X_a^\nu \partial_\nu (X_{\bar{g}}^\rho \partial_\rho \xi_r^\mu(X_r(\sigma))) + X_a^\nu (\partial_\nu X_{\bar{g}}^\rho) \partial_\rho \xi_r^\mu(X_r(\sigma)) \\ &= -X_{\bar{g}}^\nu X_a^\rho \partial_\nu \partial_\rho \xi_r^\mu(X_r(\sigma)) \end{aligned} \tag{414}$$

(The first term in the variation comes from $[Q, \xi_r^\mu(X_r(\sigma))]$, and the second from $[Q, \partial_\mu]$, on using that $\partial_\mu = \frac{\partial}{\partial X_r^\mu}$.) A similar computation shows

$$[Q, -X_{\bar{g}}^\nu X_g^\rho \partial_\nu \partial_\rho \xi_r^\mu(X_r(\sigma))] = X_{\bar{g}}^\nu X_a^\rho \partial_\nu \partial_\rho \xi_r^\mu(X_r(\sigma)) \,, \tag{415}$$

and so the $r$-variation of $X_a^\mu$ must be

$$\delta_r X_a^\mu = -X_a^\nu \partial_\nu \xi_r^\mu(X_r(\sigma)) - X_{\bar{g}}^\nu X_g^\rho \partial_\nu \partial_\rho \xi_r^\mu(X_r) + \{Q, \mathcal{X}_g^\mu\} \,. \tag{416}$$

Given $[Q, X_g^\mu] = X_a^\mu$ and $[\delta_Q, \delta_r] = 0$, it is straightforward to compute the $r$ variation of $X_g^\mu$. We find

$$\delta_r X_g^\mu = -X_g^\nu \partial_\nu \xi_r^\mu(X_r(\sigma)) + \mathcal{X}_g^\mu. \tag{417}$$

We choose $\mathcal{X}_g^\mu = 0$. Putting the pieces together, we find that the $r$-transformation laws of the ghosts are

$$\delta_r X_{\bar{g}}^\mu = -X_{\bar{g}}^\nu \partial_\nu \xi_r^\mu(X_r(\sigma)), \qquad \delta_r X_g^\mu = -X_g^\nu \partial_\nu \xi_r^\mu(X_r(\sigma)), \tag{418}$$

and that the transformations of the $a$-fields are modified in the presence of the ghosts as

$$\delta_r X_r^\mu = -\xi_r^\mu(X_r(\sigma)), \qquad \delta_r X_a^\mu = -X_a^\nu \partial_\nu \xi_r^\mu(X_r(\sigma)) - X_{\bar{g}}^\nu X_g^\rho \partial_\nu \partial_\rho \xi_r^\mu(X_r(\sigma)). \tag{419}$$

The $a$-fields in (33) vary under $a$-transformations, and as a result we do not require that $a$-transformations commute with $Q$. In this work we take a simple choice for the action of $a$-transformations on the $X$-supermultiplet: we take the ghosts to be inert, and the bosonic fields to vary according to

$$\delta_a X_r^\mu = 0, \qquad \delta_a X_a^\mu = -\xi_a^\mu(X_r(\sigma)). \tag{420}$$

A similar computation shows that the $C$-ghosts vary under $r$-transformations as

$$\delta_r C_{\bar{g}} = -X_{\bar{g}}^\mu \partial_\mu \Lambda_r(X_r(\sigma)), \qquad \delta_r C_g = -X_g^\mu \partial_\mu \Lambda_r(X_r(\sigma)), \tag{421}$$

and the transformation laws of the bosonic $C$'s are modified as

$$\delta_r C_r = -\Lambda_r(X_r(\sigma)), \qquad \delta_r C_a = -X_a^\mu \partial_\mu \Lambda_r(X_r(\sigma)) - X_{\bar{g}}^\mu X_g^\nu \partial_\mu \partial_\nu \Lambda_r(X_r(\sigma)). \tag{422}$$

We take the ghosts to be invariant under $a$-transformations, and for the bosonic fields to vary as

$$\delta_a C_r = 0, \qquad \delta_a C_a = -\Lambda_a(X_r(\sigma)). \tag{423}$$

## C  Comparison with previous work

In this Appendix we compare our construction to the work of [19, 21]. Let us start by considering the explicit form of the superfields. Expanding in components we have

$$\mathbb{B}_i = B_{r\,i} + \theta(\mathcal{L}_{\bar{\psi}} B_{r\,\mu} \partial_i X_r^\mu + \partial_i C_{\bar{g}}) + \bar{\theta}(\mathcal{L}_\psi B_{r\,\mu} \partial_i X_r^\mu + \partial_i C_g) + \bar{\theta}\theta \mathcal{B}_{a\,i}, \tag{424}$$

where we have defined

$$\begin{aligned}
\mathcal{B}_{a\,i} &= (\mathcal{L}_{X_a} B_{r\,\mu} + \delta_{\psi\bar{\psi}} B_{r\,\mu} + B_{a\,\mu}) \partial_i X_r^\mu + \partial_i C_a, \\
\partial_i X_r^\mu \delta_{\psi\bar{\psi}} B_{r\,\mu} &\equiv \partial_i X_r^\mu \bar{\psi}^j \psi^k \partial_j \partial_k B_{r\,\mu} + \bar{\psi}^j (\partial_i X_g^\mu) \partial_j B_{r\,\mu} + (\partial_i X_{\bar{g}}^\mu) \psi^j \partial_j B_{r\,\mu}.
\end{aligned} \tag{425}$$

By comparing this result to Eqs. (5.35) and (5.46) of [19], and by substituting $\gamma \leftrightarrow \bar{\psi}, \gamma_a \leftrightarrow \psi$ and noticing that $\delta_{\psi\bar{\psi}} \leftrightarrow \mathcal{L}_{\gamma\gamma_a}$, we see that the equations agree up to a minus sign of the top component which arises from the fact that the authors of [19] use a different convention for the $\bar{Q}$ charge, namely $\delta_{\bar{Q}} = \partial_{\bar{\theta}} - i\theta \delta_\beta$.

Next, let us compare the structure of the KMS symmetry. In the statistical mechanical limit we have

$$K(B_{r\,i}(\sigma)) = \eta_i B_{r\,i}(\sigma), \qquad K(B_{a\,i}(\sigma)) = \eta_i(B_{a\,i}(\sigma) + i\delta_\beta B_{r\,i}(\sigma)), \tag{426}$$

and for the dynamical variables

$$K(X_r^\mu(\sigma)) = \eta_\mu X_r^\mu(\sigma), \quad K(X_a^\mu(\sigma)) = \eta_\mu \left( X_a^\mu(\sigma) + i\beta^i \partial_i X_r^\mu \right),$$
$$K(C_r(\sigma)) = C_r(\sigma), \qquad K(C_a(\sigma)) = C_a(\sigma) + i\beta^i \partial_i C_r(\sigma). \tag{427}$$

On the other hand, the dynamical KMS transformations of [19, 21] are given by (using their notation)

$$\tilde{X}_r^\mu(\sigma) = -X_r^\mu(-\sigma), \qquad \tilde{X}_a^\mu(-\sigma) = -X_a^\mu(-\sigma) - i\beta^\mu(-\sigma) + i\beta_0^\mu,$$
$$\tilde{\varphi}_r(\sigma) = -\varphi_r(-\sigma), \qquad \tilde{\varphi}_a(\sigma) = -(\varphi_a(-\sigma) + i\beta^i \partial_i \varphi_r(-\sigma)),$$
$$\tilde{A}_{r\mu}(x) = A_{r\mu}(-x), \qquad \tilde{A}_{a\mu}(x) = A_{a\mu}(-x) + i\mathcal{L}_{\beta_0} A_{r\mu}(-x), \tag{428}$$

where $\varphi_{r,a} \leftrightarrow C_{r,a}$ in our notation and $\beta(\sigma) \equiv \beta_0 e^{\tau(\sigma)}, \beta^\mu = \beta(\sigma)u^\mu$. Clearly the dynamical KMS transformation differs from our $K$. However, the dynamical KMS transformation of [19, 21] includes a PT flip, rather than a CPT flip, and moreover is formulated in even spacetime dimension where one can take $\eta_i = \eta_\mu = -1$. Accounting for these facts, we find that our $K$ acts on the dynamical fields in the same way as their dynamical KMS under the integral. The only minor difference is that in our formalism $\delta_\beta$ includes a flavor transformation $\Lambda_\beta$ aside from the Lie derivative along $\beta^i$.

Let us turn now to the full KMS invariance of the action. Recall that we constructed a full KMS invariant action (136). This is in agreement with the analysis of [21] around equation (5.10), where they show that the following conditions are sufficient to ensure full KMS invariance

$$\mathcal{L} = \frac{1}{2}(\mathcal{L}_c + \tilde{\mathcal{L}}_c), \qquad (\tilde{\mathcal{L}}_c)_{B_a=0} = \partial_\mu V_0^\mu. \tag{429}$$

From the analysis presented in Subsection 2.5, it is not hard to see that our construction satisfies these two conditions.

## D   The structure of tensor terms

The most general expression for rank n tensor terms involving derivatives is

$$\int d^d\sigma d\theta d\bar{\theta} L^{ABC_1\cdots} D_\theta \mathbb{F}_A D_{\bar{\theta}} \mathbb{F}_B D\mathbb{F}_{C_1} \dots$$
$$= \int d^d\sigma d\theta d\bar{\theta} L(\mathbb{W}^m D_\theta \mathbb{W}^n \mathbb{F})(\mathbb{W}^p D_{\bar{\theta}} \mathbb{W}^q \mathbb{F})(\mathbb{W}^{s_1} D_\theta \mathbb{W}^{s_2} D_{\bar{\theta}} \mathbb{W}^{s_3} \mathbb{F})\dots \tag{430}$$

where $L^{ABC_1\cdots}$ is a differential operator and $L$ a scalar and we have refrained from writing most of the indices on the right-hand side of the equation to avoid clutter. The goal of this appendix is to show that this structure is redundant and that we may, using integration by parts, remove most of the derivatives.

Consider the two-tensor term

$$\int d^d\sigma d\theta d\bar{\theta} L^{AB} D_\theta \mathbb{F}_A D_{\bar{\theta}} \mathbb{F}_B = \int d^d\sigma d\theta d\bar{\theta} L(\mathbb{W}^m D_\theta \mathbb{W}^n \mathbb{F})(\mathbb{W}^p D_{\bar{\theta}} \mathbb{W}^q \mathbb{F}). \tag{431}$$

Given

$$\partial_{\bar{\theta}} \mathbb{W}_i \mathbb{F}_j = \mathbb{W}_i \partial_{\bar{\theta}} \mathbb{F}_j + \mathbb{F}_k \partial_{\bar{\theta}} \mathbb{\Gamma}^k{}_{ij}, \tag{432}$$

we want to show that the rightmost piece in (432) can be absorbed into $L(\mathbb{F}, \partial)$. To do so, it is useful to use the identity

$$\delta\mathbb{\Gamma}^k{}_{ij} = \frac{1}{2}\mathfrak{g}^{km}(\mathbb{W}_j \delta\mathfrak{g}_{im} + \mathbb{W}_i \delta\mathfrak{g}_{jm} - \mathbb{W}_m \delta\mathfrak{g}_{ij}), \tag{433}$$

which can be established by direct computation. By taking $\delta = \partial_{\bar{\theta}}$ and applying (433) to (432), we may rewrite (432) as

$$D_{\bar{\theta}} \mathbb{V}_i \mathbb{F}_j = \mathbb{V}_i D_{\bar{\theta}} \mathbb{F}_j + L_s(\mathbb{F}, \mathbb{V}) D_{\bar{\theta}} \mathfrak{g}_{ij} \tag{434}$$

with $L_s$ a scalar differential operator. Therefore, one can absorb $L_s$ into a redefinition of $L$. Since $[D_\theta, \partial_i] = 0$, a similar argument holds for $D_\theta$. Hence, without loss of generality, a generic rank 2 tensor term with derivatives can be written as

$$\int d^d \sigma d\theta d\bar{\theta} L^{AB} D_\theta \mathbb{F}_A D_{\bar{\theta}} \mathbb{F}_B = \int d^d \sigma d\theta d\bar{\theta} L(\mathbb{V}^m D_\theta \mathbb{F})(\mathbb{V}^p D_{\bar{\theta}} \mathbb{F})$$
$$\sim - \int d^d \sigma d\theta d\bar{\theta} L' D_\theta \mathbb{F}(\mathbb{V}^{p'} D_{\bar{\theta}} \mathbb{F}) . \tag{435}$$

To complete this discussion, one should also consider higher tensor terms. These are generated by introducing the operator $D = D_\theta D_{\bar{\theta}}$. In a generic situation, the differential operator, say, $L^{ABC_1}$, may act on each of the superderivatives,

$$L^{ABC_1} D\mathbb{F}_C = L\mathbb{V}^{m_1} D_\theta \mathbb{V}^{m_2} D_{\bar{\theta}} \mathbb{V}^{m_3} \mathbb{F} , \tag{436}$$

where we have suppressed unwanted indices. In this case, one can commute the covariant derivatives past the super-derivatives in a similar fashion to (435). The only terms which one might worry about in this process are those arising from

$$\partial_\theta \mathbb{V} \partial_{\bar{\theta}} \mathbb{F} = \mathbb{V}(\partial_\theta \partial_{\bar{\theta}} \mathbb{F}) + \partial_\theta \mathbb{\Gamma} \partial_{\bar{\theta}} \mathbb{F} = \mathbb{V}(\partial_\theta \partial_{\bar{\theta}} \mathbb{F}) + L_s(\mathbb{F}, \mathbb{V}) \partial_\theta \mathfrak{g} \partial_{\bar{\theta}} \mathbb{F} . \tag{437}$$

But recall that higher tensor terms always contain a $D_\theta \mathbb{F} D_{\bar{\theta}} \mathbb{F} \propto \bar{\theta} \theta$ factor. This in turn implies that the rightmost piece of (437) will inevitably produce a pure ghost contribution. Thus, to summarize, for higher tensor terms, one can restrict the derivative structure to $\mathbb{V}^m D\mathbb{F}$ without any loss of generality.

# E   Explicit expressions for $\xi_1$ and $\lambda_0$

In Section 7.2 we have computed the transport coefficients associated with second order neutral fluids, as follows from variation of the Schwinger Keldysh effective action. The expressions for $\xi_1$ and $\lambda_0$ were rather long and have been omitted from the main text. We

present them here.

$$
\begin{aligned}
\xi_1 =\,& \frac{2m_1\,(P')^3}{T^{11}\,(P'')^3} - \frac{d'_{4-}\,(P')^3}{T^8\,(P'')^3} - \frac{p'_9\,(P')^3}{T^6\,(P'')^3} + \frac{2p''_7\,(P')^2}{(P'')^2} - \frac{p''_8\,(P')^2}{T^2\,(P'')^2} \\
& - \frac{2d_{5-}\left(6P'' + TP^{(3)}\right)(P')^2}{T^6\,(P'')^3} \\
& + \frac{d_{4-}\left(P'\left(9P'' + 2TP^{(3)}\right) - T\,(P'')^2\right)(P')^2}{T^9\,(P'')^4} - \frac{2m_2\,(P')^2}{T^8\,(P'')^2} \\
& - \frac{4m_3\,(P')^2}{T^8\,(P'')^2} + \frac{2r_1^2\,(P')^2}{T^{16}\,(P'')^3} + \frac{p'_5\,(P' + TP'')^2\,P'}{T^4\,(P'')^3} + \frac{\left(2P'd'_{5-} - T^3d'_{7-}P''\right)P'}{T^5\,(P'')^2} \\
& + \frac{p'_{10}\left((P')^2 + \frac{T^2(P'')^2}{d-1}\right)P'}{T^4\,(P'')^3} + \frac{p'_8\left((d-4)T\,(P'')^2 + (d-1)P'\left(5P'' + TP^{(3)}\right)\right)P'}{(d-1)T^3\,(P'')^3} \\
& + \frac{p_9\left(-\frac{2T^2(P'')^3}{d-1} - TP'\,(P'')^2 + (P')^2\left(7P'' + 2TP^{(3)}\right)\right)P'}{T^7\,(P'')^4} \\
& + \frac{4m_6P'}{T^5P''} + \frac{4m_7P'}{(d-1)T^5P''} + \frac{2m_{10}P'}{T^5P''} + \frac{2m_{12}P'}{(d-1)T^5P''} \\
& - \frac{d'_{8-}P'}{2(d-1)T^2P''} - \frac{4r_1r_2P'}{T^{13}\,(P'')^2} - \frac{2m_{14}}{T^2} - \frac{4m_{15}}{(d-1)T^2} - \frac{2m_{16}}{(d-1)T^2} \\
& - \frac{4m_{17}}{(d-1)^2T^2} + \zeta\left(\frac{2r_1P'}{T^8\,(P'')^2} - \frac{2r_2}{T^5P''}\right) + \frac{d_{7-}\left(T + \frac{3P'}{P''}\right)}{T^3} \\
& + \frac{d_{8-}(3P' + TP'')}{2(d-1)T^3P''} + p'_7\left(-\frac{2\left(P'' + TP^{(3)}\right)(P')^2}{T\,(P'')^3} + \frac{4P'}{(d-1)P''} + \frac{2(d-2)T}{(d-1)^2}\right) \\
& + \frac{p_8\left(-(d-2)T^2\,(P'')^3 + (10-3d)TP'\,(P'')^2 - 2(d-1)\,(P')^2\left(4P'' + TP^{(3)}\right)\right)}{(d-1)T^4\,(P'')^3} \\
& - \frac{p_5\,(P' + TP'')\left(T^2\,(P'')^3 + 2TP'\,(P'')^2 + (P')^2\left(5P'' + 2TP^{(3)}\right)\right)}{T^5\,(P'')^4} \\
& + \frac{p_{10}\left(-\frac{T^3}{d-1} + \frac{P'T^2}{(d-1)P''} + \frac{(P')^2T}{(P'')^2} - \frac{(P')^3\left(5P'' + 2TP^{(3)}\right)}{(P'')^4}\right)}{T^5} + \frac{2r_2^2}{T^{10}P''}\,,
\end{aligned}
$$

(438)

and

$$
\begin{aligned}
\lambda_0 =\,& - \frac{d_{9-}P'}{2T^5P''} + \frac{d_{8-}(TP'' + 3P')}{4T^3P''} + \frac{d_{10-}P'}{2T^3P''} - \frac{P'd'_{8-}}{4T^2P''} - \frac{4m_{17}}{(d-1)T^2} \\
& + p'_7\left(\frac{P'}{P''} - \frac{2T}{d-1}\right) - \frac{P'\left(d_{2+} + T^2d_{3+}\right)}{2T^5P''} + \frac{m_7P'}{T^5P''} + \frac{m_{12}P'}{T^5P''} - \frac{m_{15}}{T^2} - \frac{m_{16}}{T^2} \\
& + \frac{p_8\left(\frac{2P'}{P''} + T\right)}{T^3} - \frac{p_{10}\left(\frac{2P'}{P''} + T\right)}{T^3} - \frac{p'_8P'}{T^2P''} + \frac{p'_{10}P'}{T^2P''} - \frac{2t_1}{T}\,.
\end{aligned}
$$

(439)

# F  Anomaly-induced transport

In this Appendix we write down the Schwinger-Keldysh effective action for field theories with 't Hooft anomalies for continuous symmetries. There is by now a wealth of literature concerning anomaly-induced transport in relativistic hydrodynamics, including e.g. [28, 36, 40–45, 59–64]. Here we build upon the results of [44], who proposed a Schwinger-Keldysh effective action for flavor anomalies at finite $\hbar$ and without ghosts. Those authors used somewhat different fluid variables than ours, and it is not clear how to account for gravitational anomalies within their formalism. Nevertheless it is easy to follow their lead and write a proposal for the bosonic part of an anomaly action for any anomaly polynomial with our fluid variables.

We proceed in three steps. First, we write down such a bosonic effective action for any anomaly polynomial. Second we show that this action is invariant under the KMS symmetry. Finally, we take the statistical mechanical limit of this action and "supersymmetrize" it. The end result is an effective action which correctly reproduces the anomalies as well as all of the other symmetries of the problem, albeit in the $\hbar \to 0$ limit. Throughout we set the background field $\mathbb{A}$ to vanish. (The results of [45] suggest a connection between some of the properties of gravitational anomalies and $\mathbb{A}$. Probing this connection is certainly worthwhile but is beyond the scope of this work.)

By construction the hydrodynamic constitutive relations that follow from this action, after setting the ghosts and $a$-fields to vanish, are precisely those previously obtained in the literature. Since we do not learn anything new about hydrodynamics per se, we consider this Appendix an existence proof, demonstrating that a Schwinger-Keldysh anomaly effective action exists.

We begin with a review of anomalies. See e.g. [36]. Quantum field theories in even spacetime dimension $d = 2n$ may possess 't Hooft anomalies for continuous global symmetries. For our purposes, these anomalies are most efficiently described via the inflow mechanism. To illustrate the idea let us consider a Euclidean field theory on a manifold $\mathcal{M}_d$ of dimension $d$ with no boundary (anomalies on manifolds with boundaries pose special problems [65] and will not be discussed here). Let $\mathbf{B} = B_\mu dx^\mu$ be the background flavor gauge field and $\mathbf{\Gamma}^\mu{}_\nu = \Gamma^\mu{}_{\nu\rho} dx^\rho$ the Christoffel connection one-form. Their curvatures are

$$\mathbf{G} = d\mathbf{B} + \mathbf{B} \wedge \mathbf{B} = \frac{1}{2} G_{\mu\nu} dx^\mu \wedge dx^\nu \,,$$
$$\mathbf{R}^\mu{}_\nu = d\mathbf{\Gamma}^\mu{}_\nu + \mathbf{\Gamma}^\mu{}_\rho \wedge \mathbf{\Gamma}^\rho{}_\nu = \frac{1}{2} R^\mu{}_{\nu\rho\sigma} dx^\rho \wedge dx^\sigma \,, \tag{440}$$

with $R^\mu{}_{\nu\rho\sigma}$ the Riemann tensor. The anomalies are encoded in a formal $d + 2$ form $\mathcal{P} = \mathcal{P}[\mathbf{G}, \mathbf{R}]$ known as the anomaly polynomial. It is a polynomial in the Chern classes of $\mathbf{G}$ and Pontryagin classes of $\mathbf{R}$. The anomaly polynomial is closed,

$$d\mathcal{P} = 0 \,, \tag{441}$$

and so can be written as the derivative of a Chern-Simons form

$$\mathcal{P} = d\mathbf{I} \,. \tag{442}$$

Let $\mathcal{M}_d$ be the boundary of a $d + 1$-dimensional manifold $\mathcal{M}_{d+1}$, and extend the sources on $\mathcal{M}_d$ to sources on $\mathcal{M}_{d+1}$. Then the statement of anomaly inflow is most efficiently stated as the definition of a "covariant generating functional" $W_{E,cov}$,

$$W_{E,cov} = W_E + \int_{\mathcal{M}_{d+1}} \mathbf{I} \,, \tag{443}$$

with $W_E = -i \ln \mathcal{Z}_E$ and $\mathcal{Z}_E$ the usual Euclidean partition function. The object $W_{E,cov}$ has the virtue of being invariant under flavor gauge transformations and diffeomorphisms. The Chern-Simons form is invariant up to a boundary term, and this boundary term is precisely minus the anomalous variation of $W_E$. Writing the Chern-Simons form as a function of $\{\mathbf{B}, \mathbf{G}, \boldsymbol{\Gamma}, \mathbf{R}\}$, that variation is given by [36]

$$\delta_\chi W_E = - \int_{\mathcal{M}_d} \boldsymbol{\Lambda} \cdot \mathcal{J} + \partial_\nu \xi^\mu \mathcal{T}^\nu{}_\mu \,, \tag{444}$$

with

$$\mathcal{J} = \frac{\partial \mathbf{I}}{\partial \mathbf{B}} \,, \qquad \mathcal{T}^\nu{}_\mu = \frac{\partial \mathbf{I}}{\partial \boldsymbol{\Gamma}^\mu{}_\nu} \,. \tag{445}$$

The currents obtained by varying $W_{E,cov}$ with respect to the sources are called covariant currents. They differ from the currents obtained by using $W_E$ (known as the consistent currents) by local polynomials in the sources (known as Bardeen-Zumino polynomials) which are known and tabulated. We refer the reader to, e.g., [36] for details.

The contribution of 't Hooft anomalies to hydrostatic response was computed in detail in [36]. Here we would like to account for 't Hooft anomalies in our Schwinger-Keldysh effective actions. The reader may think of the terms we write down as Wess-Zumino terms constructed from the fluid variables. Our starting point is the Schwinger-Keldysh analogue of the inflow mechanism. Consider a theory with 't Hooft anomalies described by an anomaly polynomial $\mathcal{P}$ and a Schwinger-Keldysh generating functional $W = W[B_1, g_1; B_2, g_2]$. The 1 sources live on a manifold $\mathcal{M}_{d,1}$, and the 2 sources on a manifold $\mathcal{M}_{d,2}$. We extend the sources $A_1, g_1$ to sources on a $d+1$-dimensional manifold $\mathcal{M}_{d+1,1}$ with $\mathcal{M}_{d,1}$ as its boundary, and similarly for the 2 sources. We then define a covariant Schwinger-Keldysh generating functional $W_{cov}$ by

$$W_{cov} = W + \int_{\mathcal{M}_{d+1,1}} \mathbf{I}_1 - \int_{\mathcal{M}_{d+1,2}} \mathbf{I}_2 \,, \tag{446}$$

where $\mathbf{I}_1$ is the Chern-Simons form evaluated as a function of the 1 fields, and $\mathbf{I}_2$ is similarly defined. This way $W_{cov}$ is invariant under the doubled flavor transformations and diffeomorphisms. In the remainder of this Subsection we will write down fluid effective actions for $W_{cov}$ rather than $W$ itself. The virtue of this is that, while $W$ is not flavor and/or diffeomorphism invariant, $W_{cov}$ is, and thus our effective action for $W_{cov}$ is invariant under all of the symmetries of the problem. We can then relate the currents associated with $W_{cov}$ to those associated with $W$ by subtracting the appropriate Bardeen-Zumino polynomial.

We begin by extending the $d$-dimensional worldvolume $\mathsf{M_d}$ to a $d+1$-dimensional worldvolume $\mathsf{M_{d+1}}$ with $\mathsf{M_d}$ as its boundary. We also extend all of the worldvolume fields to fields on $\mathsf{M_{d+1}}$. In particular, we extend the thermal data $(\beta^i, \Lambda_\beta)$ to data $(\beta^I, \Lambda_\beta)$ on $\mathsf{M_{d+1}}$, the embeddings $X_1^\mu(\sigma)$, $X_2^\mu(\sigma)$ to mappings $X_1^M(\sigma)$ and $X_2^N(\sigma)$ from $\mathsf{M_{d+1}}$ to $\mathcal{M}_{d+1,1}$ and $\mathcal{M}_{d+1,2}$, and the phase fields $C_1(\sigma)$ and $C_2(\sigma)$ to phase fields on $\mathsf{M_{d+1}}$. The embeddings and phase fields allow us to pullback the extended sources on $\mathcal{M}_{d+1,i}$ to sources on $\mathsf{M_{d+1}}$, which we denote as $B_{1I}(\sigma), B_{2J}(\sigma)$, and so on. The difference in Chern-Simons terms in $W_{cov}$ (446) may also be pulled back to $\mathsf{M_{d+1}}$.

The difference of Chern-Simons forms may be split into an exact term and a bulk term,

$$\mathbf{I}_1 - \mathbf{I}_2 = d\mathbf{W}_{12} + \mathbf{V}_{12} \,. \tag{447}$$

By construction $\mathbf{W}_{12}$ is a non-invariant $d$-form and $\mathbf{V}_{12}$ is a flavor/diffeomorphism-invariant $d+1$-form. There are simple integral expressions for both $\mathbf{W}_{12}$ and $\mathbf{V}_{12}$ given in [36]. Since the covariant generating functional is invariant under flavor transformations and

diffeomorphisms, our candidate contribution of the anomaly to the effective action of the covariant Schwinger-Keldysh generating functional $W_{cov}$ is

$$S_{anom} = \int_{\mathsf{M_{d+1}}} \mathbf{V}_{12} \,. \tag{448}$$

The effective action $S_{anom}$ is a Wess-Zumino term for the anomalies: it accounts for the microscopic anomalies, and it is a topological term on a closed manifold as we presently demonstrate. On a compact manifold the action (448) is just the difference of Chern-Simons terms, $S_{anom} = \int \mathbf{I}_1 - \mathbf{I}_2$. The flavor and spin currents that follow are [36],

$$\star \mathbf{J}_1 = \frac{\partial \mathcal{P}_1}{\partial \mathbf{B}_1} \,, \qquad \star \mathbf{L}^N{}_M = \frac{\partial \mathcal{P}_1}{\partial \mathbf{R}^M{}_N} \,, \tag{449}$$

and similarly for the 2 fields. Here $\star$ is the Hodge star operator, and the stress tensor is given by suitable derivatives of the spin current. These currents are identically conserved on account of the flavor and diffeomorphism invariance of the action $S_{anom}$ [36]. However, on a compact manifold the equations of motion for the $X$ and $C$-fields are the conservation of these currents, and so those equations of motion are automatically satisfied.

Thus, despite extending the various fields from $d$-dimensional ones on $\mathsf{M_d}$ to $d+1$-dimensional ones on $\mathsf{M_{d+1}}$, the equations of motion for $X$ and $C$ that follow from $S_{anom}$ only have boundary terms. There are no $d+1$-dimensional fluid degrees of freedom. Moreover, the equations of motion for the dynamical fields are precisely the anomalous Ward identities of the anomalous microscopic theory. (More precisely the equations of motion become the Ward identities for the covariant stress tensor and flavor current. See [36].) That is, the anomaly action (448) correctly accounts for the microscopic anomalies (as explicitly demonstrated in [44] for non-gravitational anomalies).

The action (448) respects the (bosonic) Schwinger-Keldysh symmetry as well as the reality condition. In the absence of ghosts, the Schwinger-Keldysh symmetry is the statement that the action vanishes when setting the $a$-fields to vanish, which the action $S_{anom}$ clearly does on account of (447). The action is purely real and odd under exchanging 1 and 2 fields, and so it respects the reality condition.

We presently take the statistical mechanical limit of the anomaly action, and then supersymmetrize it. Before proceeding, it is expedient to break up the action (448) into a form that appears to be more complicated, but which will be useful when supersymmetrizing. We define independent velocities, temperatures, and chemical potentials on $\mathsf{M_{d+1}}$ via

$$
\begin{aligned}
u_{1I} &= T_1 g_{1IJ} \beta^J \,, \\
T_1 &= \frac{1}{\sqrt{-g_{1IJ}\beta^I \beta^J}} \,, \\
\mu_1 &= T\left(\beta^I B_{1I} + \Lambda_\beta\right) \,, \\
(\mu_{R1})^I{}_J &= T_1 (D_1)_J \beta^I \,,
\end{aligned}
\tag{450}
$$

and similarly for the 2 fields. Here $\mu_R$ is the spin chemical potential [36], the gravitational analogue of the flavor chemical potential $\mu$ and $D_1$ the covariant derivative with respect to $g_{1IJ}$. We also define "hatted" connections and curvatures,

$$
\begin{aligned}
\hat{\mathbf{B}}_1 &= \mathbf{B}_1 + \mu_1 \mathbf{u}_1 \,, \\
\hat{\mathbf{G}}_1 &= d\hat{\mathbf{B}}_1 + \hat{\mathbf{B}}_1 \wedge \hat{\mathbf{B}}_1 \,, \\
(\hat{\mathbf{\Gamma}}_1)^I{}_J &= (\mathbf{\Gamma}_1)^I{}_J + (\mu_{R1})^I{}_J \mathbf{u}_1 \,, \\
(\hat{\mathbf{R}}_1)^I{}_J &= d(\hat{\mathbf{\Gamma}}_1)^I{}_J + (\hat{\mathbf{\Gamma}}_1)^I{}_K \wedge (\hat{\mathbf{\Gamma}}_1)^K{}_J \,.
\end{aligned}
\tag{451}
$$

We then decompose

$$\mathbf{I}_1 - \mathbf{I}_2 = \left(\mathbf{I}_1 - \hat{\mathbf{I}}_1\right) - \left(\mathbf{I}_2 - \hat{\mathbf{I}}_2\right) + \left(\hat{\mathbf{I}}_1 - \hat{\mathbf{I}}_2\right), \tag{452}$$

where $\hat{I}_1$ is the Chern-Simons form evaluated on the "hatted" 1 fields and similarly for $\hat{I}_2$. Each term in parentheses can be written as the sum of a non-invariant boundary term and an invariant bulk term, i.e.

$$\begin{aligned}
\mathbf{I}_1 - \hat{\mathbf{I}}_1 &= d\mathbf{W}_1 + \mathbf{V}_1, \\
\mathbf{I}_2 - \hat{\mathbf{I}}_2 &= d\mathbf{W}_2 + \mathbf{V}_2, \\
\hat{\mathbf{I}}_1 - \hat{\mathbf{I}}_2 &= d\hat{\mathbf{W}} + \hat{\mathbf{V}},
\end{aligned} \tag{453}$$

and so

$$S_{anom} = \int_{\mathsf{M}_{d+1}} \left(\mathbf{V}_1 - \mathbf{V}_2 + \hat{\mathbf{V}}\right). \tag{454}$$

The $\mathbf{V}_1$ and $\mathbf{V}_2$ are relatively simple and are closely related to the anomaly effective action in hydrostatics. Decomposing the derivative of $\mathbf{u}_1$ into longitudinal and transverse parts,

$$d\mathbf{u}_1 = -\mathbf{u}_1 \wedge \mathbf{a}_1 + 2\boldsymbol{\omega}_1, \tag{455}$$

where $(\omega_1)_{IJ}(u_1)^J = 0$, they are [36]

$$\mathbf{V}_1 = \frac{\mathbf{u}_1}{2\boldsymbol{\omega}_1} \wedge \left(\mathcal{P}_1 - \hat{\mathcal{P}}_1\right), \tag{456}$$

and similarly for $\mathbf{V}_2$. In equation (456) we have divided by a 2-form which may seem a bit jarring at first sight. This type of notation is explained in detail in [36]. The gist is that the difference $\mathbf{u}_1 \wedge (\mathcal{P}_1 - \hat{\mathcal{P}}_1)$ is a sum of terms of the form $(2\boldsymbol{\omega}_1)^{n+1} \wedge \mathbf{V}_n$ with $n = 0, 1, \dots$ Dividing by $2\boldsymbol{\omega}_1$ is an instruction to replace $(2\boldsymbol{\omega}_1)^{n+1} \wedge \mathbf{V}_n$ with $(2\boldsymbol{\omega}_1)^n \wedge \mathbf{V}_n$.

Let us now take the $\hbar \to 0$ limit. Recall that in this limit we take the $a$-type fields to be $O(\hbar)$. Because the coefficients of the anomaly polynomials are numbers, the effect of the $\hbar \to 0$ limit is, in terms of an expansion in $a$-type fields, to only keep the leading $O(a)$ term in the action. A simple computation gives that

$$S_A = \lim_{\hbar \to 0} \frac{S_{anom}}{\hbar} = \int_{\mathsf{M}_{d+1}} \left(\frac{\delta \mathbf{V}_r}{\delta B_{r\,I}} B_{a\,I} + \frac{\delta \mathbf{V}_r}{\delta g_{r\,IJ}} g_{a\,IJ} + \widehat{\mathbf{B}}_a \wedge \frac{\partial \widehat{\mathcal{P}}_r}{\partial \widehat{\mathbf{G}}_r} + (\widehat{\boldsymbol{\Gamma}}_a)^I{}_J \wedge \frac{\partial \widehat{\mathcal{P}}_r}{\partial (\widehat{\mathbf{R}}_r)^I{}_J}\right). \tag{457}$$

Here we have defined $\mathbf{V}_r$ to be the transgression form $\mathbf{V}$ in (456) evaluated as a function of the $r$-fields, and similarly $\widehat{\mathcal{P}}_r$ to be the anomaly polynomial $\mathcal{P}$ as a function of the hatted $r$-fields. The first two terms come from the $\hbar \to 0$ limit of $\mathbf{V}_1 - \mathbf{V}_2$, and the last two from $\widehat{\mathbf{V}}_{12}$, upon using the explicit formulae for transgression forms given in [36]. These last two terms may be further simplified, by using that

$$\beta^I \widehat{B}_{a\,I} = 0, \qquad \beta^K (\widehat{\Gamma}_a)^I{}_{JK} = 0, \tag{458}$$

and

$$\hat{G}_{r\,IJ}\beta^J = -\delta_\beta \widehat{B}_{r\,I}, \qquad (\hat{R}_r)^I{}_{JKL}\beta^L = -\delta_\beta (\widehat{\Gamma}_r)^I{}_{JK}. \tag{459}$$

Because the integrand must have one leg in the $\beta$ direction and $\widehat{\mathbf{B}}_a$ and $\widehat{\boldsymbol{\Gamma}}$ do not, it follows that

$$\begin{aligned}
S_A = \int_{\mathsf{M}_{d+1}} \Bigg\{ &\frac{\delta \mathbf{V}_r}{\delta B_{r\,I}} B_{a\,I} + \frac{\delta \mathbf{V}_r}{\delta g_{r\,IJ}} g_{a\,IJ} \\
&- iT_r \mathbf{u}_r \wedge \Big( \frac{\partial^2 \widehat{\mathcal{P}}_r}{\partial \widehat{\mathbf{G}}_r^2} \wedge \widetilde{\widehat{\mathbf{B}}}_a \wedge \widehat{\mathbf{B}}_a + \frac{\partial^2 \widehat{\mathcal{P}}_r}{\partial (\widehat{\mathbf{R}}_r)^I{}_J \partial (\widehat{\mathbf{R}}_r)^K{}_L} \wedge (\widetilde{\widehat{\boldsymbol{\Gamma}}}_a)^I{}_J \wedge (\widehat{\boldsymbol{\Gamma}}_a)^K{}_L \\
&+ \frac{\partial^2 \widehat{\mathcal{P}}_r}{\partial \widehat{\mathbf{G}}_r \partial (\widehat{\mathbf{R}}_r)^I{}_J} \wedge (\widetilde{\widehat{\mathbf{B}}}_a \wedge (\widehat{\boldsymbol{\Gamma}}_a)^I{}_J + (\widetilde{\widehat{\boldsymbol{\Gamma}}}_a)^I{}_J \wedge \widehat{\mathbf{B}}_a) \Big) \Bigg\},
\end{aligned} \tag{460}$$

where

$$\widetilde{\widehat{\mathbf{B}}}_a = \widehat{\mathbf{B}}_a + i\delta_\beta \widehat{\mathbf{B}}_r \,, \qquad (\widetilde{\widehat{\mathbf{\Gamma}}}_a)^I{}_J = (\widehat{\mathbf{\Gamma}}_a)^I{}_J + i\delta_\beta (\widehat{\mathbf{\Gamma}}_r)^I{}_J \,. \tag{461}$$

The virtue of writing the bosonic anomaly action is this. We recognize the first line as a scalar term, and the rest, because it is linear in both an $a$-field and a $\tilde{a}$-field, as a tensor term. Written this way it is straightforward to verify that the action respects the bosonic KMS symmetry, as well as to supersymmetrize the bosonic action, which we do now.

In addition to extending the thermal data and bosonic fields to fields on $\mathsf{M}_{\mathsf{d+1}}$, we extend the ghost partners to ghosts on $\mathsf{M}_{\mathsf{d+1}}$. As above, we group the pullback fields into superfields as

$$\begin{aligned}
\mathbb{X}^M &= X_r^M + \bar{\theta} X_{\bar{g}}^M + \theta X_g^M + \bar{\theta}\theta X_a^M \,, \\
\mathbb{C} &= C_r + \bar{\theta} C_{\bar{g}} + \theta C_g + \bar{\theta}\theta C_a \,, \\
\mathbb{B}_I &= (B_{r\,M}(\mathbb{X}) + \bar{\theta}\theta B_{a\,M}(\mathbb{X}))\partial_I \mathbb{X}^M + \partial_I \mathbb{C} \,, \\
\mathfrak{g}_{IJ} &= \left( g_{r\,MN}(\mathbb{X}) + \bar{\theta}\theta g_{a\,MN}(\mathbb{X}) \right) \partial_I \mathbb{X}^M \partial_J \mathbb{X}^N \,, \\
\mathbb{\Gamma}^I{}_{JK} &= \frac{1}{2}\mathfrak{g}^{IL} \left( \partial_J \mathfrak{g}_{KL} + \partial_K \mathfrak{g}_{JL} - \partial_L \mathfrak{g}_{JK} \right) \,,
\end{aligned} \tag{462}$$

along with supercurvatures $\mathbb{G}_{IJ}$ and $\mathbb{R}^I{}_{JKL}$. We also construct a fluid velocity, temperature, chemical potential, and spin chemical potential superfield via along with supercurvatures $\mathbb{G}_{IJ}$ and $\mathbb{R}^I{}_{JKL}$. We also construct a fluid velocity, temperature, chemical potential, and spin chemical potential superfield via

$$\begin{aligned}
\mathfrak{u}^I &= \mathbb{T}\beta^I \,, \\
\mathbb{T} &= \frac{1}{\sqrt{-\mathfrak{g}_{IJ}\beta^I\beta^J}} \,, \\
\mathbb{\mu} &= \mathbb{T}(\beta^I \mathbb{B}_I + \Lambda_\beta) \,, \\
(\mathbb{\mu}_R)^I{}_J &= \mathbb{T}\mathbb{W}_J\beta^I \,,
\end{aligned} \tag{463}$$

as well as hatted superconnections $\hat{\mathbb{B}}_I$ and $\hat{\mathbb{\Gamma}}^I{}_{JK}$ and hatted supercurvatures $\hat{\mathbb{G}}_{IJ}$ and $\hat{\mathbb{R}}^I{}_{JKL}$. We define the super anomaly polynomial $\mathbb{P}$ and its hatted version $\hat{\mathbb{P}}$ via

$$\mathbb{P} = \mathcal{P}(\mathbb{G}, \mathbb{R}^I{}_J) \,, \qquad \hat{\mathbb{P}} = \mathcal{P}(\hat{\mathbb{G}}, \hat{\mathbb{R}}^I{}_J) \,, \tag{464}$$

along with a super transgression form $\mathbb{V}$,

$$\mathbb{V} = \frac{\mathfrak{u}}{2\mathfrak{w}} \wedge \left( \mathbb{P} - \hat{\mathbb{P}} \right) \,. \tag{465}$$

The supersymmetrization of $S_A$ in (460) is then

$$\begin{aligned}
S_A = \int_{\mathsf{M}_{\mathsf{d+1}}} d\theta d\bar{\theta} \Big\{ \boldsymbol{V} - i\mathbb{T}\mathfrak{u} \wedge \Big( & \frac{\partial^2 \hat{\mathbb{P}}}{\partial \hat{\mathbb{G}}^2} \wedge D_\theta \hat{\mathbb{G}} \wedge D_{\bar{\theta}} \hat{\mathbb{G}} + \frac{\partial^2 \hat{\mathbb{P}}}{\partial \hat{\mathbb{R}}^I{}_J \partial \hat{\mathbb{R}}^K{}_L} \wedge D_\theta \hat{\mathbb{\Gamma}}^I{}_J \wedge D_{\bar{\theta}} \hat{\mathbb{\Gamma}}^K{}_L \\
& + \frac{\partial^2 \hat{\mathbb{P}}}{\partial \hat{\mathbb{G}} \partial \hat{\mathbb{R}}^I{}_J} \wedge (D_\theta \hat{\mathbb{B}} \wedge D_{\bar{\theta}} \hat{\mathbb{\Gamma}}^I{}_J + D_\theta \hat{\mathbb{\Gamma}}^I{}_J \wedge D_{\bar{\theta}} \hat{\mathbb{B}}) \Big) \Big\} \,,
\end{aligned} \tag{466}$$

which is the main result of this Appendix.

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
