# Peer review of "A panoply of Schwinger-Keldysh transport"

_SciPost Physics, doi:SciPost Phys. 5, 053 (2018)_

## Round 2 · Referee Report · Anonymous (Referee 1) · 2018-9-25

Strengths

The paper addresses an important and interesting topic which has attracted much recent attention. While many papers before have attempted the same calculation and performed large parts of it, the analysis here goes further, which is apparent from the fact that the authors find a new family of constraints on hydrodynamic transport coefficients. This is an important new consequence of unitarity. Despite its length, the paper is written clearly and contains a large body of background material, which will be very useful to readers less familiar with the action principle approach to hydrodynamics.

Weaknesses

The paper has very few weaknesses. In my opinion, the only weakness is the absence of a transparent discussion of implications of the new family of constraints, ideally with examples. This is most likely because such results are lacking. It would also be nice to see this analysis done without invoking the superspace approach, especially because such an approach is unnecessary and can obscure the physics, even though it may sometimes be useful.

Report

The paper addresses a difficult and long-standing question of classifying hydrodynamic transport. While most results derived in this paper were known in some form or another from previous works, it is useful to see them rederived here. Beyond that, the precision of the analysis preformed here enabled the authors to find a seemingly completely new family of constraints, which also follow from unitarity and the condition that $\text{Im}(S) \geq 0$, where $S$ is the complex Schwinger-Keldysh action. This is in itself sufficiently interesting for the paper to be published. Moreover, given that the paper is written very clearly, is well structured and nicely formatted, I can recommend for the paper to be published without any changes, should the authors decide to do so (see requested changes). It is somewhat unfortunate that the new constraints seem very weak, but this cannot be held against the authors who could not have known this before embarking on their formidable journey. For this very reason, it would be great to see the implications of this new constraint discussed more clearly, ideally with some examples, although such examples might be hard to find. Also, it would be interesting to understand whether there is nevertheless some historic precursor to this new type of constraint, again in reference to examples.

Requested changes

If the authors feel like it, I would suggest extending the discussion of the expected applicability and implications of the new constraint. However, I do not think that this change is compulsory in order for the paper to be published.

---

## Round 2 · Referee Report · Anonymous (Referee 2) · 2018-10-22

Strengths

While long, this paper is well-written and consistently to the point. Besides the main result (discussed below), it provides a useful introduction and overview of recent progress in formulating a consistent action principle for hydrodynamics. The key ingredients -- doubled diffeomorphism invariance, topological Schwinger-Keldysh symmetry, a reality condition on the generating functional, and KMS symmetry -- are clearly introduced, and their implementation leads to a field theory formalism of considerable sophistication and intrinsic interest.

Weaknesses

Naturally at this stage, it is not yet clear how widespread a role the new Schwinger-Keldysh positivity constraints play. As the authors note, the main example in which an exceptional transport coefficient vanishes requires a highly atypical situation in which the ordinary conductivities and viscosities vanish. Nevertheless, with further study it seems likely that a wider range of examples will emerge enabling a better appreciation of the significance of the new constraints.

One might wonder though whether the Schwinger-Keldysh positivity condition can ultimately be re-formulated in terms of the usual on-shell phenomenological approach. (After all, this is possible for the existing constraints from CPT invariance and entropy production.) While indeed it seems unlikely this can be done, it would nevertheless be very attractive to have a more elementary (if less systematic) approach.

Report

In the traditional phenomenological approach to hydrodynamics, constraints on transport coefficients are derived by requiring that the entropy production is non-negative and that CPT invariance is respected at the microscopic level. More recently, there has been considerable interest in re-formulating hydrodynamics in terms of an action principle, offering a more systematic treatment of such constraints as well as the possibility of incorporating quantum fluctuations. An action-based approach also raises the exciting possibility of identifying new constraints on transport coefficients missed by the standard phenomenological treatment, and indeed just such a claim is made in this paper. This is an important result, and provides strong justification for the considerable technical effort required to set up the action formalism.

The new constraints derive from unitarity in the form of the Schwinger-Keldysh positivity condition on the effective action first identified in [14,25]. Here, the authors show that positivity of the entropy production is a necessary but not sufficient condition for Schwinger-Keldysh positivity: that there are specific classes of transport coefficients for which the entropy production vanishes and Schwinger-Keldysh positivity must be separately imposed. In the process, the authors introduce an attractive classification scheme broadly similar to that of [26,27] based on whether the transport coefficients are (a) associated with entropy production, (b) are constrained by Schwinger-Keldysh positivity, (c) their parity under KMS symmetry, and (d) the (anti)symmetry of their tensorial index structure. This scheme is summarised in table 2 on page 76.

Overall, identifying Schwinger-Keldysh positivity as a source of new constraints on transport coefficients appears to be a significant original result, especially given the long history of the hydrodynamics. It also raises very interesting questions: in particular, whether this condition has a geometrical holographic interpretation in the same way that positivity of the fluid entropy production is tied to the area increase of the bulk black hole. I recommend this impressive work for publication.

Requested changes

Table 2 and the summary of the classification scheme in section 8.1 are very helpful in orienting the reader, and many might benefit from skimming this material prior to tackling the main text. Perhaps this material should be more strongly advertised in the last paragraph of the introduction?

The second sentence of section 6.2.4 is rather confusing: it seems the aim is to explain the difference between the pseudo-dissipative and exceptional terms, however the justification given in brackets is the same in both cases. In the main classification, the distinction between these classes derives from their KMS parity and the $\sigma$-symmetry of the index structure.

In the final paragraph of p76 and the first of page 77, the references to the columns in table 2 are mislabelled: the initial reference to the second column is correct, then all the following column references need to be incremented by one (second->third, etc).

Some typos:

p 50, penultimate line: "various transport into scalar terms" -> "transport coefficients" (?)
Second line of section 7: "we consider two more complicated examples" -> omitting the hyphen here leads to a possibly unintended misreading!
Below (7.10): "allow one" -> "allows one"
Eq (7.78a): a stray comma in penultimate line?
Table 2 caption: missing period at end.
p76 last para: "time the KMS" -> "times the KMS"
p79, second para: "One can not" -> "One cannot"
p79, third para: "then captured" -> "than captured"

---

## Round 3 · Author Response

We would like to thank both of the Referees for their comments and careful reading of our manuscript. We have fixed the various typos pointed out by Referee 2, as well as implemented the other two suggestions made.

---

## Round 3 · List of Changes

- We fixed the typos noted by Referee 2.
- We added a brief comment to the end of the Introduction referring the curious reader immediately to our summary and table in Section 8.1.
- We fixed the (incorrect as written) second sentence in Section 6.2.4.

---

## Editorial Decision

published